# Systematic discovery of CRISPR-boosted CAR T cell immunotherapies

Paul Datlinger[1,3,4 ✉], Eugenia V. Pankevich[1,4], Cosmas D. Arnold[1,4], Nicole Pranckevicius[1], Jenny Lin[1], Daria Romanovskaia[1,2], Moritz Schaefer[1,2], Francesco Piras[1,2], Anne-Christine Orts[1], Amelie Nemc[1], Paulina N. Biesaga[1], Michelle Chan[1], Teresa Neuwirth[1], Artem V. Artemov[2], Wentao Li[1], Sabrina Ladstätter[1], Thomas Krausgruber[1,2] & Christoph Bock[1,2 ✉]

Chimeric antigen receptor (CAR) T cell therapy has shown remarkable success in treating blood cancers, but CAR T cell dysfunction remains a common cause of treatment failure[1]. Here we present CELLFIE, a CRISPR screening platform for enhancing CAR T cells across multiple clinical objectives. We performed genome-wide screens in human primary CAR T cells, with readouts capturing key aspects of T cell biology, including proliferation, target cell recognition, activation, apoptosis and fratricide, and exhaustion. Screening hits were prioritized using a new in vivo CROP-seq[2] method in a xenograft model of human leukaemia, establishing several gene knockouts that boost CAR T cell efficacy. Most notably, we discovered that *RHOG* knockout is a potent and unexpected CAR T cell enhancer, both individually and together with *FAS* knockout, which was validated across multiple in vivo models, CAR designs and sample donors, and in patient-derived cells. Demonstrating the versatility of the CELLFIE platform, we also conducted combinatorial CRISPR screens to identify synergistic gene pairs and saturation base-editing screens to characterize *RHOG* variants. In summary, we discovered, validated and biologically characterized CRISPR-boosted CAR T cells that outperform standard CAR T cells in widely used benchmarks, establishing a foundational resource for optimizing cell-based immunotherapies.

CAR T cell therapy is a groundbreaking advance in cancer treatment and a convincing proof of concept for genetically engineered cells as therapeutics. Beyond clinically approved products for blood cancers[3,4], CAR T cells hold great potential for solid tumours[5,6], autoimmune diseases[7] and tissue regeneration[8]. However, clinical trials have uncovered major obstacles to CAR T cell efficacy[9], including limited T cell fitness and proliferation, insufficient T cells in pretreated patients with cancer, exhaustion from chronic CAR stimulation, immunosuppressive and nutrient-poor tumour environments, and fratricide when CAR T cells acquire antigen from their target cells via trogocytosis[10].

This study pursues the idea that barriers to CAR T cell efficacy can be overcome by disabling biological functions that are important for T cells over a human lifetime but hinder short-term CAR T cell efficacy. Unlike T cells, which have been optimized by evolution, CAR T cells are products of cell engineering and lack evolutionary optimization. Hence, certain genes essential for T cell function may actively impair the therapeutic potential of CAR T cells. This concept is supported by the surprising success of a TET2-deficient CAR T cell clone in a patient with leukaemia[11] and by studies showing that genetic perturbations can enhance CAR T cell performance, for example, by knocking out genes encoding immune checkpoints (PD-1 and CTLA4)[12,13], epigenetic regulators (PRDM1)[14], signalling proteins (RASA2)[15] and mediator subunits

(MED12 and CCNC)[16] or by overexpressing transcription factors (c-Jun and FOXO1)[17–19].

Given that CAR T cells are genetically engineered medicinal products, CRISPR screening offers a natural and clinically translatable strategy to optimize and adapt them for diverse applications[16,20,21], profiting from recent advances in genetic screening for human primary T cells[15,22–24]. We propose that genome-wide screens with clinically relevant readouts enable artificial evolution of CAR T cells to identify perturbations that boost their efficacy.

To support the systematic discovery of CRISPR-boosted CAR T cell immunotherapies, we developed the CELLFIE platform for high-content CRISPR screening[25] in human primary CAR T cells. CELLFIE resolves important experimental challenges including efficient co-delivery of CAR, guide RNA (gRNA) library and CRISPR editor; screenable readouts that capture clinical limitations; and the cost and scale of genome-wide screens. We further adapted our CROP-seq method[2] for in vivo screening and identified *RHOG*, *PRDM1* and *FAS* knockouts as boosters of CAR T cell efficacy in a xenograft model of human leukaemia. *RHOG* is an unexpected discovery, as RHOG deficiency causes a monogenic immunodeficiency in humans[26], underlining the different biological realities of T cells and CAR T cells. Combinatorial screens and individual validations further established that *RHOG*-and-*FAS* double knockout

[1]CeMM Research Center for Molecular Medicine of the Austrian Academy of Sciences, Vienna, Austria. [2]Medical University of Vienna, Institute of Artificial Intelligence, Center for Medical Data Science, Vienna, Austria. [3]Present address: Arc Institute, Palo Alto, CA, USA. [4]These authors contributed equally: Paul Datlinger, Eugenia V. Pankevich, Cosmas D. Arnold. ✉e-mail: paul.datlinger@cellfie.org; cbock@cemm.oeaw.ac.at

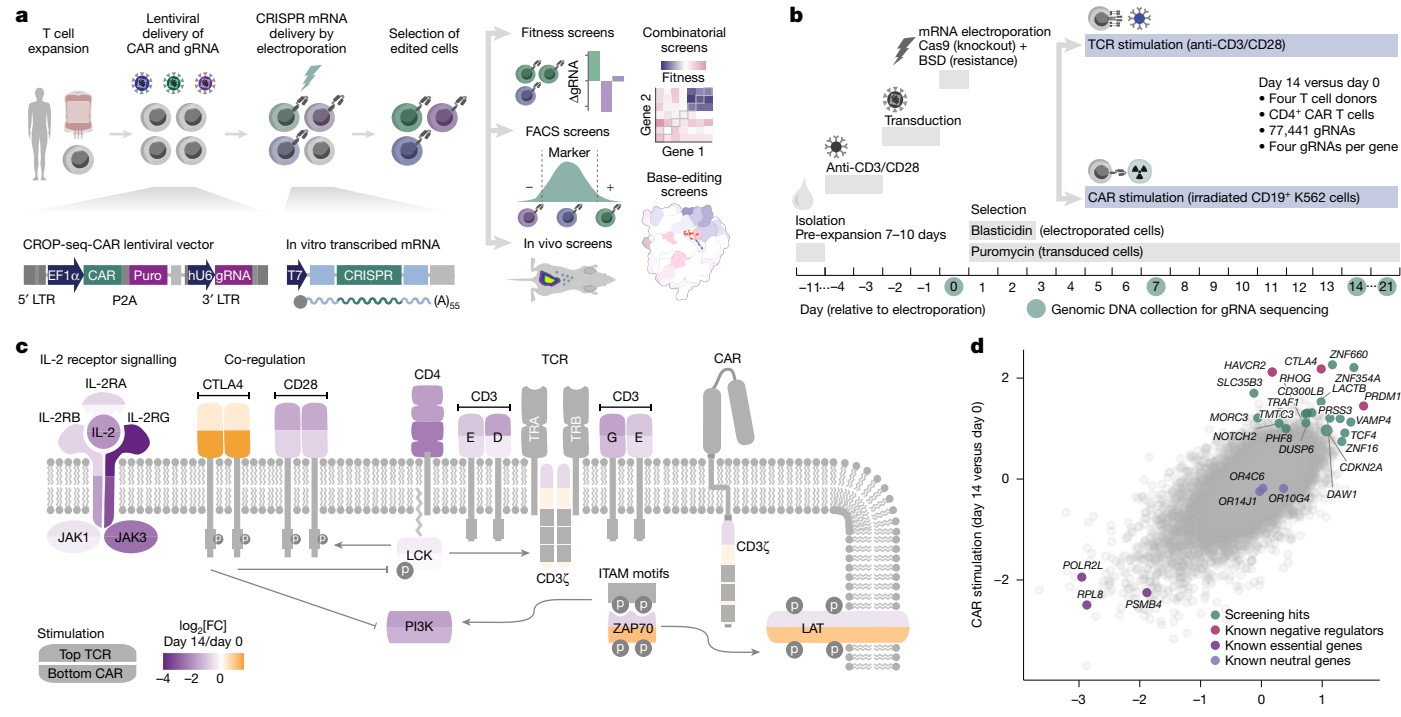

**Fig. 1 | Genome-wide fitness screens in human primary CAR T cells with the CELLFIE platform. a**, Overview of the CELLFIE platform for highly scalable CAR T cell engineering and CRISPR screening. Human primary T cells are isolated, activated and expanded, transduced with a lentivirus carrying sequences for the CAR and gRNA library and electroporated with synthetic mRNA delivering the CRISPR editor. The CRISPR-edited CAR T cells are functionally screened in vitro and in vivo using multiple readouts. hU6, human U6 promoter; LTR, long terminal repeat; puro, puromycin. **b**, Experimental timeline for genome-wide fitness screens. Human primary T cells are isolated from whole blood, activated, pre-expanded, activated again and transduced with the CROP-seq-CAR lentivirus for co-delivery of sequences for the CAR and the genome-wide gRNA library. Two days later, cells are electroporated with synthetic mRNA for Cas9 and blasticidin resistance (blasticidin-S deaminase, BSD), followed by antibiotic selection for successful lentiviral transduction (puromycin) and successful mRNA electroporation (blasticidin). CRISPR-edited CAR T cells are

expanded under repeated TCR stimulation with anti-CD3/CD28 beads or repeated CAR stimulation with CD19⁺ K562 cells. Genomic DNA is collected, and gRNA representation is analysed by sequencing on days 0, 7, 14 and 21. **c**, Gene-level log₂[fold change (FC)] between day 14 and day 0 for the fitness screens (four donors in two independent screens), mapped onto a schematic of the TCR and CAR signalling pathways. For each protein shape, the colour in the top half corresponds to the TCR stimulation screens and the colour in the bottom half corresponds to the CAR stimulation screens. **d**, Effect sizes for fitness screens with TCR stimulation ($x$ axis) and CAR stimulation ($y$ axis). MAGeCK MLE $\beta$ values comparing day 14 and day 0 are normalized to values for essential genes to account for the different proliferation rates upon TCR or CAR stimulation. Colours denote screening hits with increased fitness (green), known negative T cell regulators (magenta), known essential genes (purple) and neutral olfactory receptors (blue).

strongly enhances anti-tumour activity. Finally, we performed tiling base-editing screens in human primary CAR T cells to map functional *RHOG* mutations and identify missense variants for safe clinical translation without double-strand breaks.

In summary, CELLFIE provides an integrated platform for CAR T cell engineering and CRISPR screening in primary immune cells, enabling genome-wide discovery, in vivo pooled validation, combinatorial knockout screening and base editing for preclinical development. This platform allowed us to create a major resource of CRISPR screens in CAR T cells, on the basis of which we identified multiple gene edits that enhance CAR T cell performance, culminating in the discovery of *RHOG* knockout as a potent CAR T cell booster.

## CELLFIE platform for CAR T cell screening

Our CELLFIE platform (cell engineering for immunotherapy enhancement) uses human primary T cells as starting material and introduces three components to enable CAR T cell screening: (1) a CAR; (2) a CRISPR editor for genome engineering; and (3) a library of gRNAs encoding genetic perturbations (Fig. 1a). We developed the CROP-seq-CAR vector to co-deliver sequences for the CAR and gRNA with a single lentivirus that supports high CAR expression for effective cancer cell killing and a sequencing-based readout of gRNA-encoded perturbations. For most

screens in this study, we used a 19-BBz CAR that closely resembles tisagenlecleucel (Kymriah), which is approved for treating B cell leukaemia and lymphoma. Importantly, our vector was designed for flexibility and modularity, enabling straightforward exchange for other CARs and T cell receptors (TCRs; Supplementary Table 1).

CELLFIE employs messenger RNA (mRNA) technology[27] to deliver CRISPR editors, which avoids inefficient lentivirus production and delivery of large transgenes. Compared with recombinant proteins, mRNA is more versatile, given the limited commercial availability of CRISPR editors other than Cas9 and the high setup cost for custom protein purification. We developed an optimized workflow to produce diverse CRISPR editors as electroporation-ready mRNA (Extended Data Fig. 1a) and designed mRNAs for a range of CRISPR editors and selection markers, establishing a versatile toolbox for genome engineering in CAR T cells (Extended Data Fig. 1b–h and Supplementary Table 1).

To quantify the efficiency of CRISPR knockout, we transduced human primary T cells with a single gRNA targeting the *IL2RA* gene (which codes for CD25) and electroporated *Cas9* mRNA. Genome-editing efficiency exceeded 80% across multiple blood donors, as measured on the DNA level by amplicon sequencing and on the protein level by flow cytometry (Extended Data Fig. 1d). Our custom-made mRNA achieved consistent editing efficiency comparable with that of commercially

available TriLink *Cas9* mRNA, at a tenfold lower cost (Extended Data Figs. 1e and 2a,b).

The CELLFIE platform supports various CRISPR modalities including knockout, base editing and activation, enabled by a flexible mRNA production and delivery workflow. For base editing, we tested the A-to-G editor ABEmax and the C-to-T editor AncBE4max[28] and their near-protospacer adjacent motif (PAM)-less variants based on SpRY[29]. We assessed multiple genomic loci and observed base-editing efficiencies above 75% for A-to-G editors and up to 50% for C-to-T editors (Extended Data Fig. 1f), introducing precise mutations within a narrow editing window (Extended Data Fig. 1g). Our platform is also compatible with CRISPR-based epigenetic editing and modulation of gene regulation. In particular, we tested CRISPR activation using the Synergistic Activation Mediator (SAM) system[30] and achieved strong activation of CD34 at the protein level in up to 50% of CAR T cells (Extended Data Fig. 1h).

To enable genome-wide, multi-readout CRISPR screens using CELLFIE, we performed extensive optimization experiments (Supplementary Note), which established highly efficient CAR T cell culture, cell stimulation, lentiviral transduction, gRNA integration, mRNA electroporation and cell selection (Extended Data Fig. 2c–q).

## Genome-wide screens of CAR T cell fitness

Tackling cell-intrinsic obstacles to CAR T cell function, we performed genome-wide CRISPR screens for CAR T cell fitness (Fig. 1b). Human primary T cells were stimulated with antibodies against CD3 and CD28 (anti-CD3/CD28) and expanded for 7–10 days, followed by restimulation, lentiviral transduction and electroporation of *Cas9* mRNA together with mRNA conferring blasticidin resistance. Cells were then subjected to antibiotic selection for successful lentiviral transduction and successful mRNA electroporation. We validated this proliferation-based fitness screening setup in a pilot experiment using a library of positive controls (gRNAs targeting the puromycin resistance gene *PAC*) and negative controls (gRNAs targeting a safe harbour locus). Reassuringly, we observed strong and selective depletion of the positive control gRNAs during culture with puromycin, both in CD4[+] and CD8[+] CAR T cells (Extended Data Fig. 3a).

For genome-wide CRISPR knockout screening, we cloned the widely used Brunello gRNA library[31] into the CROP-seq-CAR vector, confirmed balanced gRNA representation (Extended Data Fig. 3b) and conducted screens with CAR T cells derived from four blood donors (Extended Data Fig. 3c). After expansion and genome editing, the CAR T cells were stimulated either via their endogenous TCR using anti-CD3/CD28 beads or via their transduced CAR through repeated exposure to CD19[+] K562 cancer cells (Supplementary Fig. 1 and Supplementary Table 2).

Our screens recapitulate important cell-intrinsic limitations of CAR T cells. First, we observed slower expansion upon CAR stimulation than upon TCR stimulation (Extended Data Fig. 3d), consistent with the lower effectiveness of CAR signalling[32]. Second, we detected high levels of CD19 antigen on the CAR T cells, which can be explained by acquisition of membrane components from their target cells through trogocytosis (Extended Data Fig. 3e). This causes killing by other CAR T cells (fratricide), a known challenge in CAR T cell therapy[10]. Third, the CAR T cells upregulated certain markers of T cell exhaustion, including PD-1, LAG3, TIM3 and TIGIT (Extended Data Fig. 3f), which has been linked to poor clinical performance[33]. These observations support the idea that our genome-wide screens of CAR T cell fitness can identify gene knockouts that enhance CAR T cell function across multiple clinically relevant dimensions.

The technical quality of our genome-wide CRISPR screens was consistently high, with stronger depletion of essential genes than in prior screens with human primary T cells[15,20,22] (Extended Data Fig. 4a–f and Supplementary Table 4). Together with the dataset from

Freitas et al.[16], our screens approach the quality achieved in the Jurkat cell line[34], despite the challenges of screening in human primary cells compared to screening in a cancer cell line (Extended Data Fig. 4f). Our screening hits were enriched for immune cell functions, including T cell activation, differentiation and apoptosis, and for general cell functions such as cell cycle, splicing, RNA transport and translation (Extended Data Fig. 3h,i). Focusing on the TCR pathway (Fig. 1c), both TCR and CAR stimulation required signalling through the interleukin (IL)-2 receptor, including IL-2RG, JAK3 and the co-receptor CD4. Our screens also captured more subtle aspects of TCR signalling, including positive and negative regulation. For example, the positive co-receptor CD28 was strongly depleted, whereas its negative counterpart CTLA4 was enriched. As expected, the TCR subunits CD3E and CD3D were essential only in TCR-stimulated samples, in which they are activated by the bead-bound anti-CD3 antibody.

To identify potential boosters of T cell or CAR T cell therapies, we analysed our screening data with the MAGeCK maximum likelihood estimation (MLE) algorithm[35,36] and selected knockouts with strong enrichment upon stimulation of either the CAR or both the CAR and the TCR (Fig. 1d). Supporting the quality of our dataset, this analysis rediscovered CTLA4, a well-established immune checkpoint inhibitor, and PRDM1, an epigenetic factor implicated in terminal T cell differentiation and a known suppressor of T cell activity. Both *CTLA4* and *PRDM1* knockouts have been reported to enhance CAR T cell efficacy[13,14]. Our fitness screens also uncovered multiple new hits for genetic enhancement of CAR T cells (Fig. 1d).

## Genome-wide FACS screens of CAR T cell biology

To broaden our coverage of human CAR T cell biology, we established fluorescence-activated-cell-sorting (FACS)-based screening readouts for four biological processes that constitute clinical limitations of CAR T cell therapy (Fig. 2a and Extended Data Fig. 5a–d): (1) we devised a marker of cumulative target cell recognition by measuring the amount of CD19 that CAR T cells have taken up from their target cells by trogocytosis. (2) We use CD69 expression as a marker of T cell activation to screen for T cells with elevated CAR signalling or reduced negative regulation. (3) We use FAS expression, which is markedly upregulated upon TCR stimulation, to identify knockouts that reduce the detrimental effect of apoptosis and fratricide in CAR T cell therapy. (4) Given elevated expression of T cell exhaustion markers in our co-culture system, we use a combined PD-1, LAG3 and TIM3 readout to screen for modulators of early exhaustion.

We optimized CELLFIE for FACS-based screens (Supplementary Note and Extended Data Fig. 5e–n) and performed 45 genome-wide screens with four FACS readouts, 12 sorted cell populations, two cell types (CD4[+] and CD8[+] CAR T cells) and cells derived from multiple blood donors (Extended Data Fig. 5k, Supplementary Figs. 2 and 3 and Supplementary Table 3). Data quality was consistently high, as confirmed by gRNAs targeting the FACS markers as built-in controls (Fig. 2b). We investigated four CAR T cell populations with potentially beneficial properties: CD19 high (increased target cell recognition), CD69 high (strong CAR T cell activation), FAS very low or absent (reduced apoptosis and fratricide) and PD-1, LAG3 and TIM3 triple negative (low early exhaustion indicators) (Fig. 2c), and we analysed these screens together with our fitness screens to prioritize all screening hits (Fig. 1d).

Using stringent thresholds, we identified 43 gene knockouts from our 58 genome-wide screens, 25 of which were shared across several screens (Fig. 2d and Extended Data Fig. 4g). We identified many known modulators of T cell and CAR T cell function and genes not previously implicated in CAR T cell function. Supporting the discovery power of the CELLFIE platform, our screening hits included the immune checkpoint regulators PD-1 (encoded by *PDCD1*), CTLA4, TIM3 (encoded by *HAVCR2*) and TIGIT, which are all clinically validated targets of immunotherapeutic drugs.

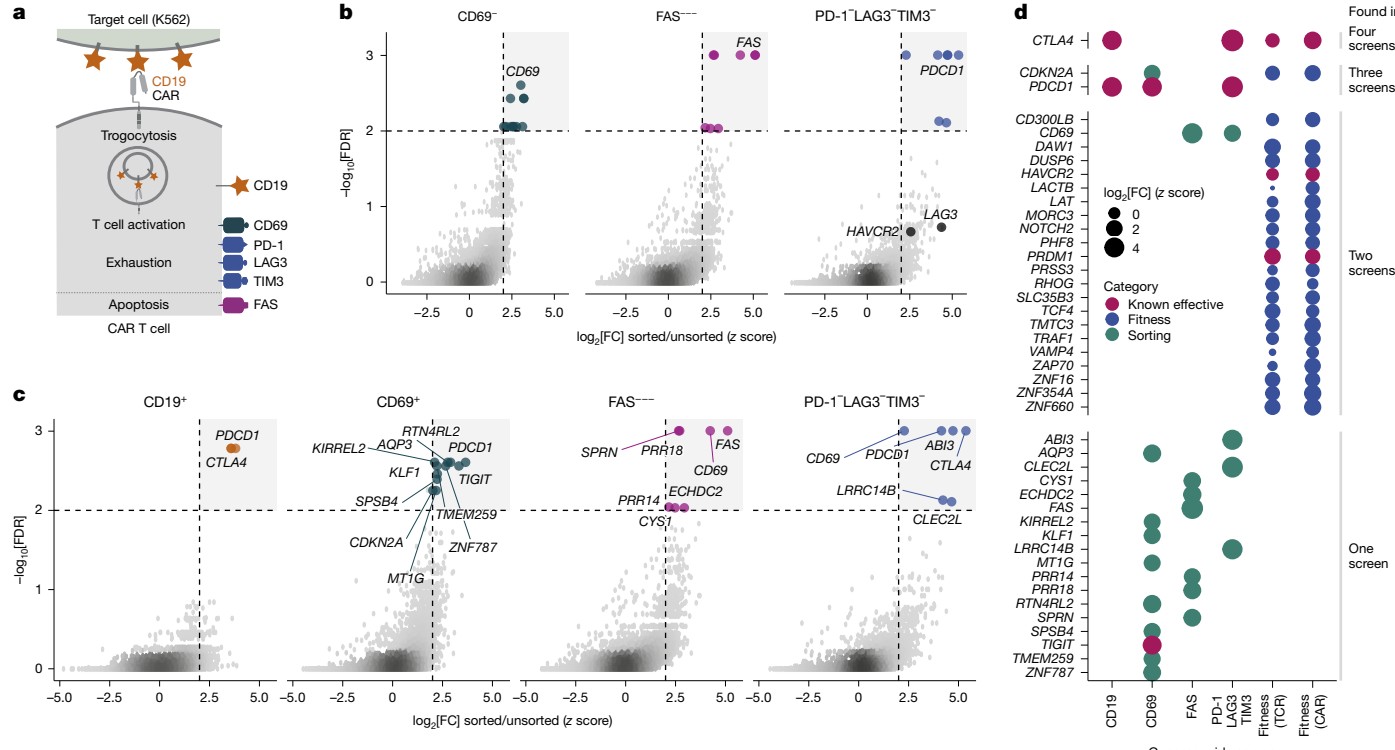

**Fig. 2 | Genome-wide FACS-based screens for clinical limitations of CAR T cells. a**, Overview of CAR T cell surface proteins used as screening readouts. Antigen recognition by the CAR can result in CD19 transfer from target cells to the CAR T cell surface (trogocytosis), serving as a marker of target cell recognition. CD69 expression is a marker of CAR T cell activation, FAS expression mediates apoptosis, and the combined expression of PD-1, LAG3 and TIM3 serves as an early indicator of T cell exhaustion. **b**, Validation analysis showing strong enrichment of FACS marker gene knockouts in marker-negative cells. Significantly enriched genes (FDR < 0.01 and $\log_2$[FC] > 1.5; thresholds indicated by dashed lines) are coloured, and FACS marker genes are labelled.

**c**, Identification of knockouts that improve CAR T cell function by FACS-based screening. Scatterplots show gene-level $\log_2$[FC] between sorted and unsorted cells (x axis) plotted against FDR-adjusted P values (y axis). Genes that passed stringent significance thresholds (FDR < 0.01 and $\log_2$[FC] > 1.5; indicated by dashed lines) are highlighted. **d**, Summary of screening hits (top hits across all screens, n = 43) that improved CAR T cell properties, identified in genome-wide fitness screens (blue) and FACS-based screens (green). The bubble plot shows z-normalized $\log_2$[FC] for significant gene knockouts. Validated immunotherapy target genes are highlighted in magenta.

## In vivo screen of CRISPR-boosted CAR T cells

To validate our screening hits, we extended CELLFIE to support pooled in vivo screens in mice with xenografts (Fig. 3a and Extended Data Fig. 6a), which are widely used for preclinical testing of CAR T cells. We injected 0.5 million human CD19[+] NALM6 cells into immunodeficient NOD−SCID−IL-2Rγ-null (NSG) mice, which induced systemic leukaemia that the mice succumbed to after 3 weeks when left untreated (Extended Data Fig. 6b,c). On day 5 of the leukaemia model, we injected 1 million CD19-targeting CAR T cells containing a pooled library of 39 perturbations. This CAR T cell dose was chosen to induce a suboptimal response with initial leukaemic cell clearance followed by swift relapse, leaving room to screen for knockouts that enhance the therapeutic response (Extended Data Fig. 6b,c). We collected spleen and bone marrow on day 9 and day 21 after CAR T cell injection, dissociated the organs and sequenced the gRNAs to determine which gene knockouts enhanced CAR T cell survival and proliferation (Fig. 3a).

To enable robust and scalable in vivo screens, we needed to overcome major technical hurdles that have thus far prevented the broad adoption of in vivo CRISPR screening in cancer immunology. First, the low frequency of CAR T cells in the collected organs led to very poor gRNA amplification efficiency from genomic DNA extracted from whole organs, with near-zero gRNA alignment rates (Fig. 3c). A nested primer design for genomic DNA yielded only marginal improvements. We thus developed a new in vivo screening method based on the CROP-seq technology[2], in which the gRNA is incorporated into a highly expressed mRNA transcript. In vivo CROP-seq replaces the conventional

DNA readout of gRNAs with an mRNA readout, profiting from the high copy number of gRNA-containing mRNAs (Fig. 3b and Supplementary Fig. 4a). We amplified gRNAs from reverse-transcribed mRNAs using whole-organ RNA extraction (without cell sorting) and routinely reached gRNA alignments of over 90% in pooled in vivo screens (Fig. 3c and Extended Data Fig. 6d). Second, in vivo screens are affected by strong cell population bottlenecks, clonal competition and random drift, which can obscure biological signals if not carefully controlled for. Therefore, we equipped the CROP-seq-CAR construct with unique molecular identifiers (UMIs) and developed a sequencing strategy to read the UMI along with the gRNA (Fig. 3b and Supplementary Fig. 4b).

We used our in vivo CROP-seq method to validate screening hits that we selected from the fitness screens (Fig. 1) and FACS-based screens (Fig. 2) using stringent thresholds (normalized MAGeCK MLE $\beta$ > 0.99 for the fitness screens and $\log_2$[fold change (FC)] > 1.5, false discovery rate (FDR) < 0.01 for the FACS-based screens), while also filtering for genes that are robustly expressed in CAR T cells (Extended Data Fig. 5n and Supplementary Tables 5 and 6). Each gene was targeted with eight gRNAs, and we included gRNAs targeting known essential genes, non-essential olfactory receptor genes and a safe harbour locus as controls. This in vivo screen comprised 39 target genes and 339 gRNAs with 20 mice (five mice per donor and time point), which reduced the necessary animal numbers by about 20-fold compared with individual knockout validation.

Supporting high screening quality, we observed strong depletion of essential genes (*POLR2L*, *PSMB4* and *RPL8*) across organs, donors and time points and neutral $\log_2$[FC] values with low noise for non-essential

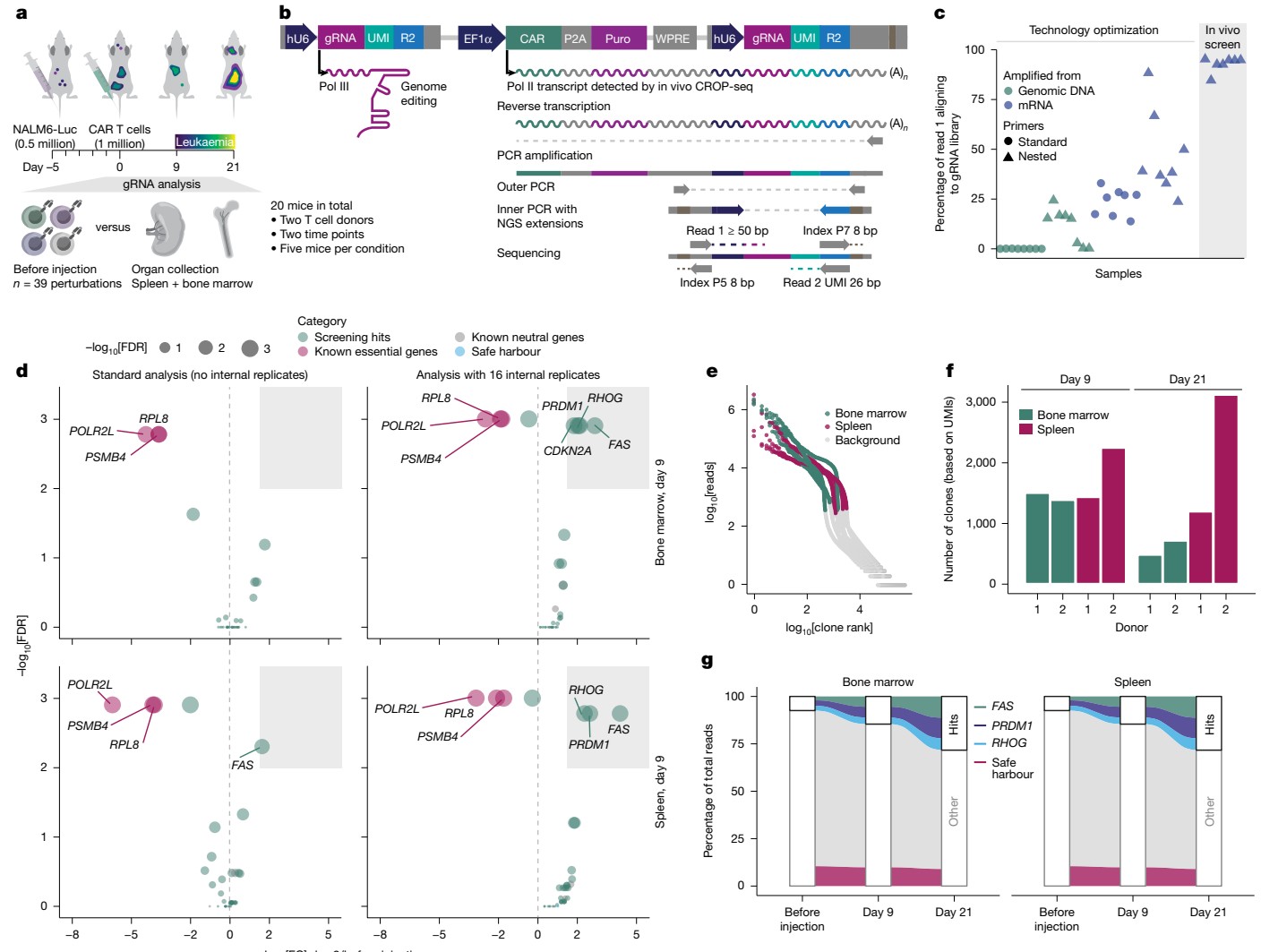

**Fig. 3 | In vivo CROP-seq screening of putative CAR T cell boosters in leukaemic mice. a**, Experimental timeline for pooled in vivo screening. Immunodeficient NSG mice are injected with human NALM6 cells to induce B cell leukaemia and then treated with a deliberately noncurative dose of CAR T cells with CRISPR knockout for the in vitro screening hits. Spleen and bone marrow are collected on day 9 and 21, and gRNA representation is compared with that of the pre-injection sample. Luc, firefly luciferase. **b**, Vector design and technical outline of the in vivo CROP-seq method. The gRNA cassette, including the UMI and the Illumina read 2 sequence, is located in the 3′ LTR of the CROP-seq-CAR vector. The gRNA and UMI are detected from the polymerase II transcript encoding CAR-P2A-Puro using reverse transcription and nested PCR. The gRNA is sequenced with read 1, and the UMI is sequenced with read 2 by paired-end sequencing. **c**, gRNA detection performance using conventional gRNA amplification from genomic DNA or gRNA amplification from

mRNA/cDNA (in vivo CROP-seq), both tested with single-PCR and nested PCR amplification. R2, read 2; WPRE, woodchuck hepatitis virus post-transcriptional regulatory element. **d**, In vivo CROP-seq data analysis with standard methods (left) or using UMIs to group reads into 16 internal replicates (right) for bone marrow (top) and spleen (bottom). *RHOG*, *FAS*, *PRDM1* and *CDKN2A* were identified as significant (FDR < 0.01 and $\log_2$[FC] > 1.5). The separation between depletion (left) and enrichment (right) is denoted by a dashed line at $\log_2$[FC] = 0. **e**, Knee plot showing rank versus read number for each UMI (clones versus background). The inflection point separates distinct CAR T cell clones (coloured) from background noise (grey). **f**, Number of distinct CAR T cell clones detected for each organ, donor and time point. **g**, CAR T cells for the three top-ranked gene knockouts (*FAS*, *PRDM1* and *RHOG*) dominate the CAR T cell response, collectively accounting for over 25% of gRNA reads on day 21 after injection.

genes (*OR4C6*, *OR10G4* and *OR14J1*) (Fig. 3d and Extended Data Fig. 6e). Using the UMIs for clonal analysis, we detected thousands of individual CAR T cell clones for each donor, indicating strong engraftment and adequate screening coverage (Fig. 3e,f and Extended Data Fig. 6j,k). We also used the UMIs to create internal replicates within each screen[37,38], which substantially improved sensitivity (Fig. 3d and Extended Data Fig. 6e–g). With stringent statistical thresholds ($\log_2$[FC] > 1.5 and FDR < 0.01), our in vivo CROP-seq screens consistently identified three gene knockouts as boosters of CAR T cell performance for spleen and bone marrow: *FAS*, *PRDM1* and *RHOG*, and a fourth gene knockout (*CDKN2A*) was specifically enriched in bone marrow (Fig. 3d, right). Notably, these genes would have been missed by standard analysis

without internal replicates (Fig. 3d, left), highlighting the importance of UMIs for successful in vivo screening.

Of these screening hits, FAS is a cell death receptor highly expressed in CAR T cells and linked to FASL-induced apoptosis. Its expression has been shown to limit CAR T cell efficacy, driven at least partially by FAS interaction with FASL on the CAR T cells themselves[39]. PRDM1 is an epigenetic regulator, and *PRDM1* knockout is known to enhance CAR T cell persistence, likely owing to altered T cell differentiation[14,21]. The third gene, which codes for the GTPase RHOG, is a new and unexpected hit emerging from our in vitro and in vivo screens. RHOG has previously been linked to a monogenic form of immunodeficiency[26], with its loss apparently detrimental to normal immune function. By contrast, we

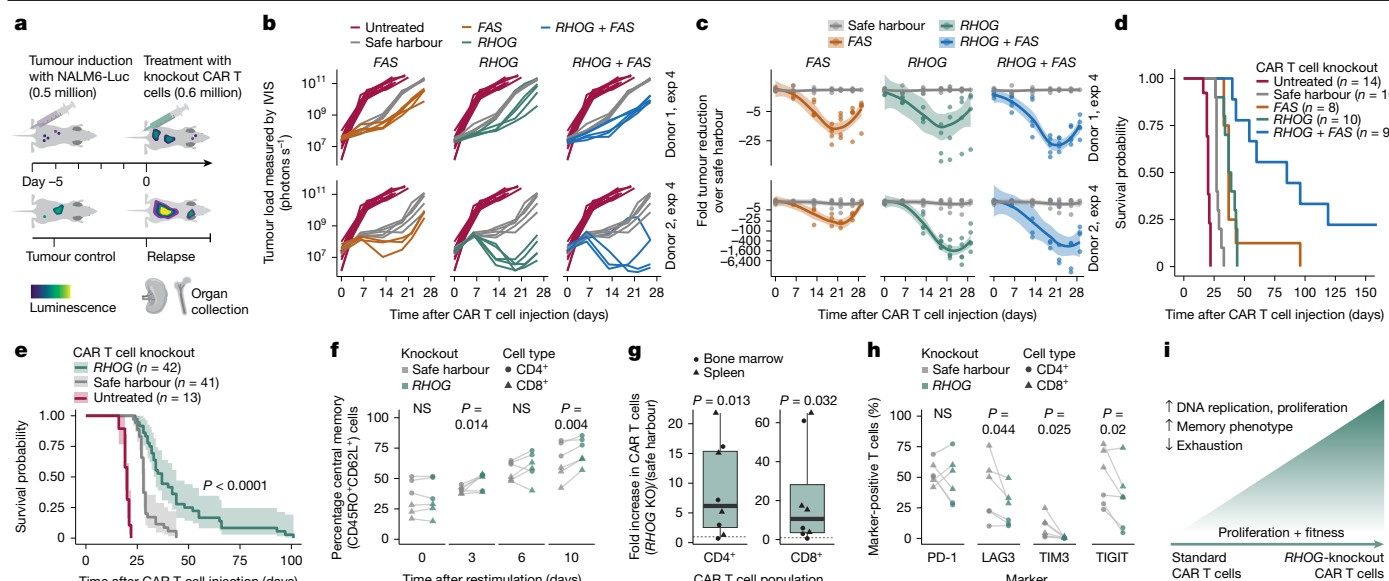

**Fig. 4 | *RHOG* and *FAS* knockout CAR T cells enhance leukaemic cell control and survival in vivo. a**, Experimental timeline of in vivo validation experiments (leukaemia model). Immunodeficient NSG mice are injected with human NALM6 cells to induce B cell leukaemia and treated with a deliberately noncurative dose of standard (safe harbour locus-edited) CAR T cells or with knockout CAR T cells for *RHOG*, *FAS* or both. **b**, Leukaemic cell load over time measured by bioluminescence imaging (two donors) comparing standard CAR T cells (ten mice) with the *RHOG* knockout (ten mice), *FAS* knockout (nine mice) and *RHOG*-and-*FAS* double-knockout (nine mice) CAR T cells. IVIS, in vivo imaging system; Exp, experiment. **c**, Leukaemia-reducing effect of knockout CAR T cells compared with standard CAR T cells for the mice shown in **b** (tumour reduction over time in vivo). Colored lines were obtained by local regression (LOESS); shaded areas represent 95% confidence intervals. **d**, Survival analysis for the mice shown in **b**. **e**, Survival analysis for all mice treated with 0.6 million *RHOG*-knockout (42 mice) or standard (41 mice) CAR T cells from all eight

donors; the *P* value is from the log-rank test comparing *RHOG*-knockout and standard CAR T cell treatment); shaded areas represent 95% confidence intervals. **f**, Percentage of central memory cells (CD45RO+CD62L+) among *RHOG*-knockout or standard CAR T cells after repeated in vitro CAR stimulation with K562-CD19 target cells (three donors). **g**, Fold increase in CD4+ or CD8+ CAR T cell numbers in mice treated with *RHOG*-knockout (KO) or standard CAR T cells, measured on day 15 after CAR T cell injection (four donors). The box plot's centre line is the median, the box limits are the upper and lower quartiles, and the whiskers extend to 1.5 times the interquartile range. **h**, Percentage of *RHOG*-knockout CAR T cells positive for the T cell exhaustion markers PD-1, LAG3, TIM3 and TIGIT on day 15 after CAR T cell injection (three donors). NS, not significant. **i**, Conceptual summary of the biological effects underlying the enhanced efficacy of *RHOG*-knockout CAR T cells. *P* values in **f**–**h** are from paired one-tailed *t*-tests.

observed a strong beneficial effect of *RHOG* knockout in CAR T cells. Knockout CAR T cells for these three genes rapidly expanded and jointly accounted for 25% of all CAR T cells after 21 days (Fig. 3g).

## *RHOG* and *FAS* knockouts enhance in vivo efficacy

To further validate and quantify the therapeutic benefit of the identified knockouts, we first produced single-knockout CAR T cells for *RHOG* and *PRDM1* and treated leukaemic mice with a reduced dose of 0.6 million CAR T cells (Extended Data Fig. 6a, top). Leukaemic cell load and therapeutic response were monitored by in vivo imaging of the luciferase-expressing NALM6 cells (Fig. 4a and Extended Data Fig. 7a–c). *PRDM1*-knockout CAR T cells achieved better initial leukaemic cell clearance than standard CAR T cells, as previously reported[14,21]. However, *PRDM1*-knockout CAR T cells could not significantly delay relapse and death of the treated mice (Extended Data Fig. 7a–c). By contrast, *RHOG*-knockout CAR T cells achieved striking leukaemic cell clearance and a sustained response (Extended Data Fig. 7a–c), which significantly extended survival compared with that of standard CAR T cells across all our in vivo validations (Fig. 4e).

We also compared *RHOG* knockout with *FAS* knockout and the *RHOG*-and-*FAS* double knockout (Fig. 4b–d). We chose the top-performing gRNA from our in vivo screen (Extended Data Fig. 6h,i and Supplementary Table 1) and delivered the gRNA in complex with the Cas9 protein by ribonucleoprotein (RNP) electroporation (Extended Data Fig. 6a, bottom), whereas the anti-CD19 CAR was separately delivered with a lentiviral vector[12]. *RHOG* knockout and *FAS* knockout CAR T cells clearly outperformed standard CAR T cells (Fig. 4b–d).

The beneficial effect of *FAS* knockout in CAR T cells aligns with prior studies[39–42] and is currently being evaluated in a clinical trial for ovarian cancer (NCT05617755). *RHOG* knockout resulted in leukaemia clearance and survival comparable or potentially superior to *FAS* knockout (Fig. 4c). Moreover, we discovered a strong combined effect when knocking out both *RHOG* and *FAS*, which greatly extended survival compared with both single knockouts (Fig. 4b–d). We validated this effect in patient-derived CAR T cells (Extended Data Fig. 7d), and we found that the double-knockout CAR T cells were curative for two of the treated mice, whereas this leukaemia model was rapidly and universally fatal in mice treated with the same dose of standard CAR T cells, with relapses that did not downregulate the CD19 target antigen (Extended Data Fig. 7e). At a higher dose, *RHOG*-knockout CAR T cells were also curative in this model (Extended Data Fig. 7f). Long-term follow-up (up to 392 days) showed no signs of malignant transformation of the CRISPR-boosted CAR T cells (Extended Data Fig. 7g).

Finally, we evaluated the effect of *RHOG* knockout across multiple CAR designs. We found that it consistently boosted T cell and CAR T cell expansion in vitro (Extended Data Fig. 7h) and improved leukaemia clearance in vivo for 19-BBz, 19-28z and GD2–BBz CARs (Extended Data Fig. 7i,j), confirming broad efficacy across antigens and signalling domains. The *RHOG–FAS* combination also enhanced tumour clearance in a solid tumour model of subcutaneously injected Huh7 cancer cells, which were treated with GPC3–BBz CAR T cells (Extended Data Fig. 7k).

In summary, all three top hits (*RHOG*, *FAS* and *PRDM1*) of our in vivo screens were validated with highly consistent results, underlining the reliability of in vivo CROP-seq (Fig. 4 and Extended Data Figs. 7

and 8a–d), and we established knockout of *RHOG* and double knockout of *RHOG* and *FAS* as potent strategies to enhance CAR T cell performance.

## RHOG affects proliferation and persistence

The strong in vivo performance of *RHOG*-knockout CAR T cells prompted us to investigate the biological role of RHOG in CAR T cells, which is entirely unexplored. In T cells, RHOG supports immune synapse formation, cytoskeletal remodelling, TCR signalling, proliferation and anergy[43,44]. RHOG deficiency causes inborn errors of immunity with impaired T cell cytotoxicity[26]. These biological functions make *RHOG* an unexpected target for engineering more potent CAR T cells. At the same time, it represents a striking example of our foundational hypothesis that genes essential for normal T cell biology can actively restrict therapeutic performance in CAR T cell therapy.

We characterized *RHOG*-knockout CAR T cells over 10 days of repeated exposure to CD19+ cancer cells, comparing them to standard CAR T cells edited at a safe harbour locus (Fig. 4f and Extended Data Fig. 8e–m). CD4+ and CD8+ CAR T cells were profiled by flow cytometry and RNA-seq at time points corresponding to acute stimulation (days 1–3) and chronic stimulation (days 6–10). We first investigated their cell differentiation status, which is a key predictor of expansion and clinical performance[45]. Flow cytometry revealed a higher fraction of CD62L+CD45RO+ cells in *RHOG*-knockout CAR T cells (Fig. 4f and Extended Data Fig. 8e), suggesting a shift toward a central memory phenotype, which may contribute to their observed proliferation advantage.

RNA-seq revealed widespread transcriptional differences between *RHOG*-knockout and standard CAR T cells (Extended Data Fig. 8h–j and Supplementary Table 6). Gene set enrichment analysis identified an upregulation of genes involved in DNA replication and the cell cycle, ribosome metabolism and protein translation (Extended Data Fig. 8k,l). These changes were accompanied by elevated expression of S phase marker genes at later time points (Extended Data Fig. 8m), consistent with sustained proliferation during chronic stimulation. These effects were observed in both CD4+ and CD8+ cells, indicating that *RHOG* knockout might be broadly beneficial across CAR T cell subsets.

We further evaluated the effects of *RHOG* knockout on CAR T cells in the NALM6 xenograft leukaemia model. On day 15 after CAR T cell injection, we isolated spleen and bone marrow and profiled CD4+ and CD8+ CAR T cells by flow cytometry. *RHOG* knockout led to markedly increased CAR T cell abundance in both organs (sixfold for CD4+ cells and tenfold for CD8+ cells, Fig. 4g) and reduced expression of the exhaustion markers LAG3, TIM3 and TIGIT (Fig. 4h). These findings indicate that *RHOG* knockout enhances persistence and reduces exhaustion in vivo.

Effective CAR T cell therapies must preserve T cell activation and effector functions, and *RHOG*-knockout CAR T cells indeed maintained these properties, with flow cytometry showing no significant changes in CD69, CD107a, IL-2 or interferon-γ after in vitro stimulation with CD19+ K562 cells (Extended Data Fig. 8f). The cells' cytotoxic capacity was also preserved, as shown in luciferase-based in vitro killing assays (Extended Data Fig. 8b). Finally, only *FAS* knockout but not *RHOG* knockout led to decreased apoptosis levels (Extended Data Fig. 8g).

In summary, our results demonstrate that *RHOG* knockout enhances CAR T cell proliferation, promotes a central memory phenotype, reduces exhaustion and maintains effector function (Fig. 4i). These characteristics are distinct from and complementary to the anti-apoptotic and fratricide-inhibiting effects of *FAS* knockout, which helps explain the strong in vivo performance of *RHOG*-and-*FAS* double-knockout CAR T cells.

## Combinatorial screening for synergistic genes

The high in vivo efficacy of *RHOG*-and-*FAS* double-knockout CAR T cells indicates that targeting complementary aspects of CAR T cell biology

can yield strong combined effects. To explore such pairwise combinations more systematically, we integrated combinatorial screening into the CELLFIE platform by adapting the CROPseq-multi technology[46] and developing the CROP-seq-CAR-multi vector for delivery of two gRNAs from a single transcript, separated by transfer RNA (tRNA) sequences that are cleaved by the endogenous tRNA-processing system (Fig. 5a,b and Supplementary Table 7).

For our combinatorial screening in CAR T cells, we selected six genes our in vivo screens (*RHOG*, *FAS*, *PRDM1*, *CDKN2A*, *HAVCR2* and *CTLA4*), along with three essential genes and safe harbour locus controls, and designed a dual-gRNA library covering all 238 pairwise combinations of two gRNAs per gene (Fig. 5a). We cloned this library into the CROP-seq-CAR-multi vector and screened it across three CARs targeting two different antigens (CD19 and GD2) and using two different co-stimulatory domains (4-1BB and CD28).

We confirmed efficient delivery and detection of the gRNA pairs, with low recombination rates and strong depletion of essential genes, which was more pronounced for the second gRNA position in the vector (Extended Data Fig. 9a–c). This combinatorial screen validated the strong performance of the double-knockout CAR T cells across all three CAR designs (Fig. 5c and Extended Data Fig. 9d). We also identified additional gene-knockout pairs with the potential to further enhance CAR T cell function (Fig. 5c and Extended Data Fig. 9e,f). Together, these results establish combinatorial screening as a robust and flexible extension of the CELLFIE platform, which enables the systematic discovery of combined and synergistic genetic perturbations to improve CAR T cell function.

## Base editing for clinical translation

Although CRISPR nucleases are effective discovery tools, their reliance on double-strand breaks poses potential safety concerns for clinical translation. CRISPR base editing offers an alternative approach with enzymatic modification of DNA instead of double-strand breaks[47]. Moreover, tiling base-editing screens enable high-resolution functional mapping of protein function[24,48–51]. Leveraging CELLFIE's support for diverse base editors, we performed saturation mutagenesis of *RHOG* to identify critical amino acids and to prioritize base-editing gRNAs for clinical use.

We prepared a screening library of 3,755 base-editing gRNAs (Fig. 5d and Supplementary Table 8), which provides a dense tiling of the gene bodies of *RHOG* (1,169 gRNAs) and of the puromycin resistance gene *PAC* (1,140 gRNAs) and includes gRNAs targeting essential genes (positive controls) and a safe harbour locus (negative controls). We cloned this gRNA library into the CROP-seq-CAR vector, transduced CD4+ T cells from two blood donors and electroporated mRNAs for four different base editors: the A-to-G editor ABEmax[28], the C-to-T editor AncBE-4max[28] and their near-PAM-less variants[29], which together enabled dense mutagenesis of both target genes (Fig. 5e). As an additional control, we also included standard *Cas9* mRNA, which generates frameshift mutations across the gene body.

After 12 days of repeated CAR stimulation with CD19-expressing cancer cells, CAR T cells that received standard *Cas9* mRNA showed the expected enrichment of *RHOG*-targeting gRNAs and depletion of gRNAs targeting essential genes and *PAC*, which was true for all four base editors and validates our setup (Extended Data Fig. 10ba–e).

By plotting the enrichment or depletion of base-editing gRNAs along the protein sequence, we can map functionally important amino acids. We tested this approach for *PAC* and found several highly depleted amino acids colocated around the puromycin-binding site (Extended Data Fig. 10f,g). For *RHOG*, we observed enrichment of multiple gRNAs, including those predicted to introduce missense mutations into the GTP-binding pocket (for example, Lys16 and Cys18, which form hydrogen bonds with GTP) (Fig. 5f,g). These results indicate that the catalytic function of RHOG is important for its detrimental effect in CAR T cells,

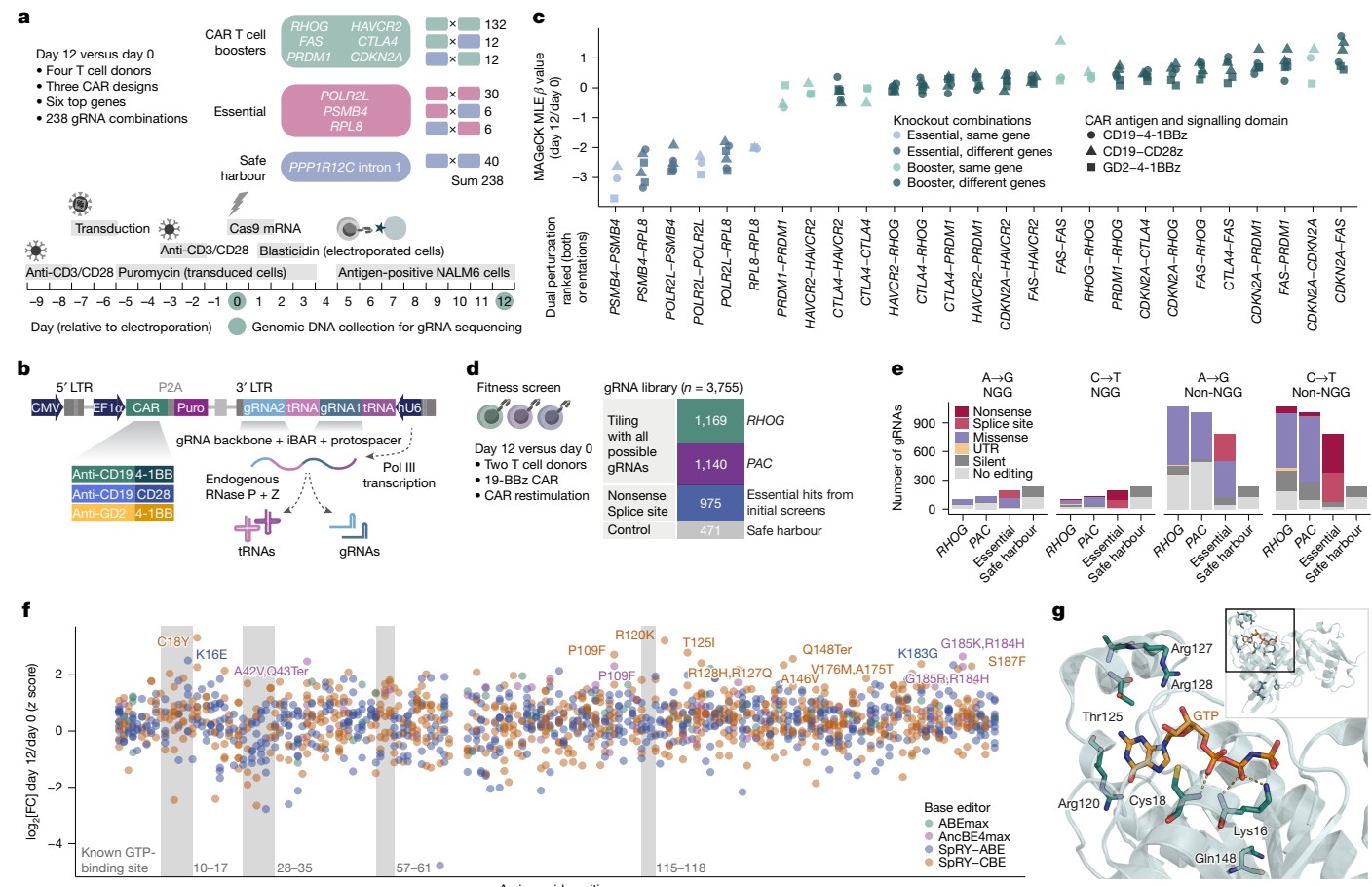

**Fig. 5 | Combinatorial knockout and base-editing screening with the CELLFIE platform. a**, Experimental timeline of combinatorial fitness screens (four donors) using CROP-seq-CAR-multi vectors for three different CARs (19-BBz, 19-28z, GD2–BBz). The screening library comprises 238 combinations of gRNAs targeting six top hits from the in vivo pooled screening and control gRNAs targeting essential genes and a safe harbour locus. **b**, Vector design and technical outline of combinatorial screening with the CROP-seq-CAR-multi construct with three different CARs. The vector adapts the CROPseq-multi technology[46] with a dual-gRNA cassette separated by a tRNA sequence that is cleaved by cellular enzymes. CMV, cytomegalovirus promoter. **c**, Ranking of fitness effects (MAGeCK MLE $\beta$ value comparing day 12 versus day 0 gene interactions) for pairwise gene knockouts. **d**, Overview of saturation base-editing screening. The library includes all possible gRNAs tiling the gene bodies of *RHOG* and of the puromycin resistance gene *PAC* (for validation,

encoding puromycin *N*-acetyltransferase) and gRNAs targeting essential genes (positive controls) and a safe harbour locus (negative controls). **e**, Number and type of base-editing mutations introduced by the gRNA library with each of the four CRISPR base editors (A to G, C to T, and their near-PAM-less variants), based on computational predictions ($n = 3,755$). UTR, untranslated region. **f**, Mutagenesis map for RHOG based on base-editing tiling screens (two donors), comparing gRNA distribution (introducing missense or nonsense mutations) between day 12 and day 0 after CAR restimulation. The plot shows z-normalized $\log_2[\text{FC}]$ (*y* axis). The top 15 most enriched gRNAs are labelled with their predicted amino acid changes. **g**, Amino acids with the strongest mutagenesis effects in the screen, mapped onto the RHOG protein structure (PDB 6UKA). The visualization is a magnified view of the GTP-binding site, with the full protein shown in the inset.

and they suggest that pharmacological RHOG inhibition could provide an alternative way to enhance CAR T cell function.

Finally, we validated a subset of the base-editing gRNAs in a follow-up base-editing screen (Extended Data Fig. 10h–i). Targeted sequencing of the *RHOG* gene identified missense mutations conferring a strong proliferative advantage to CAR T cells (Extended Data Fig. 10h,i), which we validated individually (Extended Data Fig. 10j,k). These experiments establish base-editing screens as a powerful extension of the CELLFIE platform for functional characterization of screening hits and data-driven selection of base-editing gRNAs for clinical translation.

## Discussion

CAR T cells are revolutionary therapeutic agents, but optimizing them for their many potential uses is complex and labour intensive. In this study, we developed the CELLFIE platform for systematic engineering of CAR T cells, integrating genome-wide discovery, in vivo pooled

validation, combinatorial knockout screening and tiling base editing to identify perturbations that enhance their therapeutic efficacy.

The core of this technology is the CROP-seq-CAR vector for co-delivery of the CAR and gRNA library, with built-in support for pooled, single-cell[2,52] and optical screens[53]. We also optimized mRNA production and delivery of various CRISPR editors for flexible genome engineering. To facilitate adoption of the CELLFIE platform, we provide open-source reagents and detailed protocols that will make such screens realistic for many laboratories. Although this study focused on CAR T cells, the platform is readily transferable to other cell types and cell-based therapies.

To enable large-scale validation of our genome-wide screening hits, we developed the in vivo CROP-seq method for pooled screening of CRISPR-perturbed CAR T cells in mice with UMI-based clonal tracking. Using this method, we tested 39 perturbations in 20 mice, constituting an approximate 20-fold reduction in animal numbers compared with one-by-one validations. Although used here for validation, in vivo

CROP-seq also enables large discovery screens in mice. To guide future study design, we extrapolated from our in vivo data to estimate the number of perturbations achievable in different screening configurations (Extended Data Fig. 6k). For example, with four gRNAs per gene and a dual-gRNA vector, up to 156 genes can be tested in five animals. Scaling to 50 animals while targeting two lentiviral integrations per cell could support pooled testing of over 3,000 target genes.

Using the CELLFIE platform, we conducted 58 genome-wide screens in human primary CAR T cells, targeting multiple biological processes implicated in therapy failure: CAR T cell fitness and proliferation upon CAR or TCR stimulation, target cell recognition, activation, apoptosis and fratricide, and expression of inhibitory receptors. Our study constitutes the largest resource of functional genomic screens in CAR T cells to date, which enabled the discovery of multiple gene knockouts that improve therapeutic performance in mouse models of human cancer.

*RHOG* knockout is a particularly fascinating discovery that underlines the value of genome-wide screening. Previously linked to a monogenic immunodeficiency in humans, *RHOG* was an unlikely candidate but a strong hit in our in vitro and in vivo screens. *RHOG*-knockout CAR T cells showed enhanced proliferation, preferential memory cell differentiation and reduced exhaustion. Another screening hit, *FAS* knockout, provided complementary benefits by reducing apoptosis and enhancing survival. The combination of *RHOG* and *FAS* knockouts yielded highly synergistic effects, significantly improving tumour control and long-term remissions in an intentionally noncurative xenograft model. We validated our findings across four CAR designs, two tumour models and multiple T cell donors, including a patient scheduled to receive CAR T cell therapy.

The strong effect of combining *RHOG* and *FAS* knockout exemplifies a modular approach to CAR T cell engineering, in which distinct perturbations can be combined into tailored therapeutic designs. To enable systematic discovery of such knockout pairs, we integrated combinatorial CRISPR screening into the CELLFIE platform and conducted a proof-of-concept screen that validated our finding that combinations of gene knockouts can enhance CAR T cell performance well beyond the effect of any single knockout. Finally, we used base editing to conduct tiling screens across the *RHOG* sequence to map those parts of the gene that are critical for the inhibitory effect of *RHOG* on CAR T cells and to support the selection of base-editing gRNAs for future clinical development.

In conclusion, this work establishes a blueprint for systematic, screening-based optimization of cell therapies, which may help realize the vast potential of cells as flexible, context-reactive and genetically programmable therapeutics.

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

# Methods

## Cloning the CROP-seq-CAR screening vectors

The design of the CROP-seq-CAR vectors builds on the CROP-seq technology[2], which enables CRISPR genome editing with polymerase III-transcribed gRNAs and incorporation of the gRNAs into a separate polymerase II transcript for versatile detection in single cells. We designed a CROP-seq-CAR lentiviral transfer plasmid that uses the EF1α promoter to drive expression of the CAR, followed by a P2A self-cleaving peptide and a puromycin resistance marker. In CROP-seq, the gRNA is placed inside the lentiviral 3' LTR, just before the polyadenylation signal of the vector. There, it becomes part of the highly expressed CAR-P2A-Puro transcript and is readily detected by bulk and single-cell RNA-seq assays. The gRNA cassette is also duplicated into the 5' LTR during lentiviral integration and transcribed from a polymerase III promoter for effective genome editing. This way, CROP-seq-CAR vectors simultaneously achieve high genome-editing performance and an efficient sequencing-based readout of the gRNA.

The CROP-seq-CAR vectors were cloned from the CROPseq-Guide-Puro plasmid (Addgene, 86708), which was digested with XmaI and SalI. The resulting 7,737-bp fragment was extracted from a gel using the S.N.A.P. UV-Free Gel Purification Kit (Thermo Fisher Scientific, K200025) and dephosphorylated with Shrimp Alkaline Phosphatase (rSAP, NEB, M0371S) for 1 h at 37 °C, followed by heat inactivation for 20 min at 80 °C. We ordered two gBlocks (IDT) encoding EF1α-CAR (gBlock-17) and P2A-mCherry-WPRE (gBlock-18), with the CAR consisting of a leader peptide, the single chain variable fragment (scFv) of an antibody binding CD19 (clone FMC63), the transmembrane domain of CD8, the intracellular co-stimulatory domain of 4-1BB and the CD3ζ chain. The gBlocks were amplified by PCR using Q5 High-Fidelity 2X Master Mix (NEB, M0492L) with primers gBlock-17-FWD and gBlock-17-REV for gBlock-17 or with gBlock-18-FWD and gBlock-18-REV for gBlock-18. These primers extend the gBlocks with BsmBI type IIS restriction sites and overhangs for Golden Gate cloning. PCR-amplified and gel-purified gBlocks were digested with BsmBI-v2 (NEB, R0739S), column purified and ligated with the Quick Ligation Kit (NEB, M2200S). The ligation product was amplified by PCR with primers gBlock-17-FWD and gBlock-18-REV, column purified, digested one more time with BsmBI-v2 and inserted into the CROP-seq backbone by Gibson assembly using NEBuilder HiFi DNA Assembly Master Mix (NEB, E2621S). Sequences of the gBlocks and amplification primers are provided in Supplementary Table 1i. This cloning yielded the pPD0010_CROP-seq-CAR-mCherry2 vector (Addgene, 242533), from which all CROP-seq-CAR variants were derived.

For efficient selection of transduced cells, mCherry was replaced with the *PAC* antibiotic resistance gene, resulting in the pPD0039_CROP-seq-CAR-Puro vector (Addgene, 242532) used in most screens. To generate variants with different CARs, this vector was linearized by PCR using Q5 High-Fidelity 2X Master Mix (NEB, M0492L) and primers prCART0026 and prCART0028, which exclude the sequence for the CD19 scFv. Sequences for the scFvs for GD2 (14g2a scFv) and GPC3 were synthesized as gBlocks (IDT), amplified by PCR with primers adding overhangs that are homologous to the ends of the linearized backbone (sequences are provided in Supplementary Table 1i) using Q5 High-Fidelity 2X Master Mix (NEB, M0492L) and cloned by Gibson assembly using NEBuilder HiFi DNA Assembly Master Mix (NEB, E2621S). This resulted in the pCART0017_CROP-seq-CAR-Puro_GD2-BBz (Addgene, 242985) and pCART0016_CROP-seq-CAR-Puro_GPC3-BBz (Addgene, 242984) vectors. The pPD0035_CROP-seq-CAR-Puro_CD19-28z (Addgene, 242983) vector was generated by replacing the sequence for the intracellular 4-1BB co-stimulatory domain with the sequence for the CD28 co-stimulatory domain ordered as a gBlock (IDT).

## Cloning the CROP-seq-CAR-multi screening vectors

The CROP-seq-CAR-multi vectors were cloned from the CROPseq-multi-v2-mTagBFP2-NLS plasmid (Addgene, 225754), which was amplified with primers CROPseq_FWD_016-minusACC and CROPseq-multi-v2_REV1 (Supplementary Table 1i) using Q5 High-Fidelity 2X Master Mix (NEB, M0492L) to generate the backbone for all variants of the CROP-seq-CAR-multi vectors (7,622 bp, including 40-bp extensions), followed by DpnI digestion (NEB, R0176L) to remove the parental plasmid. Sequences for the CARs were amplified from the corresponding CROP-seq-CAR vectors with primers PD39-FWD1 and CROPseq_REV_019 (Supplementary Table 1i), followed by DpnI digestion. PCR products were digested with PaqCI (NEB, R0745S), heat inactivated and cleaned with the Monarch PCR & DNA Cleanup Kit (NEB, T1030). The purified PCR fragments were ligated using T4 DNA Ligase (NEB, M0202T) at a 3:1 ratio of insert:backbone, resulting in the CROP-seq-CAR-multi vectors pCROP-seq-CAR-19-BBz-multi, pCROP-seq-CAR-19-28z-multi and pCROP-seq-CAR-GD2-BBz-multi.

## Cloning the mRNA expression vectors

Open reading frames (ORFs) for mRNA production were cloned into the pIVTRup vector (Addgene, 101362) by Gibson assembly. The pIVTRup backbone was linearized by PCR with the primers pIVTRup_linearize_FWD and pIVTRup_linearize_REV (Supplementary Table 1i). ORFs for CRISPR modifiers and selection markers were amplified from existing plasmids (Supplementary Table 1b) with ORF-specific primers that included extensions for Gibson assembly: mRNA_ORF_Gibson_FWD and mRNA_ORF_Gibson_REV (Supplementary Table 1i). Both backbone and insert were amplified using Q5 Hot Start High-Fidelity 2X Master Mix (NEB, M0494L). Each 50-μl PCR reaction was mixed with 6 μl Cut-Smart Buffer, 3 μl nuclease-free water and 1 μl DpnI (NEB, R0176S) and incubated for 1 h at 37 °C to digest the plasmid template, followed by heat inactivation for 20 min at 80 °C. PCR products were cleaned with the QIAquick PCR Purification Kit (Qiagen, 28106). The final product was assembled with NEBuilder HiFi DNA Assembly Master Mix (NEB, E2621L) using a 2:1 molar ratio of insert:backbone. After incubating the mix for 15 min at 50 °C, the samples were stored at −20 °C or directly used for bacterial transformation. All plasmids were validated by Sanger sequencing.

## Cloning of gRNAs and gRNA libraries

Individual gRNAs and small-scale gRNA libraries were cloned into CROP-seq-CAR vectors as described previously[2]. To clone the genome-wide Brunello gRNA library into the CROP-seq-CAR vector, we amplified the gRNA insert from Addgene plasmid 73178 by PCR[31]. We mixed Q5 Hot Start High-Fidelity 2X Master Mix (NEB, M0494L), SYBR Green, 0.5 μM of the primers gRNA_library_FWD and gRNA_library_REV and 1 μg of plasmid DNA per 50-μl reaction. The amplification was stopped at approximately 3,000 relative fluorescence units (RFU) in the exponential phase to avoid library overamplification. We treated the PCR product with DpnI (NEB, R0176L) to remove plasmid DNA and concentrated the sample by ethanol precipitation to achieve high purity. The Brunello gRNAs were assembled into the BsmBI-linearized CROP-seq-CAR vector using NEBuilder HiFi DNA Assembly Master Mix (NEB, E2621L) as described previously[2].

## Cloning of the gRNA library for in vivo CROP-seq

To prepare the gRNA library for in vivo CROP-seq (Supplementary Fig. 4a), a backbone fragment was amplified by PCR from the pPD0039_CROP-seq-CAR-Puro (Addgene, 242532) plasmid using Q5 High-Fidelity 2X Master Mix (NEB, M0492L) and 0.5 μM of the primers CART52_Bkbn_FWD and CART52_Bkbn_REV (Supplementary Table 1f). For quality control, we ran 1 μl of the PCR reaction on a 0.8% agarose gel containing SYBR Safe stain and verified the product size of 9,783 bp. Template plasmid was removed by DpnI digestion (NEB, R0176L) for 1 h at 37 °C, followed by inactivation of the enzyme for 20 min at 80 °C, cleaning with the QIAquick PCR Purification Kit (Qiagen, 28104) and quantification with the Qubit dsDNA HS assay (Thermo Fisher Scientific, Q32854). The gRNA library (libCART011, Supplementary Table 5b) was obtained

as an oPool (IDT). The construct's constant region (CART52_const_ultra, Supplementary Table 1f) was ordered as an ultramer oligonucleotide (IDT). The gRNA library oPool and the constant ultramer oligonucleotide were diluted to 2 µM and annealed in T4 DNA Ligase Buffer (NEB, B0202S). The library insert cassette was extended and amplified using Q5 High-Fidelity 2X Master Mix (NEB, M0492L) with 0.5 µM of the primers CART52_cassette_FWD and CART52_cassette_REV (Supplementary Table 1f). The product size of 225 bp was confirmed on a 2% agarose gel. The backbone and library insert were assembled by Gibson assembly using NEBuilder HiFi DNA Assembly Master Mix (NEB, E2621L). Gibson assembly reactions were desalted by filter dialysis (Merck, VMWP04700) and electroporated into Lucigen Endura *Escherichia coli* cells (Lucigen, 60242-2). The library plasmid was prepared using the Plasmid Plus Giga Kit (Qiagen, 12991).

## Cloning of the gRNA library for combinatorial screening

For combinatorial screening, we selected the six strongest CAR T cell boosters based on the pooled in vivo screen (*FAS*, *RHOG*, *PRDM1*, *CDKN2A*, *HAVCR2* and *CTLA4*) and three essential genes (*RPL8*, *PSMB4* and *POLR2L*). For each gene, we chose the two best-performing gRNAs (of a total of eight). As negative controls, we selected the 15 most neutral *PPP1R12C* safe-harbour-targeting gRNAs, from which we randomly picked gRNA pairs (Fig. 5a and Supplementary Table 5b). The gRNAs that target boosters and essential genes were paired separately, yielding combinations targeting the same genes, different genes within each set or one target gene and the safe harbour locus. This resulted in 238 dual-gRNA combinations (Fig. 5a and Supplementary Table 7a), for which we designed a CROP-seq-multi library based on published source code (https://github.com/rtwalton/CROPseq-multi)[46].

CROP-seq-CAR-multi library inserts (238 oligonucleotides, Supplementary Table 7a) were obtained as an oligonucleotide pool (Twist Bioscience) and amplified by PCR using 2× KAPA HiFi HotStart ReadyMix (Roche, KK2601) with 1 M betaine (MilliporeSigma, B0300), 300 nM primers (oRW1083 and oRW1086) and 12 pg µl$^{-1}$ template per 25-µl PCR reaction. Thirty-three PCR reactions were set up with the following conditions: 95 °C for 3 min, 13 cycles of (98 °C for 20 s, 62 °C for 15 s, 72 °C for 15 s), 72 °C for 1 min. Here, the low input and few amplification cycles help reduce recombination during the PCR[46]. The pooled PCR products were purified using the Monarch PCR & DNA Cleanup Kit (NEB, T1030), digested with BsmBI-v2 (NEB, R0739L) and purified. The CROP-seq-CAR-multi vectors (pCROP-seq-CAR-19-BBz-multi, pCROP-seq-CAR-19-28z-multi and pCROP-seq-CAR-GD2-BBz-multi) were digested with BsmBI-v2, dephosphorylated using rSAP (NEB, 0371L) and purified. In a total volume of 20 µl, 20 fmol of digested oligonucleotide pool and 20 fmol of digested plasmid backbone were ligated using 400 U µl$^{-1}$ of T4 DNA ligase (NEB, M0202S) at 16 °C overnight, followed by heat inactivation and cleanup with solid-phase reversible immobilization (SPRI) beads (Beckman Coulter, B23317) at a 1.5× ratio. The purified ligation product was electroporated into Lucigen Endura *E. coli* cells (Lucigen, 60242-2) with a Gene Pulser Xcell system (Bio-Rad, 1652662) (settings: 1.8 kV, 600 Ω and 10 µF), which were allowed to recover for 90 min at 30 °C in 1 ml of Endura recovery medium, followed by growth in a 100-ml liquid culture in selection LB medium (100 µg ml$^{-1}$ carbenicillin) at 30 °C, with shaking at 225 rpm. Library plasmids were prepared using the Plasmid Plus Midi Kit (Qiagen, 12943).

## Production of lentivirus

Lentivirus was produced using Lipofectamine 3000 (Invitrogen, L3000150) as described previously[2], followed by 100× concentration using the Lenti-X Concentrator (Takara, 631232). For large-scale lentivirus production, all reagent and cell culture volumes were scaled accordingly for Nunc EasyFill Cell Factory Systems (Thermo Fisher Scientific). Lentivirus titre was quantified by quantitative PCR (qPCR) using the Lenti-X qRT–PCR Titration Kit (Takara Bio, 631235). For each lentivirus preparation, two aliquots of concentrated virus were diluted 100× with 1× PBS and used as input for quantification, and the mean titre (RNA copies per ml) was calculated.

## Production of mRNA

CRISPR editors were cloned into our mRNA production vector as described above. This vector contains 5′ and 3′ untranslated regions to stabilize the mRNA, and the mRNAs are produced by in vitro transcription from a T7 promoter with cotranscriptional capping and terminated by a hardcoded A-tail of 55 bases. Optionally, modified bases such as 5-methoxy-U can be incorporated to optimize mRNA properties[54]. However, this increases the cost of mRNA production and was not necessary for efficient genome engineering in our screens.

To prepare for mRNA production, the corresponding DNA template was amplified by PCR from 10 ng of pIVTRup with the cloned ORF using Q5 Hot Start High-Fidelity 2X Master Mix (NEB, M0494L) and 0.5 µM of the primers pIVTRup_FWD_AG and pIVTRup_REV (Supplementary Table 1i) in a total volume of 50 µl and using 26 amplification cycles. The PCR product was cleaned using the QIAquick PCR Purification Kit (Qiagen, 28106). Synthetic mRNA was produced by in vitro transcription with the HiScribe T7 High Yield RNA Synthesis Kit (NEB, 2040S). Reactions contained 500 ng of PCR product, 10 mM of each nucleotide and 8 mM of CleanCap Reagent AG (TriLink BioTechnologies, N-7113-10) and were incubated for 2 h at 37 °C, followed by DNase I treatment (Promega, M6101) for 15 min at 37 °C. For large-scale production, the number of reactions was scaled accordingly. Immediately after the incubation, mRNA was cleaned with the RNeasy Mini Kit (Qiagen, 74104), loading 8–16 reactions per column. mRNA concentration was measured with a Qubit RNA BR Assay Kit (Invitrogen, Q33223), and mRNA integrity was assessed with the Bioanalyzer RNA 6000 Pico Kit (Agilent, 5067-1513).

## T cell isolation

Peripheral blood samples from healthy donors were obtained through the Austrian Red Cross as blood packs with buffered sodium citrate as anticoagulant. Human primary T cells were isolated using RosetteSep Human T Cell, CD4$^+$ T Cell or CD8$^+$ T Cell Enrichment Cocktails (Stemcell, 15061, 15062 and 15063) to isolate pan-CD3$^+$, CD4$^+$ or CD8$^+$ T cells, respectively. After incubation with the antibody cocktail, cells were centrifuged using Lymphoprep (Stemcell, 7861) in SepMate-50 tubes (Stemcell, 85460), washed and treated with 1× RBC Lysis Buffer (eBioscience, 00-4333-57). Isolated T cells were immediately stimulated and cultured as described below.

Patient-derived T cells were obtained from a 57-year-old female patient diagnosed with relapsed stage III multiple myeloma who previously received multiple lines of chemotherapy and autologous stem cell transplantation and was scheduled to receive anti-BCMA CAR T cell therapy as part of routine clinical care. The apheresis product for clinical CAR T cell production was collected using the Spectra Optia Apheresis System (Terumo BCT) with ACD-A as anticoagulant. T cells were isolated from leftover apheresis product using the EasySep Human T Cell Isolation Kit (Stemcell, 17951) following the manufacturer's instructions for leukapheresis samples. Isolated patient T cells were immediately stimulated and cultured as described below.

The study complied with all relevant ethical regulations for working with human samples. Informed consent was obtained from all sample donors. The study was approved by the ethical committees of the contributing institutions (Austrian Red Cross and Medical University of Vienna).

## T cell culture

T cell culture medium was based on CTS OpTmizer T-Cell Expansion Serum Free Medium (Thermo Fisher Scientific, A1048501): 1,000 ml of CTS OpTmizer T-Cell Expansion Basal Medium was supplemented with 26 ml of CTS OpTmizer T-Cell Expansion Supplement, 1% GlutaMAX (200 mM, 100×, Gibco, 35050061), 1% penicillin–streptomycin (10,000 U ml$^{-1}$, Gibco, 15140122) and 2% human AB serum (heat

inactivated according to the manufacturer's instructions, Sigma, H4522). The complete medium was supplemented with 12.1 ng ml$^{-1}$ human recombinant IL-2 (Stemcell, 78145.2) immediately before use. T cells were cultured at 37 °C with 5% CO$_2$. For stimulation, T cells were seeded at 1 million cells per ml in T cell medium with 25 μl ml$^{-1}$ ImmunoCult Human CD3/CD28 T Cell Activator (Stemcell, 10991). After 2–3 days of stimulation, the medium was exchanged completely. For collection, T cells were centrifuged at 400 rcf for 5 min. For optimization experiments, CD4$^+$ T cells were frozen after 1 week of pre-expansion in CTS OpTmizer T-Cell Expansion SFM with 20% human AB serum and 10% DMSO. Immediately after thawing, cells were stimulated and cultured. Cells were counted using CASY TT (Schärfe System, 3222).

### T cell transduction
T cells were collected 2–3 days after stimulation and seeded in fresh T cell medium at 1 million cells per ml. Lentivirus was added to the medium and mixed with cells. After 24–48 h, the medium was exchanged with a 4× larger volume. For single-locus targeting, 100× concentrated virus containing a single gRNA was added to the medium at a 1:10 (vol/vol) ratio. For screens, the lentivirus titre was quantified by qPCR, and lentivirus was added at 247 lentivirus RNA copies per cell for CRISPR screening in CD4$^+$ T cells and at 432 copies per cell for CRISPR screening in CD8$^+$ T cells.

### T cell electroporation
For electroporation using the ExPERT ATx Electroporator (Max-Cyte), cells were pelleted, washed once with room temperature 1× PBS, washed once with room temperature MaxCyte Buffer and then resuspended in MaxCyte Buffer at a concentration of 100 million cells per ml. Synthetic mRNA encoding CRISPR editors was added right before electroporation at a concentration of 150–300 μg ml$^{-1}$. mRNA for fluorescent markers or the blasticidin resistance gene (encoding BSD) was added at a concentration of 75 μg ml$^{-1}$. For RNP electroporation, synthetic gRNA (TrueGuide Synthetic gRNA, Synthego) and Alt-R S.p. Cas9 Nuclease V3 (IDT, 1081059) were mixed at a molar ratio of 2:1 and incubated for 30 min at room temperature for RNP complex formation. Immediately before electroporation, RNP was added to cells with a final gRNA concentration of 3 μM and a Cas9 concentration of 1.5 μM. Cells were mixed with mRNA or RNP and were transferred to MaxCyte cuvettes. Cuvettes were pulsed with a customized programme, Optimization 3-2. For electroporation using the Amaxa Nucleofector II system, cells were pelleted, washed once with room temperature 1× PBS and resuspended in 100 μl of Electroporation Master Mix from the Amaxa Human T Cell Nucleofector Kit (Lonza, 1002) immediately before electroporation at a maximum concentration of 15 million cells per ml and mixed with mRNA. Cells were transferred to the cuvettes and pulsed using the programme X-001. In both cases (electroporation with MaxCyte and Amaxa systems), cells remained in the cuvette for a 10 min recovery period and were then transferred to culture flasks with fresh prewarmed medium without puromycin at a concentration of 1 million cells per ml.

### CRISPR editing of single genomic loci
T cells were freshly isolated or thawed from samples frozen after 1 week of pre-expansion. T cells were stimulated in T cell culture medium supplemented with 25 μl ml$^{-1}$ ImmunoCult Human CD3/CD28 T Cell Activator (Stemcell, 10991). At 2–3 days after stimulation, cells were transduced with a lentivirus encoding the locus-specific gRNA (Supplementary Table 1c) and a fluorescent marker (mCherry) and the puromycin resistance gene (*PAC*). Selection with 1 μg ml$^{-1}$ puromycin was started 2–3 days after transduction and maintained until the end of the experiment. Cells were restimulated 7–9 days after stimulation. Cells were co-electroporated 2–3 days later with mRNAs encoding a CRISPR editor and a fluorescent marker or the blasticidin selection gene. When using *GFP* or *BFP* mRNA as a co-electroporated

selection marker, we gated successfully electroporated (BFP$^+$ or GFP$^+$) and transduced (mCherry$^+$) live T cells by flow cytometry or FACS. When using the blasticidin resistance gene (encoding BSD) as the co-electroporated selection marker, cells were selected with 50 μg ml$^{-1}$ blasticidin starting 24 h after electroporation, blasticidin was washed out after 1–2 days of selection and cells were further expanded for an additional 3–7 days.

**CRISPR knockout and base editing.** Cells were lentivirally transduced to express the locus-specific gRNA and electroporated with mRNA encoding Cas9, ABEmax, AncBE4max, ABEmax7.10-SpRY or CBE4max-SpRY (Supplementary Table 1b,c). After genome editing, cells were collected and genomic DNA was extracted. To measure editing efficiency at the DNA level, target loci were amplified using Q5 Hot Start High-Fidelity 2X Master Mix (NEB, M0494L) with locus-specific primers (Supplementary Table 1c). Amplicons shorter than 300 bp were sequenced directly. Longer amplicons were further fragmented with the Nextera XT Library Prep Kit (Illumina, 15032352). Libraries were sequenced on the Illumina MiSeq platform using MiSeq Reagent Kit v3 reagents (Illumina, MS-102-3001 or MS-102-3003). The frequency of insertions and deletions was determined based on the aligned sequencing reads. For CRISPR base editing, reads were considered successfully edited when they had at least one expected base conversion and no insertions or deletions. To assess editing at the protein level, we measured CD25 or CD44 levels by flow cytometry. For CD25, T cells were stimulated with 25 μl ml$^{-1}$ ImmunoCult Human CD3/CD28 T Cell Activator for 24 h before flow cytometry to induce CD25 expression. For CD44, cells were cultured normally, given that CD44 is stably expressed in human T cells without need for stimulation.

**CRISPR activation.** Cells were lentivirally transduced to express a gRNA targeting the *CD34* promoter. The gRNA backbone contained sequences for two MS2 aptamers for recruitment of the activators dCas9–VP64 and MCP–p65–HSF1, which were electroporated as separate mRNAs. Induction of CD34 surface protein was assessed by flow cytometry.

### CRISPR screening
For genome-wide knockout screens and tiling base-editing screens, T cells (either pan-CD3$^+$ or CD4$^+$ only, depending on the experiment and documented in Supplementary Tables 2, 3 and 8) were freshly isolated, pre-expanded for 7–10 days and restimulated once more. For focused knockout and base-editing screens, T cells were freshly isolated or thawed from samples frozen after 1 week of pre-expansion and used immediately. T cells were stimulated with 25 μl ml$^{-1}$ ImmunoCult Human CD3/CD28 T Cell Activator (Stemcell, 10991). At 2–3 days after stimulation, cells were transduced with CROP-seq-CAR lentivirus to deliver sequences for the CAR and the gRNA library and the puromycin resistance gene (*PAC*). At 2 days after transduction, cells were electroporated with mRNAs encoding the CRISPR editor and the blasticidin resistance gene (encoding BSD). Puromycin (0.5 μg ml$^{-1}$) and blasticidin (50 μg ml$^{-1}$) were added 24 h after electroporation. Puromycin selection was maintained until the end of the experiment. Blasticidin was removed after 24–48 h by a full exchange of cell culture medium. Cell numbers for transduction and electroporation were chosen to maintain at least 1,000× gRNA coverage (Supplementary Tables 2l and 3p).

For in vivo and combinatorial screens, CD4$^+$ T cells were stimulated with 25 μl ml$^{-1}$ ImmunoCult Human CD3/CD28 T Cell Activator (Stemcell, 10991). Two days after stimulation, cells were transduced with the lentivirus to deliver sequences for the CAR and the gRNA library and the puromycin resistance gene (*PAC*). Cells were subjected to puromycin selection (0.5 μg ml$^{-1}$) 2 days after transduction. Selected CAR T cells were restimulated 5 days after transduction. CAR T cells were electroporated 2 days later with mRNAs encoding the CRISPR editor and BSD and further cultured.

## Cancer cell lines

Antigen-expressing cell lines (K562-CD19 and K562-CD20) were produced by lentiviral transduction of K562 cells (ATCC, CCL-243) with the constructs pLenti-C-CD19-mGFP-P2A-Puro (OriGene, RC230267L4) or pLenti-C-MS4A1-mGFP-P2A-Puro (OriGene, RC221570L4). Cells were selected with 2 µg ml$^{-1}$ puromycin (10 mg ml$^{-1}$, Gibco, A1113802), which was maintained throughout cell culture. The suspension cell lines K562 and NALM6, the cell lines derived from them and firefly luciferase-expressing NALM6-GD2 cells were cultured in RPMI10 medium containing RPMI 1640 medium (Gibco, 11875085), 10% FBS (Sigma, F7524) and 1% penicillin–streptomycin (10,000 U ml$^{-1}$, Gibco, 15140122). Cells were split, and culture medium was exchanged every 3–4 days. For genetically engineered cell lines, antibiotic selection was maintained throughout cell culture. The adherent cell line Huh7 was cultured in high-glucose DMEM (Sigma, D5796) supplemented with 10% FBS (Sigma, F7524) and 1% penicillin–streptomycin (10,000 U ml$^{-1}$, Gibco, 15140122). Cells were split, and culture medium was replaced twice a week following standard procedures for adherent cells. For irradiation, cells were resuspended at 4 million cells per ml in RPMI10 medium and irradiated with 100 Gy using the Yxlon X-Ray System. Irradiated cells were centrifuged, washed once with fresh medium and either freshly used or frozen at 20 million cells per ml in freezing medium (RPMI1640 (Gibco, 11875085) with 10% FBS (Sigma, F7524) and 10% DMSO (Sigma, D2650)). The K562, NALM6 and NALM6-GD2 cell lines were authenticated using the Cell Line Identification service by Eurofins. The Huh7 cell line was authenticated by the G. Superti-Furga laboratory. All cell lines used in this study were negative for *Mycoplasma* contamination throughout the study, which was ensured by regular testing using PCR.

## Readouts of fitness screens

For TCR stimulation, cells were seeded at 1 million cells per ml in T cell medium containing 25 µl ml$^{-1}$ ImmunoCult Human CD3/CD28 T Cell Activator (Stemcell, 10991) and cultured for 2–3 days before medium exchange and further culture. For CAR stimulation, cells were seeded at 1 million cells per ml in T cell medium containing 1 million irradiated K562-CD19 target cells per ml. The medium contained 0.5 µg ml$^{-1}$ puromycin throughout and was exchanged every 2–4 days. On day 0 (before electroporation) and on days 7, 14 and 21 after electroporation, T cells were collected for gRNA sequencing. For screens of CD4$^+$ CAR T cells, cells were lysed for DNA isolation immediately. For screens of CD8$^+$ CAR T cells, cultures containing pan-CD3$^+$ T cells were stained with live–dead stain and anti-CD8 antibody, fixed and purified by FACS to obtain a CD8$^+$ CAR T cell population for genomic DNA isolation.

## Readouts of FACS-based screens

For flow cytometry profiling, the BD LSRFortessa cell analyser was used. For cell sorting, the Sony SH800S cell sorter was used for optimization experiments and the BD FACSAria Fusion system was used for genome-wide screening. Cells were stained as follows: the cells were pelleted, washed with 1× PBS, stained with 1/1,000 Zombie Viability Dye (BioLegend) at room temperature for 10 min, washed and pelleted, stained with a mix of the relevant antibodies in Cell Staining Buffer (BioLegend, 420201) at 4 °C for 30 min, washed and pelleted, resuspended in the desired volume and filtered through a cell strainer. For storage before cell sorting, cells were fixed in FluoroFix Buffer (BioLegend, 422101) at room temperature for 30 min in the dark, followed by washing cells twice and storing them in Cell Staining Buffer at a concentration of 50 million cells per ml. For large-scale screens, staining and fixation steps were carried out in 50-ml tubes placed on a rotator to avoid cell pelleting and clumping. The staining reagents used in this study are listed in Supplementary Table 1h. Flow cytometry data were analysed using FlowJo v.10.810.0.

## Amplification and sequencing of gRNAs

Genomic DNA was isolated using the DNeasy Blood & Tissue Kit (Qiagen, 69506) or the QIAamp DNA Blood Maxi Kit (Qiagen, 51194), depending on the number of cells. For cell lysis, we used AL buffer for freshly collected cells and ATL buffer for fixed and sorted cells. In both cases, cell lysates were incubated with proteinase K for 10 min at 56 °C, and fixed cells were additionally incubated for 4–12 h at 65 °C for DNA decrosslinking.

Library preparation for gRNA sequencing was performed as described in the CROP-seq protocol[2]. Up to 2 µg of purified genomic DNA was amplified and indexed with the primers CROPseq_libQC_i5 and CROPseq_libQC_i7 (Supplementary Table 1e), resulting in the sequencing-ready library. To avoid sequencing library overamplification, we added SYBR Green for qPCR quantification and stopped the amplification once it reached the exponential phase. For genome-wide screens, multiple PCR reactions were set up to retain 1,000× coverage for the individual gRNAs of the genome-wide gRNA library. Multiple 50-µl PCR reactions for the same sample were pooled together, and a minimum of 10% of the amplified material was used for purification. For plasmid libraries, we set up one to eight separate PCRs with 100 ng of input depending on the gRNA library size. We aimed for 1,000 sequencing reads per gRNA in the plasmid library. gRNAs amplified by PCR were cleaned by SPRI using Mag-Bind TotalPure Next-Generation Sequencing (NGS) Beads (Omega Bio-tek, M1378-01). Large DNA fragments were removed by cleanup with SPRI (0.45×) beads, followed by purification of the gRNA amplicon from the supernatant with a bead ratio of 1.0× to remove primer dimers.

Desired read numbers were calculated based on cell numbers and the estimated fraction of T cells with a gRNA, aiming for four to ten reads per cell. DNA concentrations for sequencing libraries were measured with the Qubit dsDNA HS assay (Thermo Fisher Scientific, Q32854), and 0.25 ng was analysed on a Bioanalyzer High Sensitivity DNA chip (Agilent, 5067-4626) to confirm the purity of the expected single band at 282 bp. Sequencing libraries were diluted to 3.5 nM with EB buffer containing 0.1% Tween and pooled according to the desired read number. Libraries were sequenced with 1% PhiX on the Illumina NovaSeq 6000 platform using a 100-cycle v1.5 flow cell with a read configuration of 122 bp for read 1, 8 bp for index 1 and 8 bp for index 2.

## Library preparation for in vivo CROP-seq

**Reverse transcription.** RNA was isolated using the AllPrep DNA/RNA Mini Kit (Qiagen, 80204) and quantified with the Qubit RNA HS Assay Kit (Invitrogen, Q32852). For reverse transcription, 33 µl of RNA sample per reaction (maximum 15 µg) was mixed with 3 µl of 10 mM dNTPs (Thermo Fisher Scientific, R0193) and 3 µl of 2 µM CART0065_RT_002 primer (Supplementary Table 1f). To resolve RNA secondary structures, the reaction was incubated at 65 °C for 5 min and immediately placed on ice to prevent reformation. Next, a master mix of 12 µl of 5× SuperScript IV Buffer, 3 µl of 100 mM DTT (Sigma-Aldrich, 646563-10x.5ML), 3 µl RNaseOUT RNase inhibitor (Invitrogen, 10777019) and 3 µl SuperScript IV Reverse Transcriptase (Thermo Fisher Scientific, 18090010) was added. The reverse transcription reaction was incubated at 55 °C for 10 min, followed by heat inactivation for 10 min at 80 °C, and then placed on ice. Template RNA was digested with 3 µl RNase A (NEB, T3018L) for 30 min at 37 °C. Afterward, the cDNA was cleaned using 108 µl of Mag-Bind TotalPure NGS Beads (Omega Bio-tek, M1378-01), eluting in 40 µl nuclease-free water. Several reactions were carried out to ensure processing of the entire RNA isolated for each organ.

**Nested PCR.** The PCRs were set up with 19.5 µl template DNA, 25 µl Q5 High-Fidelity 2X Master Mix (NEB, M0492L), 2.5 µl of 10 µM FWD and REV primers and 0.5 µl of 100× SYBR Green. PCRs were incubated at 98 °C for 30 s (initial denaturation) and then cycled between 98 °C for

10 s and 72 °C for 40 s in a Bio-Rad CFX96 qPCR cycler until the reaction reached >1,500 RFU. PCRs were finished in a second cycler at 72 °C for 2 min and then cooled on ice. PCRs were cleaned with SPRI (0.8×) using Mag-Bind TotalPure NGS Beads (Omega Bio-tek, M1378-01). The outer PCR used primers CART0065 FWD-001 and CART0065 REV-001 (Supplementary Table 1f). The inner PCR used CROPseq_libQC_i5 with different stagger lengths and i5 indices, and TruSeq_i7 with different i7 indices. Primer sequences are listed in Supplementary Table 1g, and a schematic overview is provided in Supplementary Fig. 4a,b.

**NGS library quality control and sequencing.** Sequencing libraries were quantified with the Qubit dsDNA HS assay (Thermo Fisher Scientific, Q32854). A portion (1.5 ng) of the sequencing library was run on a Bioanalyzer High Sensitivity DNA chip (Agilent, 5067-4626) to verify the final size of 288–294 bp (depending on stagger length). Sequencing libraries were diluted to 2.0 nM with EB buffer containing 0.1% Tween and sequenced on the Illumina NovaSeq 6000 platform using 100-cycle v1.5 reagents with a read configuration of 50 bp for read 1, 8 bp for index 1, 8 bp for index 2 and 26 bp for read 2.

### Library preparation for combinatorial screens
**Amplification of plasmid gRNA library.** Amplification of the plasmid library was performed as described previously[46], using 2× NEBNext Ultra II Q5 Master Mix (NEB, M0544), 2.5 µl of 10 µM FWD and REV primers (Supplementary Table 1j), 0.5 µl of 100× SYBR Green and 100 ng plasmid template, with the following cycling conditions: 98 °C for 1 min, followed by ten cycles of 98 °C for 10 s, 67 °C for 10 s and 72 °C for 15 s, followed by 72 °C for 1 min. The PCR reactions were cleaned with 1× Mag-Bind TotalPure NGS Beads (Omega Bio-tek, M1378-01). Quality control and dilution were performed as described above.

**Amplification of genomic DNA from combinatorial screens.** DNA amplification was performed in a Bio-Rad CFX qPCR cycler to monitor amplification using 2× NEBNext Ultra II Q5 Master Mix (NEB, M0544), 2.5 µl of 10 µM FWD and REV primers (Supplementary Table 1j), 0.5 µl of 100× SYBR Green and 2.5 µg genomic DNA with the following cycling conditions: 98 °C for 1 min, 26–32 cycles of 98 °C for 10 s, 67 °C for 10 s and 72 °C for 15 s with plate reading, followed by 72 °C for 1 min. Amplification was stopped three cycles after SYBR Green enrichment detection to avoid overamplification and unwanted recombination events. The PCR reactions were cleaned with 1× Mag-Bind TotalPure NGS Beads (Omega Bio-tek, M1378-01). Quality control and dilution were performed as described above. The reference samples were collected on the day of electroporation (day 0). Five days after electroporation, cells were restimulated with irradiated target cells (K562-CD19 or NALM6-GD2) and cultured without puromycin (as NALM6-GD2 cells are not puromycin resistant). Samples were collected for genomic DNA extraction 12 days after electroporation.

### Luciferase reporter cell lines
Synthetic DNA fragments encoding firefly luciferase, a P2A self-cleaving peptide and the zeocin resistance gene (Luc-P2A-Zeo, all human codon optimized) were ordered as gBlock (IDT). The lentiCas9-Blast plasmid (Addgene, 52962) was digested with EcoRI and XbaI to replace the insert for Luc-P2A-Zeo. Lentivirus was produced with the Lipofectamine 3000 reagent. NALM6 and K562-CD19 cells were lentivirally transduced by spinfection as described previously[2] and selected for 2–4 weeks using zeocin.

### Luciferase assay on microwell plates
CAR T cells were co-cultured with luciferase-expressing K562-CD19 cells for 18 h. Cells were transferred to a 384-well assay plate (Corning, 3707). For lysis and luciferase detection, we used the Steady-Glo Luciferase Assay System (Promega, E2510) and a PerkinElmer Victor x3 2030 Multilabel Reader.

### CAR T cell production for mouse validation experiments
**Gene knockout using mRNA electroporation.** Freshly isolated CD3+ T cells were transduced with CROP-seq-CAR lentivirus containing a pool of eight gRNAs for each target gene. Four gRNAs were taken from the genome-wide screen, and four additional gRNAs were designed using the VBC score method[55]. Selected gRNAs usually had no (but up to a maximum of two) off-target sites with two or less mismatches based on Cas-OFFinder[56] predictions (Supplementary Table 5h). Selection with puromycin (0.5 µg ml⁻¹) was started 2 days later. Cells were restimulated 7 days after isolation. Cells were electroporated 3 days later with Cas9 and mRNA encoding BSD. Blasticidin (50 µg ml⁻¹) was added 24 h later. On the following day, cell culture medium was exchanged, and cells were cultured for an additional 3 days. Cells were injected into mice 5–6 days after electroporation.

**Gene knockout using RNP electroporation.** Freshly isolated CD3+ T cells were stimulated with 25 µl ml⁻¹ ImmunoCult Human CD3/CD28 T Cell Activator (Stemcell, 10991) for 2 days before electroporation with RNPs as described above, directly followed by transduction with CROP-seq-CAR lentivirus. Puromycin selection (0.5 µg ml⁻¹) was started 2 days later. Cells were restimulated 7 days after isolation with 25 µl ml⁻¹ ImmunoCult Human CD3/CD28 T Cell Activator (Stemcell, 10991). Cell culture medium was exchanged 3 days after restimulation, and cells were cultured for an additional 3–4 days. At 5–6 days after electroporation, the cells were either directly injected into mice or frozen in Bambanker freezing medium (Nippon Genetics, BB02). CAR T cells were thawed 1 day before injection into mice in the presence of 0.1 mg ml⁻¹ DNase I (Sigma, 11284932001).

### Mouse in vivo xenograft cancer models
Mice were bred and maintained under specific-pathogen-free conditions at the Medical University of Vienna, which is authorized for animal breeding and experimentation under license BMWFW-66.009/0403-WF/V/3b/2014. Experiments were performed at the Core Facility Laboratory for Animal Breeding and Husbandry of the Medical University of Vienna and the Preclinical Imaging Laboratory at the Department for Biomedical Research of the Medical University of Vienna using individually ventilated cages with controlled temperature (20–24 °C) and humidity (30–70%) and a 12-h light–dark cycle. Experiments followed established guidelines and were approved by the institutional ethical committee at the Department for Biomedical Research of the Medical University of Vienna according to license BMWFW-2020-0.605.586 granted by the Austrian Federal Ministry of Education, Science and Research (BMBWF). Investigators were not blinded to treatment status during animal experiments, as they also conducted the outcome assessment; however, stringent objective criteria were established for data collection and analysis to avoid any potential biases. The determination of group sizes and the power calculation were carried out according to the approved animal license (BMWFW-2020-0.605.586).

**Leukaemia xenograft model.** NSG male or female mice at 8–12 weeks of age were intravenously injected with 0.5 million firefly luciferase-expressing NALM6 cells (clone G5, ATCC, CRL-3273) or 0.5 million firefly luciferase-expressing NALM6-GD2 cells in 150 µl of 1× PBS. After 5 days, CAR T cells were administered by intravenous injection. For in vivo screens, female mice were injected with 1 million CAR T cells per mouse. For single-knockout validation, we injected male mice with 0.6 million CAR T cells in 150 µl of 1× PBS per mouse. Mice were tracked by bioluminescence imaging and weight measurements and were monitored for any signs of morbidity. All experiments complied with the humane and ethical end points stated in the animal license (BMWFW-2020-0.605.586), including maximum weight loss. Owing to the fast and uniform engraftment of NALM6 cells, no randomization

was performed. To reduce biases introduced by spillover during bioluminescence imaging, we imaged treatment groups together whenever possible.

**Solid tumour xenograft model.** NSG female mice at 8–12 weeks of age were subcutaneously injected (right flank) with 1 million Huh7 cells in 100 μl of 1× PBS. After 20 days and with an average tumour volume of 0.3–0.4 cm³, CAR T cells were administered by intravenous injection. Tumour size was tracked by measurement with a precision caliper. Mice with Huh7 solid tumour xenografts were randomized before CAR T cell injection according to tumour size to ensure that all groups had an equivalent tumour load. All experiments complied with the humane and ethical end points stated in the animal license (BMWFW-2020-0.605.586), including maximum tumour size.

## IVIS bioluminescence imaging
XenoLight D-Luciferin K⁺ salt (PerkinElmer, 122799) was dissolved at 30 mg ml⁻¹ in sterile 1× PBS and injected intraperitoneally at a concentration of 150 mg per kg body weight. Mice were anesthetized with isoflurane, and bioluminescence imaging was performed 15 min after injection on the IVIS Spectrum (PerkinElmer, 124262) or the LagoX instrument (Spectral Instruments Imaging). Bioluminescence was quantified using Living Image Analysis Software v.4.7.3 (PerkinElmer) or Aura In Vivo Imaging Software v.4.5.0 (Spectral Instruments Imaging).

## Organ dissociation
During extraction, organs and cells were kept on ice for the entire procedure. Spleens were rinsed with ice-cold 1× PBS with 5 mM EDTA and 0.1% BSA (PBS + EDTA + BSA) and smashed through a 70-μm cell strainer, and the resulting cell suspensions were collected in 50-ml tubes. Femurs from both hindlimbs were collected, and soft tissue was removed with scissors. Bone marrow was flushed with PBS + EDTA + BSA. Cells were filtered through a 70-μm strainer and transferred to 50-ml tubes. Cell suspensions from both organs were centrifuged at 500 rcf for 5 min and resuspended in 3 ml of 1× RBC Lysis Buffer (eBioscience, 00-4333-57) for 5 min to lyse red blood cells; the reaction was stopped by adding 20 ml of cold PBS + EDTA + BSA. Supernatants were filtered through a 70-μm cell strainer, and cells were counted using the CASY TT (Schärfe System, 3222) system.

## RNA-seq profiling of CAR T cells
CAR T cells were produced from pan-CD3⁺ T cells as the starting material. CD4⁺ and CD8⁺ CAR T cells were isolated for bulk RNA-seq using the EasySep Human CD4 Positive Selection Kit II (Stemcell, 17852) and the EasySep Human CD8 Positive Selection Kit II (Stemcell, 17853). RNA was isolated using the Monarch Total RNA Miniprep Kit (NEB, T2010S), starting from at least 200,000 lysed cells, and the extracted RNA was stored at −80 °C in 300 μl Protection Reagent. RNA-seq libraries were prepared with the NEBNext Ultra II Directional RNA Library Prep Kit for Illumina (NEB, E7760L) and sequenced with paired-end 50-bp reads on the Illumina NovaSeq 6000 platform.

## Data preprocessing for the CRISPR screens
Sequencing data were preprocessed with a custom pipeline, starting from demultiplexed but unaligned BAM files provided by the Biomedical Sequencing Facility at CeMM. BAM files were converted to FASTQ format using bedtools v.2.30.0 bamtofastq. Reads were trimmed with cutadapt v.3.4, using the -g flag to specify the 5′ adapter and stagger (Supplementary Table 1e). For the in vitro screens, raw gRNA counts were obtained with the MAGeCK package v.0.5.9.4 using the mageck count command, specifying gRNA sequences for alignment with the -l option. For in vivo CROP-seq screens, we used MAGeCK2, cloned from the GitHub repository davidliwei/mageck2 and the mageck2 count command, providing the gRNA sequences with the -l flag and defining the UMI configuration with --umi secondpair,0,26. Finally, quality-control

metrics were derived from MAGeCK log files, including total reads, mapped reads, gRNA alignment rate, gRNAs with zero counts, Gini index and read-trimming statistics.

## Analysis of the genome-wide fitness screens
**MAGeCK RRA analysis.** Samples obtained after TCR or CAR stimulation on days 7, 14 or 21 after electroporation were separately compared with samples collected before stimulation (day 0). The analysis was performed with MAGeCK package v.0.5.9.4 using the mageck test command. The -k option was used to specify the raw gRNA count table in TSV format. Samples were provided with -t and -c flags. We used --remove-zero both --remove-zero-threshold 0 to remove gRNAs with zero reads in both of the two compared samples. For the normalization method, we used --norm-method control, and we used --control-gene to provide a set of previously described negative control genes for human samples (gene set NEGv1)[57]. Day 0 samples were selected for variance estimation with the --variance-estimation-samples option. Essential genes were defined with a stringent threshold of $\log_2[\mathrm{FC}] < -1.5$ and FDR < 0.01. Gene set enrichment for gene ontology (GO) terms labelled BP (for Biological Process) was performed in R with the clusterProfiler package (v.4.0.0)[58], using the function compareCluster with extra arguments fun = 'enrichGO', OrgDb = 'org.Hs.eg.db', keyType = 'SYMBOL' and ont = 'BP'. Next, the similarity between nodes was calculated with the pairwise_termsim function from the enrichplot package using the default Jaccard similarity coefficient method. The network graph was plotted using the enrichplot::emapplot function with parameters showCategory = 100, node_scale = 10, min_edge = 0.3, cex_line = 0.5, cex_category = 10, pie = 'Count' and cex_label_category = 1.

**MAGeCK MLE analysis.** Samples obtained before (day 0) and after (day 14) TCR or CAR stimulation were compared in one joint analysis with MAGeCK package v.0.5.9.4 using the mageck mle command. Raw gRNA counts in TSV format were provided with --count-table, and the experimental design was defined with the --design-matrix option. In the design matrix, the day 14 samples were modelled as TCR stimulated or CAR stimulated (Supplementary Table 2j). We further used --norm-method control and provided a set of non-essential genes (gene set NEGv1)[57] via the --control-gene option. The analysis was performed with --permutation-round 2. The MAGeCK MLE gene_summary.txt output file was read with R package MAGeCKFlute v.1.12.0 using the ReadBeta function. Normalizing $\beta$ values against essential genes is important when samples in the screen have different proliferation rates, as it is the case for TCR- and CAR-stimulated versus unstimulated T cells (Fig. 1b and Extended Data Fig. 3d). We thus normalized $\beta$ values with the NormalizeBeta function, specifying a set of essential genes with the arguments posControl (CEGv2) and method = 'cell_cycle'. Finally, we selected screening hits that improved the function of both T and CAR T cells and of CAR T cells alone.

## Analysis of the genome-wide FACS-based screens
The sequencing data from the genome-wide sorting screens for the markers CD19, CD69, FAS and the combination of PD-1, LAG3 and TIM3 were preprocessed as described above and filtered with quality-control thresholds of gRNA alignment ≥62.5%, ≥2 mapped reads per sorted cell and ≥100 mapped reads per gRNA in the library. The trimming auto-determination result of MAGeCK was zero for ≥96% of reads, confirming correct adapter annotation and usage. For the CD69 screens, each of the three positive populations (CD69⁺, CD69⁺⁺ and CD69⁺⁺⁺) and all their combinations were individually analysed with MAGeCK RRA, and we selected the CD69⁺ population for the main comparison, as it has the most consistent signal owing to higher numbers of sorted cells than for the CD69⁺⁺ and CD69⁺⁺⁺ populations. For the FAS screens, in which three negative populations (FAS⁻⁻⁻, FAS⁻⁻ and FAS⁻) were sorted, the FAS⁻⁻⁻ population showed the strongest enrichment of the screening marker and was selected for the main comparison.

All analyses were performed with MAGeCK package v.0.5.9.4 using the RRA algorithm with the mageck test command. We used the options --remove-zero both --remove-zero-threshold 0 to remove gRNAs with zero reads in both of the two compared samples. We specified the raw gRNA count matrix with -k, test samples with -t and unsorted reference samples with -c. Furthermore, reference samples were also provided with --variance-estimation-samples. For read normalization, we used --norm-method median and provided a set of non-essential genes (gene set NEGv1)[57] via the --control-gene option. All comparisons were run in --paired mode.

Genes were selected for inclusion in the in vivo CROP-seq experiment based on the results of the TCR and CAR fitness screens (normalized MAGeCK MLE $\beta > 0.99$) and the FACS-based screens (log$_2$[FC] > 1.5, FDR < 0.01), with RNA levels in human CAR T cells serving as an additional filter (Extended Data Fig. 5n and Supplementary Table 6b).

### Analysis of in vivo CROP-seq screens

Sequencing data from the in vivo CROP-seq screens were preprocessed as described above. Reads with UMIs containing an N were discarded. Different versions of the raw gRNA and UMI count table were prepared. For clonal tracking, the full UMI information was retained. For internal replicate analysis, only UMI bases 1 (four internal replicates), 1–2 (16), 1–3 (64), 1–4 (256) and 1–5 (1,024 internal replicates) were used. Finally, a table was prepared in which all UMI information was discarded. All tables were further processed with mageck-ibar (https://bitbucket.org/WeiLab/mageck-ibar.git). Screening hits were defined using a stringent threshold of log$_2$[FC] > 1.5 and FDR < 0.01. To determine the number of T cell clones in each sample, we prepared a knee plot by visualizing log$_{10}$[rank] against log$_{10}$[reads] for each UMI. To find the inflection point that separates T cell clones from background noise, we modelled a sigmoid curve of form $L/\{1 + \exp[-k \times (x - x_0)]\} + b$, where $L$ is the maximum value, $k$ is the steepness and $b$ is the vertical offset, using the nls function from the R package stats. For modelling the log-transformed data, we disregarded the top 50 UMIs with the highest read count and all UMIs with fewer than ten reads. The coefficient $x_0$ represents the inflection point that separates T cell clones from background noise.

### RNA-seq data processing

RNA-seq read data (unaligned BAM files) were aligned to the hg38 genome assembly using an adapted STAR pipeline[59]. The resulting count matrix was split by cell type and time point, and all downstream analyses were performed for each dataset separately. The analyses and visualizations described here were performed using a publicly available Snakemake (7.21.0)[60] workflow (v.1.0.1)[61]. Transcripts were filtered using the filterByExpr function from the R package edgeR (v.3.32.1)[62] with the following parameters: group set to cell type and time point, min.count to 30, min.total.count to 50, large.n to 20 and min.prop to 0.8. Filtering reduced the number of analysed transcripts from 60,676 to 15,856. The data were normalized using conditional quantile normalization from the R package cqn (1.44.0)[63]. The parameters used for this normalization included gene length and GC content. The normalization results were log$_2$ transformed for downstream analyses. Batch effects related to T cell donor were corrected for by using the reComBat method[64] based on the log-transformed data. Normalization and batch correction were performed on the entire dataset for downstream visualization by PCA and visualization of predefined signatures.

### Differential expression and gene set enrichment analysis

Differential expression analysis was performed on the quality-controlled, filtered and normalized transcript counts using limma (3.46.0)[65] to fit a linear model for detecting statistically significant transcripts in *RHOG*-knockout compared with standard CAR T cells. We applied a publicly available Snakemake workflow (v.1.0.2)[66]. Briefly, we used lmFit to fit the model to the data and the Bayes command of limma with incorporated mean-variance trend to compute *t*-statistics.

For each comparison, we used topTable to determine transcript-wise average expression, effect sizes (log$_2$[FC]) and statistical significance (adjusted *P* values based on the Benjamini–Hochberg method). Furthermore, we calculated transcript scores for each transcript in each comparison using the formula $-\log_{10}[P\text{ value}] \times \text{sign}(\log_2[\text{FC}])$ for use by the downstream ranked enrichment analyses.

Gene set enrichment analysis was performed with a publicly available Snakemake workflow (v.0.1.1)[67] using GSEA (https://www.genepattern.org/modules/docs/GSEAPreranked/1) with the prerank function from the GSEApy 1.0.3 package[68]. 'GO biological process sets' gene sets were selected (http://www.broadinstitute.org/gsea/msigdb). Enrichment scores (ESs) reflect the degree to which a gene set is overrepresented at the top or bottom of a ranked list of genes, and normalized ESs are ES scores normalized by mean ES across all comparisons for the dataset. The results of all queries were aggregated by method and database. Additionally, we filtered the results by retaining only the union of terms that were statistically significant (adjusted *P* value < 0.05) in at least one query. The aggregated results were visualized using hierarchically clustered heat maps and bubble plots. The union of the most significant terms per query was determined, and their effect size and significance were visualized as a hierarchically clustered bubble plot encoding both effect size (colour) and statistical significance (size), with statistical significance denoted by an asterisk. For the summary visualizations, *P* values were capped at an adjusted *P* value of 0.0001 to minimize the visual effect of genes with very low *P* values. Clustering of enriched terms was carried out using the R clusterProfiler package v.4.8.1 emapplot function[58]. Cell cycle-related genes were defined from a gene signature[69].

### Analysis of combinatorial screens

Combinatorial screening data were processed as follows: Raw reads were extracted from unaligned BAM files (samtools view with flag -f 64 for read 1 and -f 128 for read 2), joined by the read identifier (column 1 or QNAME of the BAM), and CROP-seq-multi features (spacer1, iBAR1, spacer2 and iBAR2) were extracted based on their position. To produce a count matrix for downstream analyses (Supplementary Table 7c), we matched the combination of spacer1, iBAR1, spacer2 and iBAR2 to the combinatorial gRNA library (Supplementary Table 7a, zero mismatches allowed). We calculated additional quality-control metrics, such as percent of perfect matches to library features, fold difference between the 10th and 90th percentiles of library counts and percent recombination between spacer–iBAR pairs (Supplementary Table 7d). The processed data were analysed with MAGeCK MLE, using a design matrix modelling the 19-BBz, 19-28z and GD2–BBz CARs (Supplementary Table 7e), with options --permutation-round 10 --max-sgrnapergene-permutation 40. We normalized the data relative to dual safe harbour gRNAs with the --norm-method control and --control-sgrna options. We provide sample annotation (Supplementary Table 7b), gRNA counts (Supplementary Table 7c) and MAGeCK MLE output (Supplementary Table 7f).

### Comparison of screening quality with published datasets

We obtained CRISPR screening data for primary T cells from Shifrut et al.[22], Wang et al.[20], Carnevale et al.[15] and Freitas et al.[16] and data for the Jurkat cell line from DepMap release 24Q4 (ref. 34). Details on the data sources are provided in Supplementary Table 4b. For datasets for which raw gRNA counts were available (Shifrut et al.[22] and Freitas et al.[16]), we calculated log$_2$[FC] between day 16 (Shifrut et al.[22]) or day 20 (Freitas et al.[16]) to the plasmid reference. For the Jurkat data, log$_2$[FC] values between day 21 and the plasmid reference were directly available (AvanaLogfoldChange.csv), and, for Carnevale et al.[15], we obtained log$_2$[FC] values from the MAGeCK RRA output. For Wang et al.[20], only $\beta$ values but not the full MAGeCK MLE output were available; therefore, we reprocessed our own data with the same MAGeCK MLE settings. To compare the dropout of essential genes across screens while taking the variance into account, we used the strictly standardized mean difference (SSMD) metric, calculated as

$$\text{SSMD} = \frac{\overline{X}_e - \overline{X}_{ne}}{\sqrt{s_e^2 + s_{ne}^2}},$$

where $\overline{X}_e$ and $\overline{X}_{ne}$ are the mean $\log_2[FC]$ of essential and non-essential genes, and $s_e$ and $s_{ne}$ are the corresponding standard deviations. We used sets of 684 essential genes (CEGv2) and 928 non-essential genes (NEGv1), both from the BAGEL2 publication[57]. Screens with an SSMD < 0.5 were flagged as low quality and excluded from the downstream hit analysis.

### Comparison of screening results with published datasets

We obtained MAGeCK output files for all high-quality screens (SSMD > 0.5): Shifrut et al.[22] (TCR fitness, CGS-21680), Carnevale et al.[15] (regulatory T cell, TGFβ, TCR fitness, adenosine), Freitas et al.[16] (IL-2, TNF; CAR fitness) and this study (TCR fitness, CAR fitness; FAS; PD-1, LAG3, TIM3; CD69; CD19). Most screens were processed with MAGeCK RRA, in which we defined significant hits as $z$-normalized $\log_2[FC] > 2$ and FDR < 0.05. For the TCR and CAR fitness screens in this study, we used MAGeCK MLE with $\beta$-value normalization for essential genes via MAGeCKFlute to control for the different proliferation rates between TCR- and CAR-stimulated T cells, with a selection threshold of normalized $\beta > 0.99$. For the heat map in Extended Data Fig. 4g, we included all significant hits except those from our fitness screens, for which the high number of significant genes required us to limit the selection to the genes tested in vivo.

### Base-editing gRNA library design

We first designed all possible gRNAs for each target gene, including all exons and ±20 bp into each intron or UTR. Editing outcomes were annotated using the Base Editor Design Tool (https://github.com/mhegde/base-editor-design-tool#base-editor-design-tool)[48]. We excluded gRNAs containing stretches of four or more T nucleotides, which can lead to premature termination of gRNA transcription. We also excluded gRNAs with BsmBI cut sites because the PCR-amplified gRNAs were cloned with Gibson assembly, during which the backbone was cut by BsmBI.

For the PAM-less tiling screens of *RHOG* and *PAC*, all possible gRNAs were used. For essential genes, we designed gRNAs introducing specific mutation types, depending on the base editor: 50 splice site mutations for ABEmax, 50 splice site and 50 nonsense mutations for AncBE4max, 200 splice site mutations for ABEmax7.10-SpRY and 200 splice site and 200 nonsense mutations for CBE4max-SpRY. To that end, we filtered gRNAs introducing nonsense or splice site mutations in the first half of the transcript, where the target base is in positions 5–7 (which corresponds to the central part of the editing window). We selected essential genes according to their rank in the MAGeCK RRA analysis of our genome-wide fitness screens, with a limit of four gRNAs per gene. For control gRNAs targeting a safe harbour locus, we designed gRNAs such that 50% of them contained ≥1 A but no C and 50% contained ≥1 C but no A in the editing window, such that they could serve as nontargeting controls for C-to-T and A-to-G editors, respectively. Each group contained 25% gRNAs with an NGG PAM and 25% gRNAs with a non-NGG PAM (Supplementary Table 8a).

For the base-editing validation screens, we selected gRNAs based on the $z$ score from the initial screen, including the 3.5% most enriched gRNA for *RHOG* and the 3.5% most depleted gRNA for *PAC* based on the average $z$ score of the two donors and for each donor separately and additional safe harbour-targeting control gRNAs, resulting in a final library of 323 gRNAs (Supplementary Table 8e).

### Base-editing analysis

For plasmid and pre-electroporation reference samples, sequencing reads were aligned to the gRNA library. As the gRNAs are integrated into the genome, they may be susceptible to self-editing by near-PAM-less base editors upon electroporation. Therefore, we created expanded reference files separately for the A-to-G and C-to-T editors, containing all original gRNA sequences and sequences resulting from self-editing in the editing window (gRNA positions 4–8). We aligned reads from base-edited samples to these references and produced gRNA-level count files (Supplementary Table 8c). $\log_2[FC]$ values were calculated by comparing samples from day 14 (screening readout) and day 0 (pre-electroporation reference) using MAGeCK RRA with parameters --norm-method median, --remove-zero both, --remove-zero-threshold 0, --paired and --variance-estimation-samples, pairing samples by T cell donor (2 donors for the initial screening and five donors for the base-editing validation screens) and using samples from day 0 for variance estimation. For visualization, protein 3D structures were downloaded from the PDB, and the 15 amino acids corresponding to the most impactful base-editing gRNAs were drawn as stick representation using PyMOL. Hydrogen bonds were computed between the highlighted amino acids and the small-molecule reagents using the following PyMOL command: distance hbonds, small_molecules, top15_aas, mode = 2.

### Base-editing validation

**Base-editing validation screens.** CD4+ T cells were lentivirally transduced with the CROP-seq-CAR vector containing a focused gRNA library for base-editing validation screens (Supplementary Table 8e) and electroplated with base editor mRNA as described above. gRNA amplification was performed as described above for all base editors. For genomic locus sequencing, ABEmax-edited CAR T cells were collected 2 and 12 days after electroporation. A 200-bp region of the *RHOG* locus that contains high-scoring gRNAs from the focused validation screens was amplified by PCR with primers containing overhangs compatible with TruSeq indexing (Supplementary Table 1c) using Q5 Hot Start High-Fidelity 2X Master Mix (NEB, M0494L), followed by SPRI (1.2×) bead purification using Mag-Bind TotalPure NGS Beads (Omega Bio-tek, M1378-01). The purified PCR fragments were indexed by PCR using TruSeq indexing primers (Supplementary Table 1d) to prepare them for sequencing on the Illumina NovaSeq 6000 platform.

**Single-gRNA validation.** CD4+ T cells were lentivirally transduced with CROP-seq-CAR vectors, each harbouring a single base-editing gRNA, targeting either the *RHOG* or *PAC* gene. After 7 days of puromycin selection including 2 days of restimulation, CAR T cells were electroporated with mRNA encoding the ABEmax base editor (Supplementary Table 1b). Three days after electroporation, a portion of cells was collected for genomic DNA extraction and the remaining cells were restimulated with 25 µl ml$^{-1}$ ImmunoCult Human CD3/CD28 T Cell Activator (Stemcell, 10991) and further cultured in the presence of puromycin, until they were collected for genomic DNA extraction on day 12 after electroporation. To assess editing efficiency at the target loci on the DNA level, the *RHOG* and *PAC* loci were amplified by PCR using Q5 Hot Start High-Fidelity 2X Master Mix (NEB, M0494L) with locus-specific primers (Supplementary Table 1c). The resulting amplicons from samples collected on day 3 and day 12 after electroporation underwent Sanger sequencing, and editing efficiency was analysed using the EditR tool[70] with default parameters.

### Material availability

Plasmids generated in this study, including the CROP-seq-CAR and mRNA production vectors, have been deposited at Addgene (Supplementary Table 1).

### Reporting summary

Further information on research design is available in the Nature Portfolio Reporting Summary linked to this article.

## Data availability

RNA-seq data are available from GEO (accession number GSE266618). CRISPR screening data are provided in Supplementary Table 2 (fitness screens), Supplementary Table 3 (FACS screens), Supplementary Table 5 (in vivo screens), Supplementary Table 7 (combinatorial screens) and Supplementary Table 8 (base-editing screens). Source data are provided with this paper.

## Code availability

All analyses were based on publicly available bioinformatic methods and software (see the Methods for details).

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

**Acknowledgements** We thank the Austrian Red Cross (E. Espiritu, C. Feinböck, B. Huber and R. Geretschläger) for providing healthy donor blood samples; A. Müller and S. Winter at the Department of Transfusion Medicine and Cell Therapy of the Medical University of Vienna for providing a leukapheresis product from a patient scheduled to receive CAR T cell therapy; the Biomedical Sequencing Facility at CeMM for assisting with NGS; the Core Facility Flow Cytometry of the Medical University of Vienna for providing access to cell-sorting equipment; the Core Facility Laboratory Animal Breeding and Husbandry and the Preclinical Imaging Laboratory of the Medical University of Vienna for hosting the animal work and providing access to materials, services and equipment; the C. Mackall laboratory for providing the NALM6-GD2 cell line; the G. Superti-Furga laboratory for providing the Huh7 cell line; N. Winhofer and C. Ricci-Tam for graphical refinement of the figures; and members of the Bock laboratory for advice and discussions. This work was supported by an ERC Consolidator Grant (grant agreement no. 101001971; grantee, C.B.) of the European Union's Horizon 2020 research and innovation programme and an Austrian Science Fund (FWF) Special Research Program grant (F7001; grantee, C.B.).

**Author contributions** P.D. conceived and designed the CELLFIE platform; P.D. and E.V.P. developed the CELLFIE platform and led experiments; P.D., E.V.P. and C.D.A. planned and performed screening and validation experiments with assistance from N.P., J.L., A.-C.O., A.N., P.N.B., M.C. and T.N.; S.L. and T.K. prepared the animal license and oversaw in vivo experiments; P.D., E.V.P. and C.D.A. planned and performed in vivo experiments with assistance from N.P., A.-C.O., A.N., F.P., W.L. and T.K.; T.K. provided feedback on the T cell experiments; P.D. and E.V.P. led and performed data analysis with contributions from D.R., M.S. and A.V.A.; P.D. and E.V.P. visualized results, prepared figures and wrote the draft manuscript; P.D., E.V.P. and C.B. wrote the manuscript with contributions from all authors; P.D. and C.B. supervised the project.

**Funding** Open access funding provided by Medical University of Vienna.

**Competing interests** P.D., E.V.P. and C.B. are inventors on patent applications related to the CELLFIE platform and boosters of immunotherapy. C.B. is a cofounder of and scientific advisor to Myllia Biotechnology and Neurolentech. P.D. is a shareholder in Xaira Therapeutics through employee stock options. The remaining authors declare no competing interests.

**Additional information**
**Correspondence and requests for materials** should be addressed to Paul Datlinger or Christoph Bock.

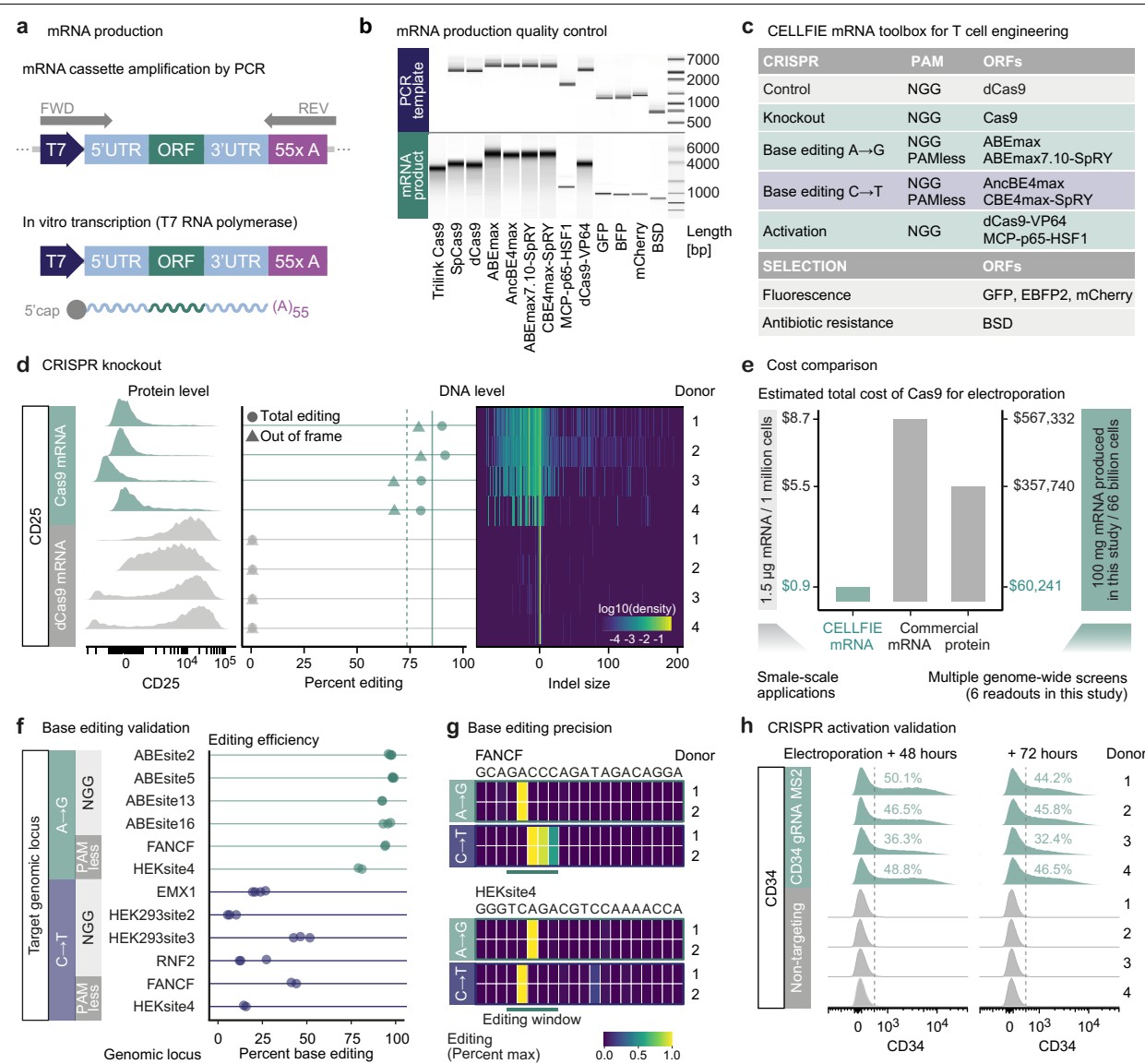

**Extended Data Fig. 1 | CELLFIE: a versatile platform for human CAR T cell engineering. a**, Design of the mRNA production plasmid with the open reading frame (ORF) of the CRISPR editor or selectable marker between untranslated regions (UTRs) to stabilize the mRNA. For mRNA production, the PCR template is amplified (top) and used for in vitro transcription (IVT) from the T7 promoter with co-transcriptional capping and transcription termination at a hardcoded A-tail of 55 bases. **b**, Representative size profiles of PCR templates and in vitro transcribed mRNA for CRISPR editors and selection markers. The leftmost sample is commercial Cas9 mRNA (Trilink), all other samples were produced with CELLFIE. **c**, Validated synthetic mRNAs for delivery of CRISPR editors and selection markers into human primary T cells and CAR T cells. **d**, Efficient knockout of IL2RA (CD25) at the DNA and protein level in CD4+ T cells (4 donors). **e**, Cost of custom-made synthetic Cas9 mRNA compared to commercial reagents. Left y-axis: mRNA cost for small-scale T cell editing. Right y-axis: Total cost for all mRNA produced for this study, enabling the delivery of CRISPR editors into 66 billion human primary CAR T cells, including assay development and optimization experiments. **f**, CRISPR base editing efficiency (top: A-to-G, bottom: C-to-T) in CD4+ T cells using standard and near-PAM-less base editors for multiple genomic loci (2 to 4 donors). **g**, CRISPR base editing specificity within the editing window for two genomic loci. **h**, CRISPR activation of CD34 using mRNA-based delivery of the SAM system into CD4+ T cells (4 donors).

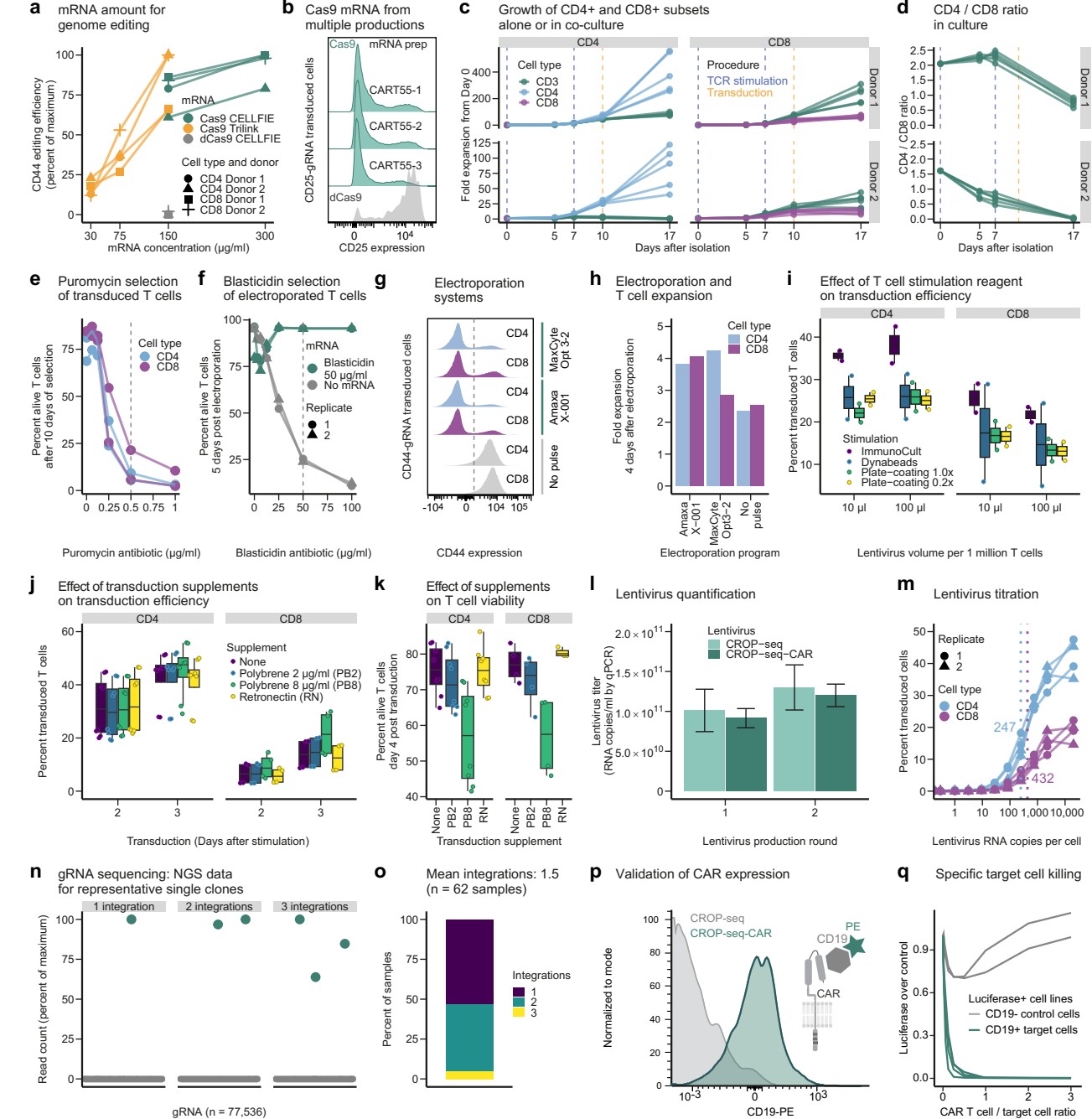

**Extended Data Fig. 2** | See next page for caption.

**Extended Data Fig. 2 | Development and optimization of the CELLFIE platform. a**, Titration of synthetic mRNA concentration for efficient CRISPR knockout of CD44 in human primary CD4[+] and CD8[+] T cells (2 donors), with pan-CD3[+] T cells as starting material. Comparable editing efficiencies were observed for custom-made and commercial Cas9 mRNA. **b**, Consistent editing efficiencies in CD4[+] T cells for custom-made Cas9 mRNA from multiple production rounds. **c**, Cell proliferation of CD4[+] and CD8[+] CAR T cells made from pan-CD3[+] T cells (green), from isolated CD4[+] T cells (blue), or from isolated CD8[+] T cells (purple) as starting material. **d**, Declining representation of CD4[+] CAR T cells in co-culture with CD8[+] CAR T cells when using pan-CD3[+] T cells as starting material. **e**, Puromycin titration to determine an optimal concentration (0.5 µg ml[−1], grey dotted line) to select for T cells that have been successfully transduced with the CROP-seq-CAR lentivirus. **f**, Blasticidin titration to determine an optimal concentration (50 µg ml[−1], grey dotted line) to select for T cells that were successfully electroporated with blasticidin resistance mRNA co-delivered with the CRISPR editor mRNA. **g**, CD44 knockout efficiency using the optimized electroporation programs for small-scale experiments (up to 1.5 million cells per cuvette, Amaxa electroporator) and large screens (up to 100 million cells per cuvette, MaxCyte electroporator). Results are shown for CD4[+] and CD8[+] T cells from pan-CD3[+] T cells (1 donor). **h**, T cell expansion after electroporation of synthetic Cas9 mRNA and blasticidin resistance mRNA (1 donor). **i**, Effect of T cell stimulation reagents on lentiviral transduction rates (4 donors). **j**, Effect of common transduction supplements on lentiviral transduction rates (4 donors). **k**, Effect of common transduction supplements on T cell viability (4 donors). **l**, Quantification of lentivirus titers using RT-qPCR of lentiviral RNA (mean ± s.e.m. for 3 technical replicates). **m**, Percent transduced CD4[+] and CD8[+] T cells for a titration of lentivirus amounts. The chosen amounts (CD4[+]: 247 copies per cell; CD8[+]: 432 copies per cell) are indicated by dotted lines (2 donors). **n**, Detection of gRNAs (y-axis: gRNA read counts) in clonally expanded human primary T cells transduced with CROP-seq-CAR lentivirus carrying the genome-wide Brunello gRNA library. Representative examples of T cell clones with 1, 2, or 3 gRNA integrations are shown, and a total of 62 clonally expanded T cell clones were profiled. **o**, Barplot showing the frequency of 1, 2, or 3 independent lentiviral integrations into the same cell across 62 clonally expanded T cell clones. The average number of gRNA integrations per cell was 1.5. **p**, CAR expression in human primary CAR T cells prepared with the CROP-seq-CAR lentivirus, using PE-labelled recombinant CD19 antigen for labeling. **q**, Specific killing of CD19[+] cancer cells by CAR T cells prepared with the CROP-seq-CAR (anti-CD19) lentivirus. For all boxplots (panels **i-l**), the center line is the median, the box limits are the upper and lower quartiles, and the whiskers extend to 1.5 times the interquartile range.

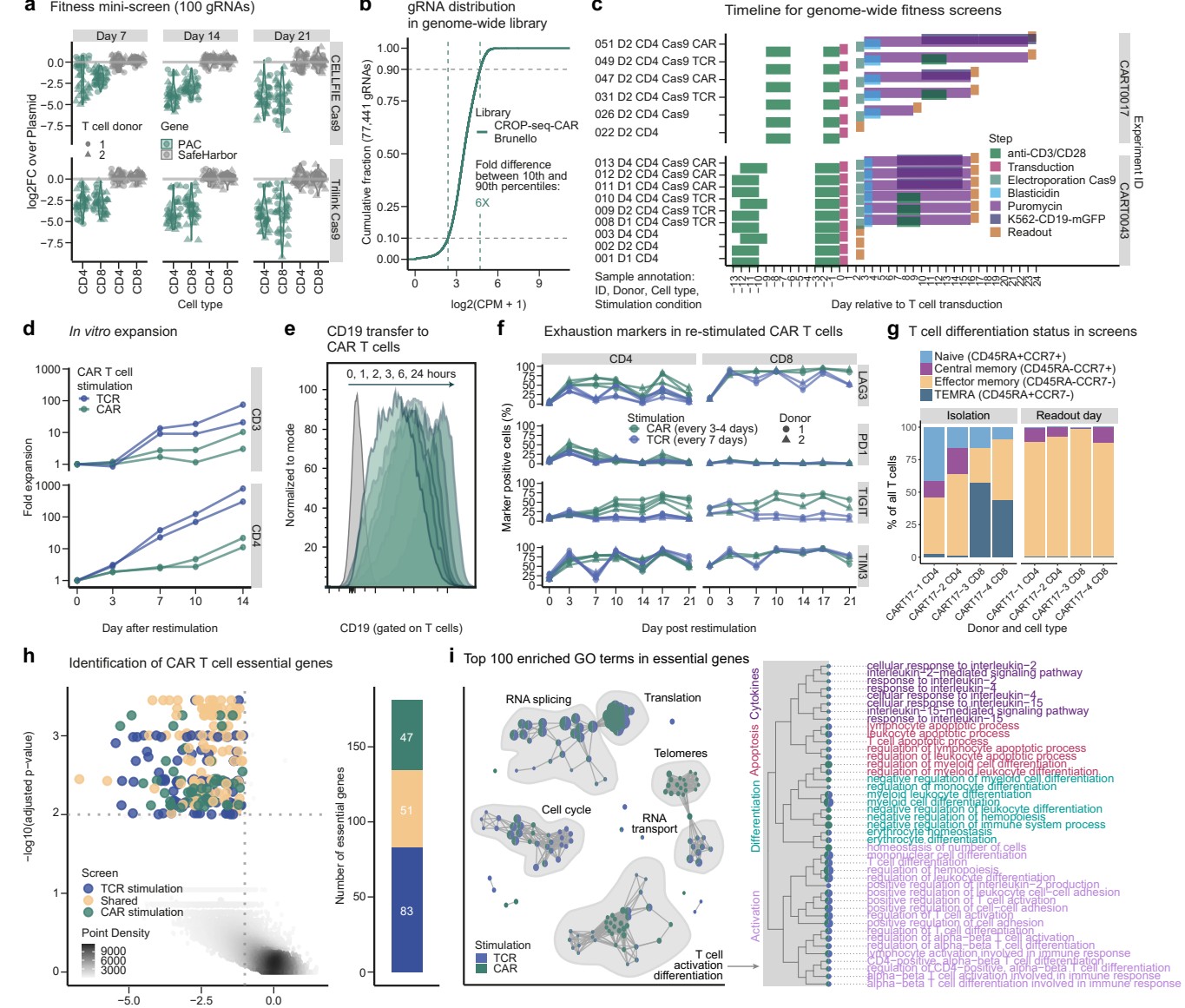

**Extended Data Fig. 3 | Optimization of genome-wide fitness screening in human primary CAR T cells. a**, Proof-of-concept fitness screen with a focused gRNA library of 100 gRNAs. gRNA-level log$_2$ fold changes are shown for positive control gRNAs (targeting the puromycin resistance gene PAC) and control gRNAs (targeting a safe harbor locus) at days 7, 14, or 21 relative to day 0. Cells were electroporated with custom-made (top) or commercial (bottom) Cas9 mRNA. **b**, gRNA representation after cloning the genome-wide Brunello library into the CROP-seq-CAR vector, plotted as a cumulative distribution function based on amplicon sequencing of the plasmid pool, with highlighted fold difference between the 10$^{th}$ and 90$^{th}$ percentiles as a measure of gRNA library balance. **c**, Detailed experimental timeline for the genome-wide fitness screens (4 donors, 2 independent experiments). Key steps in the CELLFIE workflow are highlighted. Samples for gRNA sequencing were collected at day 0 (before Cas9 electroporation), and at days 7, 14, and 21 after electroporation. **d**, Expansion of human primary CAR T cells upon repeated CAR or TCR

stimulation (2 donors). **e**, CD19 accumulation on the surface of CAR T cells during co-culture with target cells as the result of trogocytosis. **f**, Flow cytometry profiling of the T cell exhaustion markers PD1, LAG3, TIM3, and TIGIT during co-culture of CAR T cells with K562-CD19 target cells. **g**, T cell subset profiling by flow cytometry at the isolation and readout time points of the genome-wide screens. **h**, Essential genes in human primary CAR T cells under CAR and TCR stimulation (4 donors, details in Supplementary Table 2). The scatterplot shows gene-level log$_2$ fold changes comparing day 14 and day 0 of the screen (x-axis) plotted against FDR-adjusted p-values (y-axis). Genes that passed stringent significance thresholds (FDR < 0.01 and log$_2$ FC < − 1.5 based on MAGeCK RRA) are highlighted. **i**, Gene set enrichment analysis for essential genes in human primary CAR T cells under CAR and TCR stimulation. Clustering of the top-100 most enriched Gene Ontology (GO) terms from the Biological Process category (left) is shown together with one cluster related to T cell functions visualized as a tree plot (right).

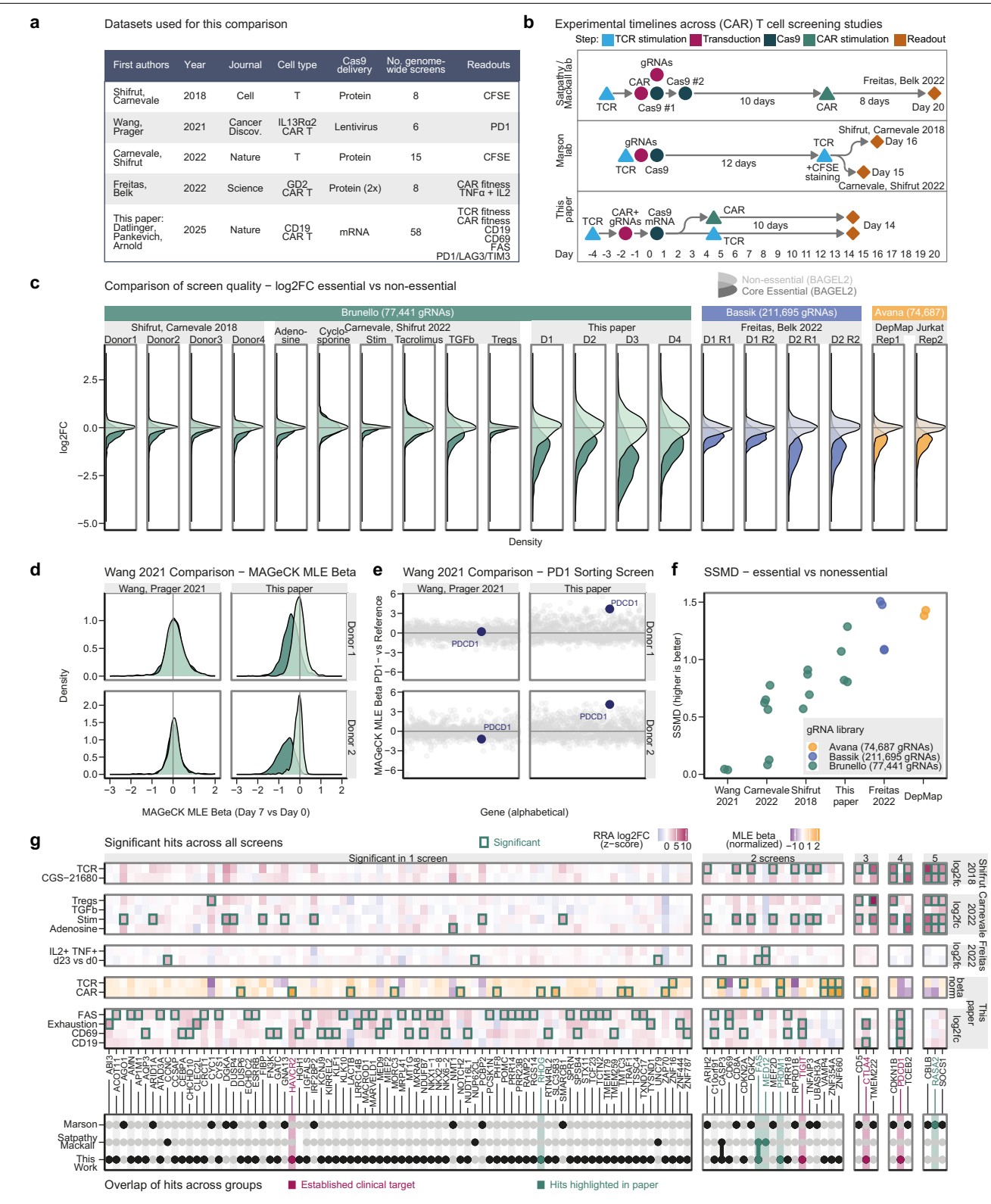

**Extended Data Fig. 4 | Comparison of data quality and gene hits between CELLFIE and published T cell or CAR T cell screening datasets. a**, Published CRISPR screens in human T cells or CAR T cells included in this comparison. **b**, Screening timelines for the studies included in this comparison. **c**, Dropout of essential genes across studies based on log₂ fold changes for known essential genes (n = 684, based on BAGEL2 predictions) relative to non-essential genes (n = 927) in genome-wide proliferation screens. **d**, Dropout of essential genes across studies based on MAGeCK MLE beta values. **e**, Comparison of *PD1* (*PDCD1*) knockout enrichment in FACS-based screens sorting for PD1-negative cells. **f**, Strength of the separation between essential and non-essential genes, quantified by the *strictly standardized mean difference* (SSMD). CRISPR screens in the Jurkat cell line from the DepMap project are included for reference. **g**, Significant screening hits across all high-quality genome-wide screens in human T cells or CAR T cells (z-normalized log₂ fold change > 2 and FDR < 0.05 for MAGeCK RRA results, normalized beta > 0.99 for MAGeCK MLE results). Genes highlighted by each study are shown in green; established clinical targets are shown in magenta.

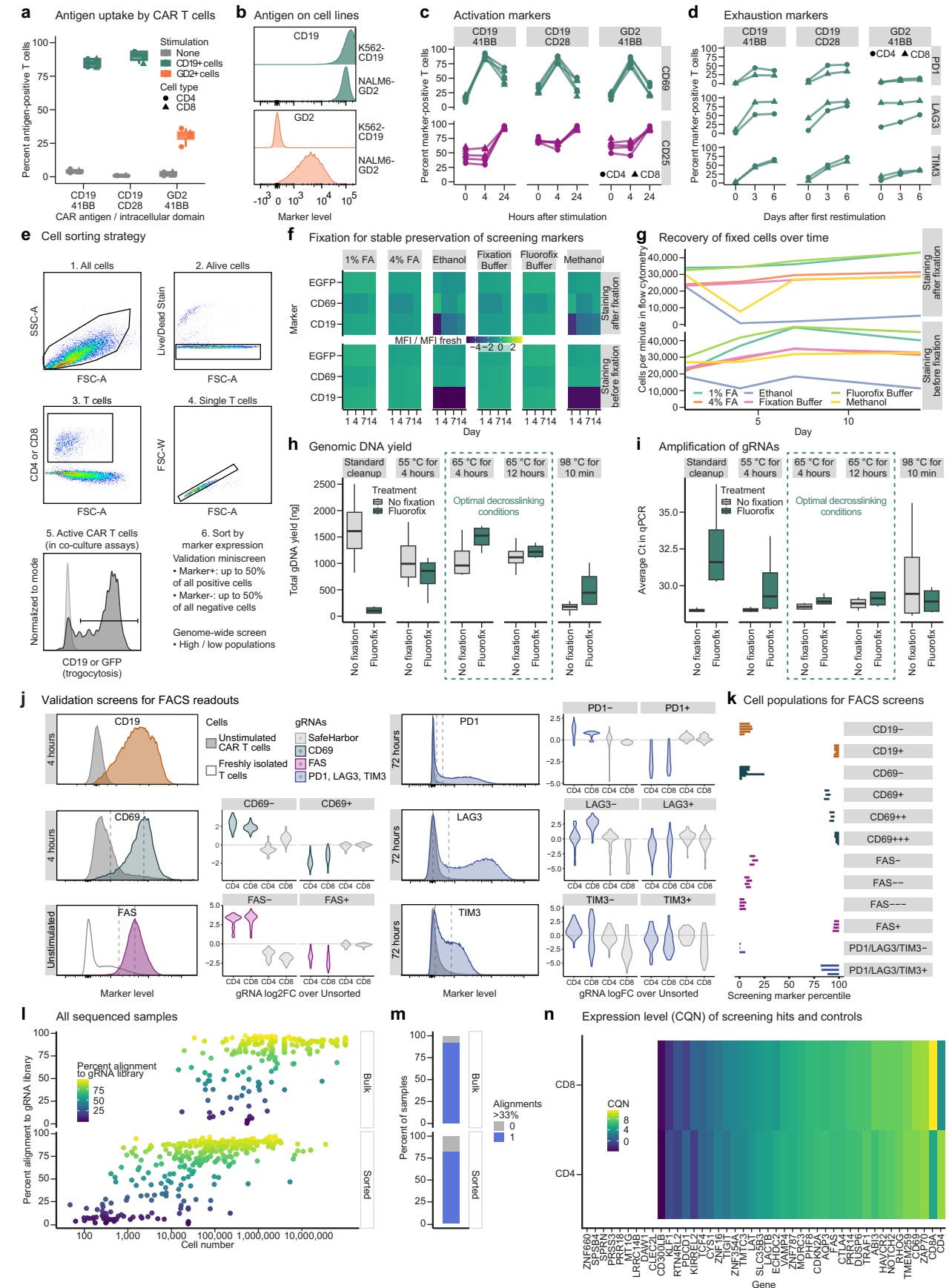

**Extended Data Fig. 5** | See next page for caption.

**Extended Data Fig. 5 | Optimization of FACS-based readouts for genome-wide screens in CAR T cells. a**, Target antigen on CAR T cells (as the result of trogocytosis) for two anti-CD19 CARs (FMC63 scFv with 41BB or CD28 co-stimulatory domains) and one anti-GD2 CAR (14g2a scFv with 41BB) following stimulation with K562-CD19 or NALM6-GD2 cells, respectively (2 donors). **b**, Antigen expression on the cell lines used for CAR T cell stimulation (K562-CD19 and NALM6-GD2). **c**, Expression of the activation markers CD69 and CD25 on CAR T cells (3 donors). **d**, Expression of the T cell exhaustion markers PD1, LAG3, and TIM3 on CAR T cells (2 donors). **e**, Overview of the FACS gating strategy. Viable T cells were selected based on physical gates, live/dead stain, and CD4 or CD8 marker expression, while excluding cell doublets. For screens involving co-culture with cancer cells, target cell engaging CAR T cells were selected based on trogocytosis-acquired CD19 (except in the CD19 screen). In the focused validation screens, cells were sorted for the top and bottom 50% of marker expression. For the genome-wide screens, cells were sorted using stringent thresholds of marker expression (panel **k**). **f**, Marker protein expression over time, comparing six methods for cell fixation with antibody staining either before or after fixation. Mean fluorescent intensities (MFI) were normalized to the staining of fresh unfixed cells. **g**, Recovery of fixed cells removed from storage at different days after cell fixation, for two alternative experimental workflows: staining after fixation (top) or staining before fixation (bottom). **h**, Genomic DNA yield from fresh and fixed cells for different de-crosslinking conditions. **i**, qPCR quantification of gRNA amplification from fresh and fixed cells for different de-crosslinking conditions. **j**, Optimization of marker protein gating and screening time points by flow cytometry (≥ 3 donors), with a representative histogram of marker-negative and marker-positive CAR T cells (left) and violin plots of log$_2$ fold change distributions (right) for gRNAs targeting the sorting markers (colored) or a safe harbor locus (grey). **k**, Sorted cell populations for all genome-wide FACS-based screens (n = 45, details in Supplementary Table 3). Within each sorted population, bars represent individual T cell donors. **l**, Scatterplot showing the percentage of reads aligning to the gRNA library as a function of the number of cells that were used as input for the gRNA amplification. Good results were obtained starting with as few as 1000 sorted cells. **m**, Barplot showing the percentage of all sequenced samples that achieved sufficient gRNA alignment rates. **n**, RNA expression of top hits from the genome-wide screens (Fig. 2d) based on RNA-seq for the CAR T cells.

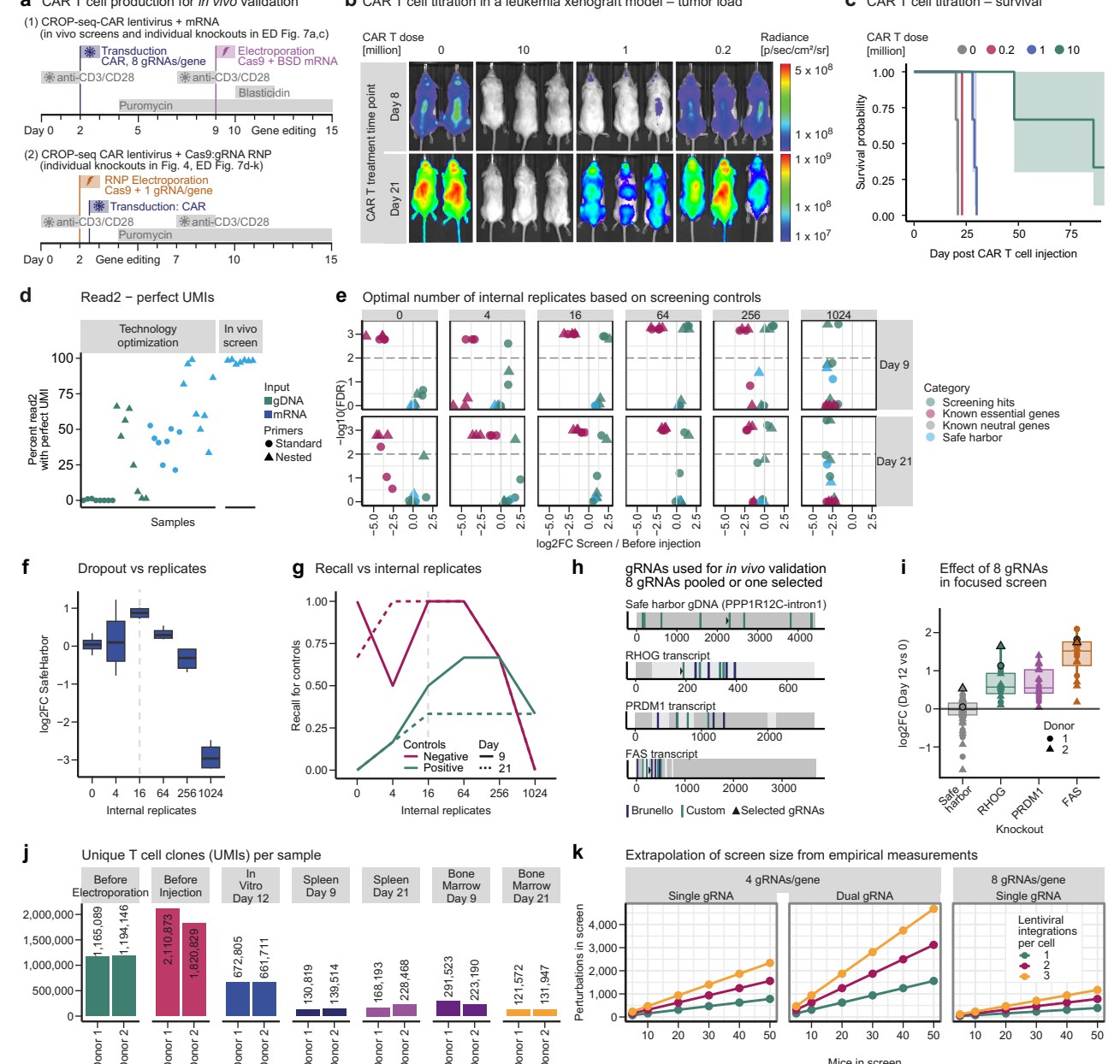

**Extended Data Fig. 6 | Leukemia xenograft model and in vivo CROP-seq optimization. a**, Experimental timeline of the in vivo validation experiments with CRISPR-boosted CAR T cells, which were genetically engineered either by lentiviral co-delivery of the CAR and a pool of 8 gRNAs followed by mRNA delivery of Cas9 (as in the in vitro screens, top) or by electroporation of a pre-assembled RNP complex of Cas9 protein and one top-performing gRNA (as is common practice in CRISPR-edited cell therapy, bottom). **b**, CAR T cell titration in a xenograft mouse model of human leukemia. Immunodeficient NSG mice were injected with 0.5 million NALM6 cells engineered to express firefly luciferase. On day 5, mice were treated with different doses of CAR T cells. Leukemic cell load was monitored using live bioluminescence imaging. When left untreated, mice succumb to the leukemia around day 21. To make the model most informative, we selected a low (and deliberately non-curative) dose of CAR T cells that leads to initial leukemic control followed by a quick relapse. **c**, Survival analysis for the mice shown in panel **b**. **d**, Percentage of UMI reads perfectly matching the reference sequence, comparing the established method (gRNA amplification from genomic DNA) with in vivo CROP-seq (gRNA amplification from mRNA), both tested with single PCR and nested PCR amplification. **e**, Optimal number of UMI-based internal replicates for data analysis based on screening controls. Given that each UMI base can be A, C, G,

or T, using UMI bases 1 to 5 results in 4, 16, 64, 256, and 1024 random internal replicates. Standard analysis is labeled as 0 internal replicates (left). Negative, neutral, and positive controls are color-coded. **f**, Dropout of neutral control gRNAs targeting a safe harbor locus when the read number in the internal replicates gets too small for large numbers of internal replicates. The box plot's center line indicates the median, the box limits represent the upper and lower quartiles, and the whiskers extend to 1.5 times the interquartile range. **g**, Recall of negative and positive controls for different numbers of internal replicates. The grey line represents the optimal number of 16 internal replicates chosen for the analysis. **h**, gRNAs selected for in vivo screens (8 gRNAs per gene) or individual validation (as pools of 8 or single gRNAs). **i**, Individual effects of the 8 gRNAs per gene in the focused validation screen, with single gRNAs used for individual validation highlighted. The box plot's center line indicates the median, the box limits are the upper and lower quartiles, and the whiskers extend to 1.5 times the interquartile range. **j**, Number of distinct T cell clones detected based on unique molecular identifiers (UMIs) in the in vivo screens. **k**, Estimation of CELLFIE's scalability to large discovery screens in vivo, extrapolating the number of screenable perturbations from empirical measurements for different screening configurations.

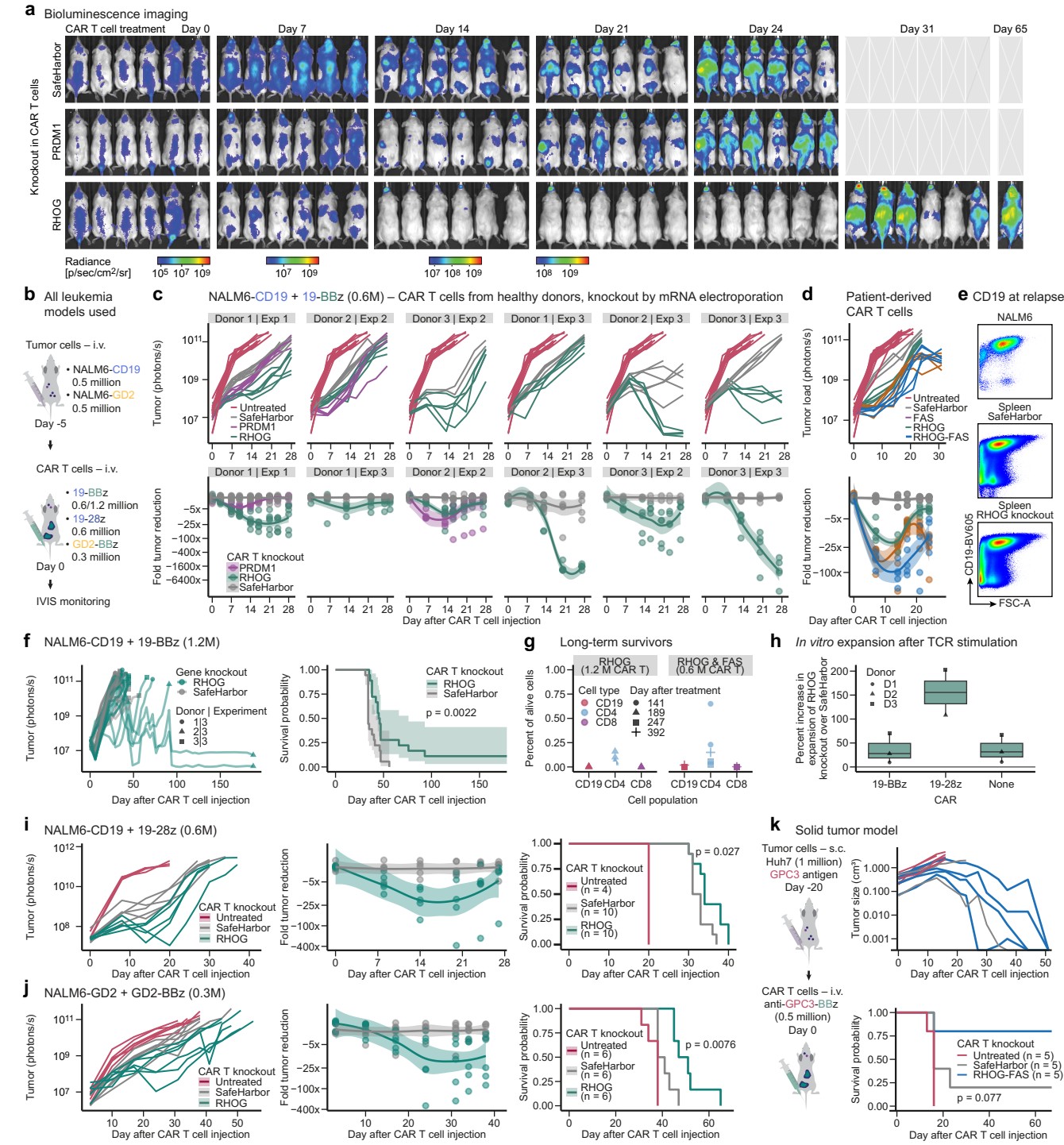

**Extended Data Fig. 7 |** See next page for caption.

**Extended Data Fig. 7 | Validation of CRISPR-boosted CAR T cells in in vivo models. a**, Bioluminescence images showing initial leukemic cell control and subsequent relapse for leukemic mice treated with standard (safe harbor locus-edited) CAR T cells or with PRDM1 or RHOG knockout CAR T cells (1 representative donor out of 9 tested). **b**, Overview of the leukemia xenograft models and CAR T cells used in this study, including 2 target antigens, 3 CAR designs, and 3 CAR T cell doses. **c**, Leukemic cell load (top: absolute values; bottom: fold reduction) for NALM6 xenograft mice treated with 0.6 million 19-BBz CAR T cells produced from six additional healthy donors (on top of those shown in Fig. 4b). **d**, Leukemic cell load (as in panel **c**) for CAR T cells produced from a leukapheresis product of one patient with multiple myeloma scheduled to receive CAR T cell therapy. **e**, CD19+ (NALM6) leukemic cells observed in mouse spleens at relapse following treatment with standard (safe harbor locus-edited) and RHOG knockout CAR T cells. NALM6 cells cultured in vitro are shown as reference. No downregulation of CD19 antigen was observed at relapse. **f**, Leukemic cell load (left) and survival analysis (right) for leukemic mice treated with a higher dose of 1.2 million 19-BBz CAR T cells (3 donors).

**g**, Percentage of CAR T cells (CD4+ or CD8+) and NALM6 (CD19+) cells in long-term survivor mice for the NALM6 xenograft model treated with RHOG knockout or RHOG-and-FAS double-knockout CAR T cells. Blood (tail bleeds) and organs (at the end point) were analyzed by flow cytometry following erythrocyte lysis. **h**, Fold increase of RHOG knockout over standard (safe harbor locus-edited) CAR T cells with two different CARs (19-BBz or 19-28z) on day 6 of expansion following restimulation with anti-CD3/CD28 5 days after Cas9 RNP electroporation (3 donors). **i**, Leukemic cell load (left: absolute values; center: fold reduction) and survival analysis (right) for NALM6 xenograft mice treated with 0.6 million 19-28z CAR T cells. **j**, Leukemic cell load (left: absolute values; center: fold reduction) and survival analysis (right) for NALM6-GD2 xenograft mice treated with 0.3 million GD2-BBz CAR T cells. **k**, Overview (left), caliper-measured tumor size (top right), and survival analysis (bottom right) for the Huh7 solid tumor xenograft model treated with 0.5 million GPC3-BBz CAR T cells. P-values for survival analysis in panels **f** and **i-k** used the log-rank test comparing mice treated with knockout versus standard (safe harbor locus-edited) CAR T cells.

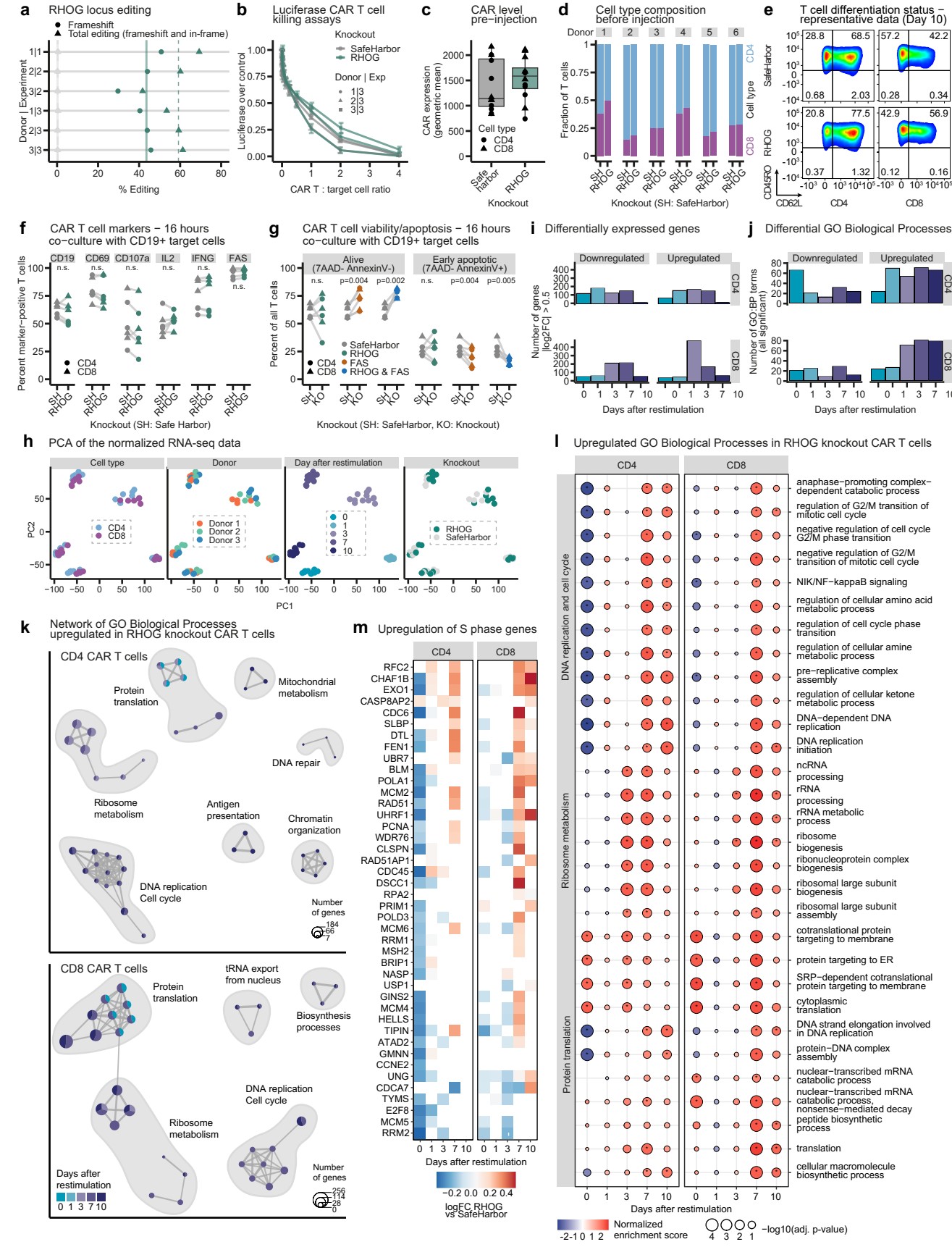

**Extended Data Fig. 8 |** See next page for caption.

**Extended Data Fig. 8 | Molecular characterization of RHOG knockout CAR T cells. a**, Percentage of successful *RHOG* editing determined by sequencing of the gene locus. **b**, Killing of luciferase-expressing NALM6 cancer cells by standard (safe harbor locus-edited) or RHOG knockout CAR T cells, measured by luciferase assays at 18 h of co-culture for different CAR T cell versus target cell ratios. **c**, CAR expression on standard (safe harbor locus-edited) CAR T cells and RHOG knockout CAR T cells, determined by flow cytometry the day before injection into mice. **d**, CD4[+] and CD8[+] cell composition among standard (safe harbor locus-edited) CAR T cells and RHOG knockout CAR T cells, determined by flow cytometry the day before injection into mice (6 donors). **e**, T cell differentiation status after 10 days of co-culture of CAR T cells and cancer cells. Representative flow cytometry data are shown (1 out of 3 donors; complete data are provided in Fig. 4f). **f**, Cell surface marker expression after 16 h of co-culture of standard (safe harbor locus-edited) CAR T cells or RHOG knockout CAR T cells with K562-CD19 target cells (3 donors). P-values used paired one-tailed t-tests. **g**, Cell viability and apoptosis marker expression after 16 h of co-culture of standard (safe harbor locus-edited) CAR T cells or RHOG knockout CAR T cells with K562-CD19 target cells (3 donors). P-values used paired one-tailed t-tests. **h**, Principal component analysis (PCA) of normalized RNA-seq data, with samples color-coded by cell type, T cell donor, time point, and RHOG knockout versus standard CAR T cells. **i**, Number of differentially expressed genes ($|\log_2FC| > 0.5$, adjusted p-value < 0.05) in RNA-seq data comparing RHOG knockout and standard CAR T cells, shown separately for CD4[+] and CD8[+] CAR T cells. **j**, Number of differential GO Biological Process terms (FDR < 0.05) in RNA-seq data comparing RHOG knockout and standard CAR T cells, shown separately for CD4[+] and CD8[+] CAR T cells. **k**, Network visualization of upregulated GO Biological Process terms in RNA-seq data comparing RHOG knockout and standard CAR T cells, shown separately for CD4[+] (top) and CD8[+] (bottom) CAR T cells. **l**, Gene set enrichment analysis of GO Biological Process terms for the three largest clusters from panel **k**. **m**, Differential expression analysis for genes associated with cell proliferation (S phase marker genes) comparing RHOG knockout and standard CAR T cells.

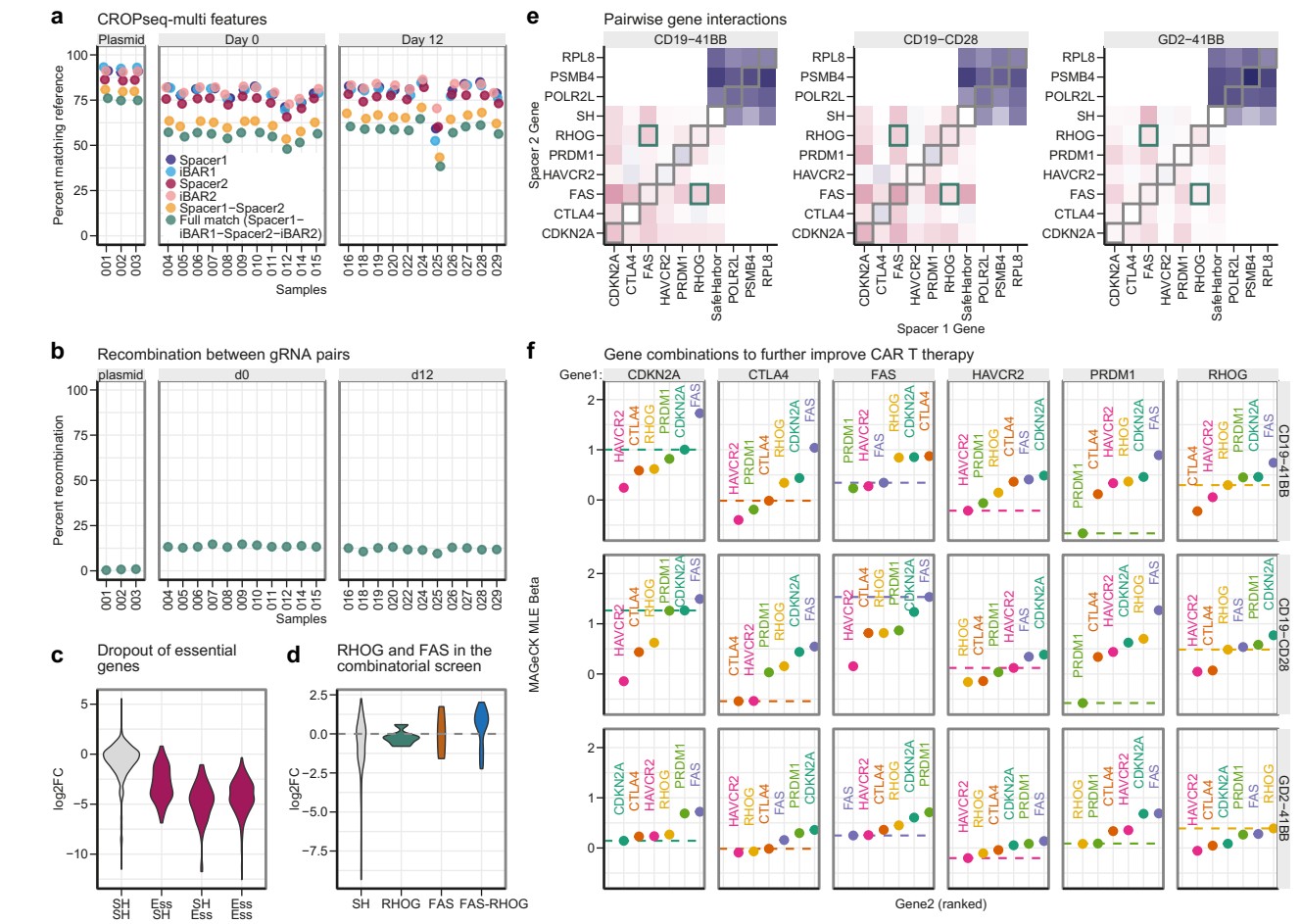

**Extended Data Fig. 9 | Optimization of combinatorial screening in CAR T cells. a**, Percentage of reads perfectly matching the full reference sequence (spacer1-iBAR1-spacer2-iBAR2) or one of its elements in the CROP-seq-CAR-multi plasmid library and at days 0 and 12 of the combinatorial screens. **b**, Percentage of reads showing evidence of recombination between the gRNA-iBAR pairs. **c**, Dropout of essential genes as determined by log₂ fold changes between day 12 and day 0, aggregated across screens with all three CARs. Results are grouped by gRNA combinations targeting only the safe harbor locus ("SH-SH"), essential genes only in position 1 ("essential-SH") or position 2 ("SH-essential"), or both positions ("essential-essential"). **d**, Log₂ fold changes between day 12 and day 0 for gRNA combinations targeting RHOG-RHOG, FAS-FAS, or RHOG-FAS, aggregated across screens with all three CARs. **e**, Heat map of fitness effects (MAGeCK MLE beta values comparing day 12 and day 0) for pairwise gene knockouts. **f**, Dot plots of fitness effects (MAGeCK MLE beta values comparing day 12 and day 0) for pairwise gene knockouts. Dotted lines indicate same-gene combinations.

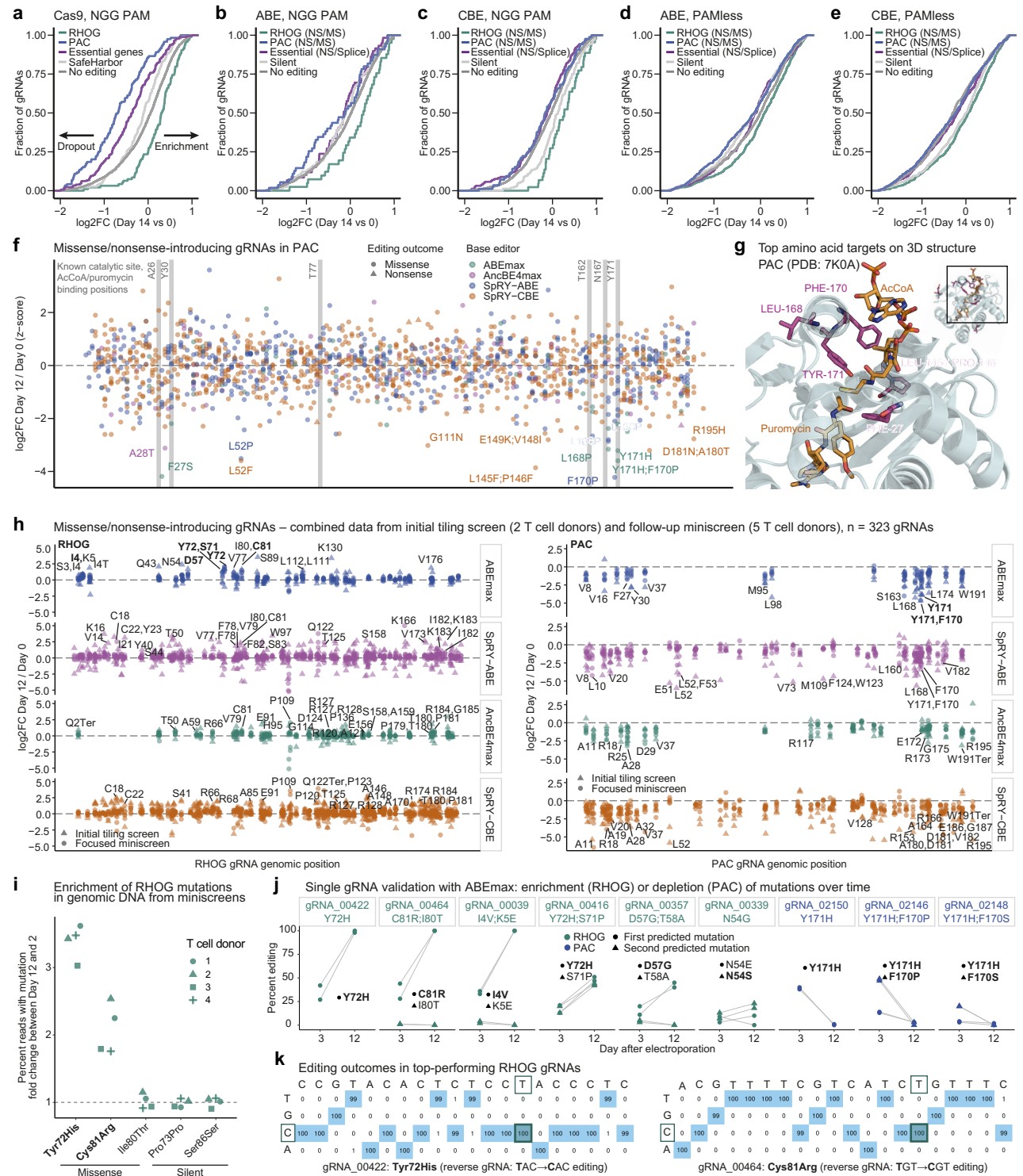

**Extended Data Fig. 10 | Optimization of base editing screening in CAR T cells and validation of single gRNAs. a-e,** Cumulative distribution plots showing enrichment and depletion of gRNAs predicted to introduce different types of mutations for Cas9 and four base editors (NS: nonsense mutation; MS: missense mutation; Splice: splice-site mutation). **f,** Mutagenesis map for the puromycin resistance gene *PAC* derived from the base editing screens (2 donors), comparing the gRNA distribution between day 12 and day 0 after CAR restimulation. Log₂ fold changes of z-scores (y-axis) are shown, and the top-15 most depleted gRNAs are labeled with the predicted amino acid changes. **g,** Amino acids with the strongest mutagenesis effects in the screen, mapped onto the PAC protein structure. The visualization is zoomed in on the PAC catalytic domain, with the full protein shown in the inset. **h,** Log₂ fold changes for 323 base editing gRNAs included in the validation screens (Supplementary

Table 8). Aggregated results are shown for gRNAs targeting *RHOG* (left) and *PAC* (right) across the original screen (2 donors) and the validation screens (5 additional donors). Top gRNAs per base editor and donor are labeled with the predicted amino acid changes. **i,** Enrichment of two *RHOG* missense mutations in CAR T cells comparing day 12 and day 2 in the focused base editing screens, as determined by next-generation sequencing of the *RHOG* locus (4 donors). **j,** Quantification of ABEmax base editing by single gRNAs targeting *RHOG* or *PAC*, analyzed by Sanger sequencing with EditR. Editing efficiency was assessed on day 3 and mutation enrichment/depletion on day 12. **k,** Detailed editing outcomes for single gRNAs targeting *RHOG*, analyzed with EditR. Highlighted in green are the base editing rates for target bases whose substitutions introduce missense mutations.

| | |
|---|---|

# Reporting Summary

## Statistics

For all statistical analyses, confirm that the following items are present in the figure legend, table legend, main text, or Methods section.

| n/a | Confirmed | |
|---|---|---|
| ☐ | ☒ | The exact sample size (*n*) for each experimental group/condition, given as a discrete number and unit of measurement |
| ☐ | ☒ | A statement on whether measurements were taken from distinct samples or whether the same sample was measured repeatedly |
| ☐ | ☒ | The statistical test(s) used AND whether they are one- or two-sided<br>*Only common tests should be described solely by name; describe more complex techniques in the Methods section.* |
| ☒ | ☐ | A description of all covariates tested |
| ☐ | ☒ | A description of any assumptions or corrections, such as tests of normality and adjustment for multiple comparisons |
| ☐ | ☒ | A full description of the statistical parameters including central tendency (e.g. means) or other basic estimates (e.g. regression coefficient) AND variation (e.g. standard deviation) or associated estimates of uncertainty (e.g. confidence intervals) |
| ☐ | ☒ | For null hypothesis testing, the test statistic (e.g. $F$, $t$, $r$) with confidence intervals, effect sizes, degrees of freedom and $P$ value noted<br>*Give P values as exact values whenever suitable.* |
| ☒ | ☐ | For Bayesian analysis, information on the choice of priors and Markov chain Monte Carlo settings |
| ☒ | ☐ | For hierarchical and complex designs, identification of the appropriate level for tests and full reporting of outcomes |
| ☒ | ☐ | Estimates of effect sizes (e.g. Cohen's *d*, Pearson's *r*), indicating how they were calculated |

*Our web collection on statistics for biologists contains articles on many of the points above.*

## Software and code

Policy information about availability of computer code

| Data collection | Sequencing: Illumina NovaSeq 6000 platform<br>Flow cytometry and cell sorting: BD LSRFortessa, Sony SH800S, BD FACSAria Fusion<br>In vivo bioluminescence imaging: IVIS Spectrum In Vivo Imaging System (PerkinElmer #124262)<br>In vitro bioluminescence imaging: Perkin Elmer Victor x3 2030 Multilabel Reader |
|---|---|
| Data analysis | The data analysis is described in detail in the Methods, including includes the following sections:<br>- Data pre-processing for the CRISPR screens<br>- Analysis of the genome-wide fitness screens<br>- Analysis of the genome-wide FACS-based screens<br>- Analysis of the in vivo CROP-seq screens<br>- RNA-seq data processing<br>- Differential expression and gene set enrichment analysis<br>- Analysis of the combinatorial screens<br>- Comparison of screening quality with published datasets<br>- Comparison of screening results with published datasets<br>- Base editing analysis<br><br>Software used: FlowJo v10.810.0, Living Image v4.7.3, Aura In Vivo Imaging Software v4.5.0; software packages used in the custom analysis code are indicated in the relevant sections of the Methods. |

For manuscripts utilizing custom algorithms or software that are central to the research but not yet described in published literature, software must be made available to editors and reviewers. We strongly encourage code deposition in a community repository (e.g. GitHub). See the Nature Portfolio guidelines for submitting code & software for further information.

## Data

Policy information about availability of data

All manuscripts must include a data availability statement. This statement should provide the following information, where applicable:
- Accession codes, unique identifiers, or web links for publicly available datasets
- A description of any restrictions on data availability
- For clinical datasets or third party data, please ensure that the statement adheres to our policy

The RNA-seq data are available from GEO (accession number: GSE266618). The CRISPR screening data are provided in Supplementary Table 2 (fitness screens), Supplementary Table 3 (FACS screens), Supplementary Table 5 (in vivo screens), Supplementary Table 7 (combinatorial screens), and Supplementary Table 8 (base editing screens).

## Research involving human participants, their data, or biological material

Policy information about studies with human participants or human data. See also policy information about sex, gender (identity/presentation), and sexual orientation and race, ethnicity and racism.

| | |
|---|---|
| Reporting on sex and gender | N/A |
| Reporting on race, ethnicity, or other socially relevant groupings | N/A |
| Population characteristics | N/A |
| Recruitment | N/A |
| Ethics oversight | N/A |

Note that full information on the approval of the study protocol must also be provided in the manuscript.

# Field-specific reporting

Please select the one below that is the best fit for your research. If you are not sure, read the appropriate sections before making your selection.

☒ Life sciences ☐ Behavioural & social sciences ☐ Ecological, evolutionary & environmental sciences

For a reference copy of the document with all sections, see nature.com/documents/nr-reporting-summary-flat.pdf

# Life sciences study design

All studies must disclose on these points even when the disclosure is negative.

| | |
|---|---|
| Sample size | Sample size was based on established standards in the field, including the number of independent biological experiments and the number of blood samples from healthy human donors for CAR T cell production (n≥3 unless otherwise indicated). For the in vivo experiments, the determination of group sizes and the power calculation have been done according to the animal license (BMWFW-2020-0.605.586). |
| Data exclusions | No data were excluded from the manuscript. |
| Replication | Fitness and FACS-based genome-wide screens were performed in at least 2 independent experiments and with least 3 independent T cell donors, as detailed in Supplementary Table 2 and 3. In vivo experiments for the validation of RHOG knockout CAR T cells were performed in 3 independent experiments using 6 independent T cells donors, as detailed in Supplementary Table 6. All experiments were replicated in at least 2 independent T cell donors, and the number is indicated. No replicates failed or had to be excluded from the analysis. |
| Randomization | Processing of samples from in vivo screening or validation experiments did not follow any particular order. Due to the uniform and fast engraftment of NALM6 cells, randomization for the NALM6 Xenograft model was not performed. To reduce biases introduced by spill-over during bioluminescence imaging, we aimed to image treatment groups together. Mouse experiments with the Huh7 solid tumor xenograft model were randomized prior to CAR T injection according to tumor size to ensure all groups had equivalent tumor load prior to treatment.<br><br>Age and sex-matched mice were used for all experiments and were maintained in a standardized location with standardized handling procedures and husbandry conditions. Experiments were performed at a consistent time of day to control for circadian effects. |
| Blinding | Investigators were not blinded to the treatment status during experiments as they also conducted the outcome assessment. Stringent, objective criteria were used for data collection and analysis to mitigate bias. |

# Reporting for specific materials, systems and methods

We require information from authors about some types of materials, experimental systems and methods used in many studies. Here, indicate whether each material, system or method listed is relevant to your study. If you are not sure if a list item applies to your research, read the appropriate section before selecting a response.

## Materials & experimental systems

| n/a | Involved in the study |
|---|---|
| ☐ | ☒ Antibodies |
| ☐ | ☒ Eukaryotic cell lines |
| ☒ | ☐ Palaeontology and archaeology |
| ☐ | ☒ Animals and other organisms |
| ☒ | ☐ Clinical data |
| ☒ | ☐ Dual use research of concern |
| ☒ | ☐ Plants |

## Methods

| n/a | Involved in the study |
|---|---|
| ☒ | ☐ ChIP-seq |
| ☐ | ☒ Flow cytometry |
| ☒ | ☐ MRI-based neuroimaging |

## Antibodies

| Antibodies used | Antibodies and other staining reagents used in this study are listed in Supplementary Table 1 and in the Methods. |
|---|---|
| | The following antibodies were used:<br>CD4 PerCP/Cyanine5.5 RPA-T4 Biolegend #300530<br>CD8 PerCP/Cyanine5.5 SK1 Biolegend #344710<br>CD19 BV605 HIB19 Biolegend #302243<br>CD19 PE-Cy7 HIB19 Biolegend #302216<br>CD69 PE-Cy7 FN50 Biolegend #310912<br>FAS (CD95) PE-Cy7 DX2 Biolegend #305622<br>PD1 PE-Cy7 EH12.2H7 Biolegend #329918<br>LAG3 PE-Cy7 11C3C65 Biolegend #369310<br>TIM3 PE-Cy7 F38-2E2 Biolegend #345014<br>TIGIT (VSTM3) PerCP-Cy 5.5 A15153G Biolegend #372717<br>CD279 (PD-1) BV605 EH12.2H7 Biolegend #329923<br>CD223 (LAG-3) AF647 11C3C65 Biolegend #369303<br>CD62L PE DREG-56 Biolegend #304806<br>CD45RO PerCP-Cy5.5 UCHL1 Biolegend #304221<br>CD107a PE-Cy7 H4A3 Biolegend #328618<br><br>Each antibody is reported by the manufacturer as being validated for specificity in flow cytometry on human samples, as described in the product data sheets and on the manufacturer's website. These antibodies are widely used and have been referenced in peer-reviewed literature for the indicated application and species. No further in-house validation was performed. |
| Validation | All antibodies were validated by the supplier. |

## Eukaryotic cell lines

Policy information about cell lines and Sex and Gender in Research

| Cell line source(s) | NALM6 and K562 cell lines were obtained from the American Type Culture Collection (ATCC), and were engineered to express transgenes as described in the Methods section "Cancer cell lines". The Huh7 cell line was a gift from the Giulio Superti-Furga Lab (CeMM). The NALM-6-GD2 cell line was a gift from the Chrystal Mackall Lab (Stanford). |
|---|---|
| Authentication | Cell line authentication was performed for the NALM6, NALM6-GD2, and K562 cell lines using Cell Line Identification service (Eurofins), which provided DNA (STR) profile results consistent with expected cells lines. The Huh7 cell line was authenticated by the Giulio Superti-Furga Lab (CeMM). |
| Mycoplasma contamination | All cell lines repeatedly tested negative for mycoplasma contamination. |
| Commonly misidentified lines<br>(See ICLAC register) | The cell lines used are not listed as commonly misidentified at the ICLAC register. |

## Animals and other research organisms

Policy information about studies involving animals; ARRIVE guidelines recommended for reporting animal research, and Sex and Gender in Research

| Laboratory animals | NOD/SCID/IL-2Rγ-null (NSG) mice were bred and maintained under specific-pathogen-free conditions at the Medical University of |
|---|---|

| Laboratory animals | Vienna. In vivo experiments were performed at the Core Facility Laboratory for Animal Breeding and Husbandry. Age-matched male or female mice (8 to 12 weeks) were used, and all experiments were performed in individually ventilated cages at ambient temperature and humidity according to the application of the Medical University of Vienna for the authorization of breeders, suppliers and users under no. BMWFW-66.009/0403-WF/V/3b/2014 at a 12 hour light/dark cycle. |
|---|---|
| Wild animals | No wild animals were used in this study. |
| Reporting on sex | Female mice were used for in vivo screens with the NALM6 xenograft model and the Huh7 solid tumor xenograft model. Male mice were used for individual validation of CAR T cells with single knockouts in the NALM6 xenograft model. |
| Field-collected samples | No field-collected samples were used in this study. |
| Ethics oversight | All experiments were performed according to the animal experiment license BMWFW-2020-0.605.586, granted by the Austrian Federal Ministry of Education, Science and Research (BMBWF) as licensing committee. All experiments were approved by the institutional ethical committee at the Department for Biomedical Research of the Medical University of Vienna and followed institutional guidelines. |

Note that full information on the approval of the study protocol must also be provided in the manuscript.

## Plants

| Seed stocks | N/A |
|---|---|
| Novel plant genotypes | N/A |
| Authentication | N/A |

## Flow Cytometry

### Plots

Confirm that:

☒ The axis labels state the marker and fluorochrome used (e.g. CD4-FITC).

☒ The axis scales are clearly visible. Include numbers along axes only for bottom left plot of group (a 'group' is an analysis of identical markers).

☒ All plots are contour plots with outliers or pseudocolor plots.

☒ A numerical value for number of cells or percentage (with statistics) is provided.

### Methodology

| Sample preparation | Cells were stained as follows: Pellet the cells, wash with 1x PBS, stain with 1/1000 Zombie Viability Dye (Biolegend) at room temperature for 10 min, wash and pellet the cells, stain with a mix of the relevant antibodies in Cell Staining Buffer (Biolegend #420201) at 4 ℃ for 30 min, wash and pellet cells, resuspend in the desired volume, and filter through a cell strainer. For storage prior to cell sorting, cells were fixed in Fluorofix Buffer (Biolegend #422101) at room temperature for 30 min in the dark, followed by washing cells twice and storing them in Cell Staining Buffer at a concentration of 50 million/ml. For large-scale screens, staining and fixation steps were done in 50 ml tubes placed on a rotator to avoid cell pelleting and clumping. The staining reagents used in this study are listed in Supplementary Table 1. |
|---|---|
| Instrument | For flow cytometry profiling, the BD LSRFortessa cell analyzer was used. For cell sorting, the Sony SH800S cell sorter was used for optimization experiments and the BD FACSAria Fusion for genome-wide screening. |
| Software | Data analysis was performed with FlowJo (Version 10.10.0) software. |
| Cell population abundance | The number of cells collected and processed in FACS-based genome-wide screens is reported in Supplementary Table 3 for each individual sample (in the range of 15,085-64,000,000 cells, depending on the sample and in particular the selected percentage of marker-positive or marker-negative cells). |
| Gating strategy | The gating strategy for FACS-based genome-wide screens is depicted in Extended Data Fig. 5e and described in Supplementary Table 3 for each individual sample. Gating was performed on all cells (using FSC-A/SSC-A axes), alive cells (live/dead stain-negative), CD4+ or CD8+ cells to select T cells, then single-cell events (using FSC-A/FSC-W axes), and then assessed the expression of individual markers used for profiling and/or cell sorting. |

☒ Tick this box to confirm that a figure exemplifying the gating strategy is provided in the Supplementary Information.

