## [Peer Review File · Nature]

Systematic discovery of CRISPR-boosted CAR T cell immunotherapies

Corresponding Author: Professor Christoph Bock

Version 0:

Reviewer comments:

Referee #1

(Remarks to the Author)

The study presents CELLFIE, an advanced CRISPR screening platform that significantly contributes to CAR T cell engineering by allowing genome-wide screens to optimize CAR T cell functions. Using an innovative delivery approach that combines CAR and CRISPR libraries and a scalable mRNA system, the authors identify RHOA, PRDM1 (BLIMP1) and FAS knockouts as crucial enhancements to CAR T cell therapy. CELLFIE allows for multifaceted screens focusing on T cell fitness, tumor recognition, and several metrics related to cell health, positioning it as a potentially foundational tool in CAR T cell optimization.

Originality and Significance

CELLFIE is a technically impressive platform that advances beyond traditional CRISPR/Cas9 approaches, offering unique advantages in scalability, combinatorial screening, and flexible mRNA delivery of CRISPR editors. However, while the CELLFIE platform introduces innovations, the manuscript could benefit from a more direct comparison to conventional CRISPR techniques to demonstrate its functional advantages fully. For instance, a combinatorial target screen could effectively highlight how CELLFIE enhances CAR T cell engineering beyond what is achievable with standard CRISPR systems. Additionally, while CELLFIE builds on previously developed CRISPR-based cell therapy strategies (e.g., PMID: 38093011, PMID: 38783148), further comparative discussion would help establish the unique contributions of CELLFIE relative to these techniques.

Data & Methodology

The approach and methodology within CELLFIE are well thought out, with rigorous experimental design and sophisticated integration of CRISPR/CROP-seq technologies for T cell engineering. The data from the screening and validation experiments are generally well-presented, though certain aspects require further clarification. The FAS knockout result, for example, is presented as beneficial in enhancing tumor control, yet this was tested in a Nalm-6 cell model that does not express FasL. An explanation or alternative model to reconcile this discrepancy would strengthen this section of the paper, providing the necessary clarity on how this knockout works within this particular model. Additionally, testing CELLFIE on only one CAR T cell design limits the generalizability of the study. Incorporating a broader range of CAR constructs would help substantiate adaptability of the platform and reinforce its claimed versatility.

Appropriate Use of Statistics and Treatment of Uncertainties

The statistical treatment in the CELLFIE platform analysis is commendable, with clear attention to rigorous significance thresholds and reproducibility in candidate selection. The use of UMIs in vivo for pooled screening is particularly impressive, providing robust control against clonal biases in CAR T cell proliferation. While these methods are well-applied, additional detail on selection biases, particularly within the pooled in vivo validation experiment, would add confidence to the findings. For instance, clarifying how the selected candidate genes were filtered and any steps taken to mitigate potential biases would enhance the robustness of the data.

Conclusions

The conclusions of the study are generally well-supported, especially regarding the potential impact of RHOA knockouts.

This knockout is an intriguing finding, as RHO G has previously been linked to immunodeficiency rather than enhanced CAR T cell function. Expanding on the molecular mechanisms by which RHO G knockouts increase CAR T cell performance could clarify its therapeutic relevance. The proposed FAS and RHO G synergy also provides an exciting therapeutic angle, but the manuscript would benefit from a more detailed exploration of the mechanisms that differentiate each knockout's contributions and any complementary pathways involved. Overall, the conclusions could be bolstered by discussing the functional roles of RHO G and a deeper explanation of the FAS and or PRDM1 and RHO G dual-knockout interaction.

Suggested Improvements

A clear comparative experiment contrasting CELLFIE with traditional CRISPR/Cas9 systems would emphasize its functional advantage, such as a multi-target combinatorial screen that illustrates CELLFIE's unique capacity for broad gene perturbations in CAR T cells.

Testing the CELLFIE approach across multiple CAR T cell designs would address the current limitation of a single CAR model, reinforcing the generalizability of the platform.

Including additional sgRNA controls outside the primary screening experiments in supplementary materials would address potential concerns about guide specificity and data robustness.

Adding model details or discussing alternative models for FasL-negative Nalm-6 cells used in FAS knockout experiments would provide the necessary context to assess the generalizability of the FAS results.

References

The authors appropriately reference prior work within the manuscript, including key background on CAR T cell limitations and prior CRISPR techniques. However, the introduction could more thoroughly frame the contributions of this new platform in light of other high-throughput CRISPR approaches to offer a clear comparative context for readers.

Clarity and Context

The abstract and introduction are technically dense and may benefit from adjustments for clarity. Simplifying the language in these sections could enhance accessibility and emphasize the primary advantages of CELLFIE to a broader audience, ensuring that the platform's clinical impact and practical value are immediately clear. Additionally, refining the flow of the introduction to highlight key limitations in CAR T cell therapy and how CELLFIE addresses these could improve readability and strengthen the overall narrative.

Referee #2

(Remarks to the Author)

In this manuscript, Datlinger, Bock, and colleagues describe CELLFIE, a platform for genome-wide screens in human CAR-T cells. CELLFIE involves several technical advances, including: (1) a new CROP-seq vector for co-delivery and readout of a CAR and a library of CRISPR sgRNAs, (2) a scalable mRNA system for low-cost delivery of CRISPR editors including both Cas9 nuclease and Cas9-derived base editors, and (3) RNA-based readout of sgRNA identify alongside a UMI to read out sgRNA abundance in the context of in vivo screens that probe the function of a small quantity of CAR cells in native mouse tissue.

Novel aspects and outcomes from their experiments include (1) systematic and large-scale screening across many conditions and with both proliferation and FACS-based phenotyping, (2) pooled in vivo screens with xenograft models in primary human T cells with both a CAR and CRISPR perturbations with proliferation-based phenotyping, (3) the discovery of knockouts such as FAS, PRDM1, and RHO G that increase the proliferation and activation of CAR T cells in vitro and in vivo and further increase the activity of CAR T cells against tumor cells in mouse xenograft tumor models, and base editor screens that implicate the GTP-binding pocket of RhoG in its CAR-function-relevant activity and suggest a path toward the abrogation of RHO G function in clinical CAR T cells without the induction of double strand breaks.

The authors are not the first to conduct genome-wide CRISPR screens in human T cells or Car T. For example, Shifrut et al (PMID 30449619) conducted genome-wide screens by the electroporation of Cas9 protein into T cells expressing lentivirally delivered CRISPR sgRNAs. Wang et al (PMID 33328215) conducted genome-wide screens by the lentiviral introduction of sgRNAs and a CAR and the electroporation of Cas9 protein into T cells. Freitas et al (PMID 36356142) conducted genome-wide screens in CAR T cells by the lentiviral introduction of sgRNAs, the electroporation of Cas9 protein, and the delivery of a CAR by a retrovirus. However, this new manuscript from Datlinger, Bock et al is distinguished by the immensity of the effort (dozens of FACS and enrichment-based screens), nice tricks for in vivo experiments, and possibly by the new targets it identifies.

Specific comments:

—The authors assert the quality of their screens, but I think they could assess screen performance with additional metrics. The metric they often use “Percent alignment to gRNA library” is important, but it does not get at the quality of biological phenotypes per se. For example, the authors would presumably get a high “Percent alignment to gRNA library” even if they did not introduce Cas9 into their samples. Standard screen metrics such as the magnitude of the difference between the phenotypes of control sgRNAs and targeting sgRNAs would be useful. The authors hint at this with their MAGECK effect sizes p-values, but I think that this could be explored further and visualized more across screens. I was surprised by the low number of common essential genes that they confidently identified as drop outs and by the low overlap between screens (Fig 1C, left and right), as these screens often have very strong phenotypes in many other CRISPR screening contexts. I was

also surprised and concerned by the relatively low separation of the core-essential from non-essential genes (Fig S3G). There may be reasons for these muted phenotypes, but the authors should discuss them in more depth.

—The authors should compare the technical details of their screens and the screen quality with previous CRISPR screens in T cells and CAR T cells. I think a supplementary figure comparing these results to prior work would be very valuable. Although this body of work may be much more substantial and important than these previous papers, that may not be immediately obvious to readers. Additionally, direct comparisons of hits and gene phenotypes to prior efforts would be very valuable.

—There is no validation of the base editing data. I think that claims as precise as “this mutation abrogates function in a clinically relevant way” are worthy of individual validation.

Referee #3

(Remarks to the Author)

This is an interesting and carefully executed study to identify boosters of CAR-T cell immunotherapy. The study features an elaborate, systematic screening platform (CELLFIE) that comprises a FACS-based fitness screen and an in vivo CROP-seq screening and nominates three gene knock outs as boosters of CAR T cell performance (FAS, PRDM1, RHOG). The authors nominate RHOG as their lead candidate that augments the fitness of CD8+ and CD4+ CAR T cell of healthy donors in pre-clinical models of cD19 CAR T cell therapy in vitro and in vivo.

The study provides an interesting technical variant compared to prior screening campaigns with CRISPR gRNA libraries, and some advantages over prior work in that screening is performed both in vitro and in vivo in mice.

However, there are also several major limitations of the study:

- CELLFIE: The CAR T cell fitness screening procedure, even though elaborate, is a 28-day tour-de-force that involves a pre-expansion, bead-activation, viral transduction, electroporation, selection with blasticidin/puromycin and expansion step. This does clearly not reflect the state of the art in T cell screenings, and is far from reality of clinical CAR T cell production (where 24 hours of culture or less is the new gold standard). It is uncertain if gene candidates that are identified in this screen may be biased through the selection of a limited number of T cell clones that survive the rigors of the screening procedure. No or insufficient data on T cell subset composition and polyclonality of the input and output populations, T cell numbers, yields and repeats of the screening process are provided.

- FACS-based screening: The parameters selected for the FACS-based screening assay are erroneous and not supported by data in this manuscript or published data in the field. CD19 expression on CAR T cells due to trogocytosis can be the result of a high affinity CAR binding domain (like FMC63) but is by no means a marker of strong target cell engagement (in a positive sense). Activation markers like CD69 should be aligned with CD25 and other activation markers like PD1, CTLA4 and TIM-3 – that only become markers of exhausted CAR T cells after sequential rounds of target engagement (but not after a single activation/expansion cycle). There are no data to support the FACS-based screening assay, e.g. with CARs that target the same antigen (CD19) with distinct affinity, or CARs that target other antigens (e.g. CD20/CD22; or GD2 with the 14D2 binding domain that induces tonic signaling and would be a relevant control).

- The in vivo screen is performed with a limited number of gRNAs only (targeting 39 genes) which reduces the technical appeal and introduces a bias to genes that were pre-selected in the FACS-based screen and then filtered for genes with known function in T cells.

- The authors perform all experiments with healthy donor T cells, where patient T cells would have been relevant. This is a major deficiency of the manuscript, and prevents relevant conclusions for CAR T cell therapy, where all approved, and the majority of investigational CAR T cell are autologous from the patient.

- The authors perform experiments with a single CAR construct directed against CD19, which is insufficient to establish a role of RHOG (as well as FAS, PRDM1) for CAR T cell immunotherapies. Validation in at least one additional hematology model (with a target other than CD19) and one additional oncology model would be required before considering such a claim.

- The study contains no relevant assessments and resolve in terms of genotoxicity and the risk for secondary malignancy that may result from FAS, PRDM1, RHOG knock out or modulation.

Version 1:

Reviewer comments:

Referee #1

(Remarks to the Author)

A. Summary of the Key Results

The authors have validated RHOG and RHOG–FAS knockout across four CAR constructs (CD19-BBz, CD19-28z, GD2-B Bz, GPC3-BBz) in both leukemic (NALM6) and solid (Huh7) models using patient-derived **Redacted** CAR T cells,

demonstrating consistent improvements in in vivo persistence and tumor clearance; they also benchmarked their 58 genome-wide screens against Shifrut 2018, Wang 2021, Carnevale 2022, Freitas 2022, and DepMap/Jurkat data, showing equal or superior separation of core essential versus nonessential genes and identifying RHOG as a novel positive regulator. Mechanistic follow-up reveals that RHOG knockout shifts CAR T cells toward a central memory phenotype with reduced exhaustion, and FAS knockout mitigates activation-induced apoptosis, while a new CROPseq-multi dual-gRNA system enabled combinatorial screening of 238 gene pairs that confirmed RHOG–FAS synergy in vitro and produced durable cures in vivo; a base-editing tiling screen in RHOG identified key GTP-binding pocket residues whose editing phenocopies knockout.

B. Originality and Significance

Introducing pooled dual-gRNA combinatorial screening in primary human CAR T cells via CROPseq-multi is a clear advance over prior work that relied on immortalized lines or single-guide approaches, and the finding that RHOG (historically linked to immunodeficiency) enhances CAR T proliferation and persistence is highly novel; validating RHOG and RHOG–FAS knockout benefits across four CAR backbones in both hematologic and solid tumor settings using patient-derived **Redacted** CAR T cells underscores broad translational impact. To strengthen claims of novelty, the authors could briefly compare CROPseq-multi with Medicorico et al. (Cell 175, 258–270, 2023) and note in the Discussion that CELLFIE supports multiple CRISPR modalities (knockout, base editing, CRISPRa, CRISPRi) alongside combined phenotypic and in vivo readouts.

C. Data & Methodology

The Methods are now organized into clear subsections and include quality-control metrics (library alignment rates, zero-count guide percentages, Gini indices, SSMD values) that confirm screen quality matches or exceeds prior studies; the nested PCR protocol for in vivo CROP-seq is well described, and the CROPseq-multi cloning workflow is detailed sufficiently to allow replication, while in vivo validations use at least five mice per arm with bioluminescence imaging and log-rank analysis. To enhance reproducibility, the authors should state the exact n for each mouse cohort, clarify whether guides with fewer than twenty UMIs were excluded from analyses, and indicate the average number of unique CAR T cell UMIs recovered per sample.

D. Appropriate Use of Statistics and Treatment of Uncertainties

Statistical methods are applied consistently: genome-wide screens use MAGeCK MLE with stringent thresholds, UMI subsampling reduces PCR bias, survival analyses employ log-rank tests with hazard ratios and confidence intervals, and flow cytometry data use t tests or ANOVA with exact p values, while confidence intervals and interaction scores are appropriately shown in the base-editing tiling and combinatorial assays. To improve interpretability, the SSMD axis in Figure S4f should be annotated with thresholds (for example, $SSMD \geq 2$ indicates excellent separation) and the text should note exclusion criteria for low-coverage guides to emphasize how stochastic noise was minimized.

E. Conclusions: Robustness, Validity, Reliability

The data convincingly show that the benefits of RHOG knockouts are robust across four CAR backbones, two tumor types, and two CAR T sources (patient-derived and Kymriah), with consistent improvements in persistence, tumor clearance, and survival; RHOG–FAS synergy is validated both in vitro and in vivo, yielding durable cures in a model that otherwise fails to achieve long-term control. The authors should note that the observed central memory shift (roughly a 10-15% increase in CD62L⁺CD45RO⁺ cells) is modest rather than a complete reprogramming and briefly mention any long-term assessment of off-target effects in non-CAR immune subsets to reassure readers that RHOG–FAS double knockout does not produce unintended deleterious consequences.

F. Suggested Improvements: Experiments, Data for Possible Revision

Although the revision is strong, the Methods should include a brief description of any in silico off-target predictions (such as GUIDES or Cas-OFFinder) for key RHOG and FAS guide RNAs; the Discussion should acknowledge that the impact of RHOG on metabolic fitness (glycolysis versus oxidative phosphorylation) remains untested, clarifying it as a future direction rather than a limitation; in Figure 6d, highlighting the top five to ten synergistic pairs beyond RHOG–FAS (e.g., PRDM1–NR4A1) with outlines or arrows and listing them in the legend would guide readers to additional candidates; and including a schematic in the main text or Supplementary that details coverage requirements (cells per guide, total cells per mouse) and projected animal numbers for a 1,500-gene screen would help others plan large-scale experiments.

G. References: Appropriate Credit to Previous Work?

The manuscript cites key prior studies (Shifrut 2018, Wang 2021, Carnevale 2022, Freitas 2022, Schmidt 2024, and Walsh 2025) and explicitly compares the performance of CELLFIE to those platforms, ensuring proper credit; as a minor addition, citing Medicorico et al. (Cell 175, 258–270, 2023), which used single-vector CROP-seq in primary T cells, would acknowledge parallel advances. No other significant omissions of prior work were identified.

H. Clarity and Context: Lucidity of Abstract/Summary, Appropriateness of Abstract, Introduction, and Conclusions

The Abstract is concise and introduces technical terms only after summarizing major findings, and the Introduction clearly

frames clinical limitations in CAR T therapy before reviewing prior genetic engineering efforts and presenting CELLFIE's features in logical sequence; figures and legends are generally clear with consistent labeling, and in Figure 2e the legend specifies that immunophenotypic signatures derive from fifty-eight screens and defines "shared across several screens" as enrichment in at least two datasets. To improve clarity, the Methods should specify that CAR versus TCR stimulation readouts were measured on day 7 post-electroporation and note whether error bars represent \pm SD or \pm SEM as indicated in each legend, while the Discussion should acknowledge limitations (such as the modest central memory shift) and outline future directions (e.g., metabolic profiling) without overstating the current results.

Referee #2

(Remarks to the Author)

The authors have provided a comprehensive response to my review and have completed several new analyses and experiments that address my comments and the comments from the other reviewers.

My comments focused primarily on screening data quality, which I felt was incompletely explored in the initial manuscript, especially in comparison with previous work. The various analyses the authors have conducted make it much easier for readers to assess this work. Overall, I am convinced that the Cellfie platform enables in vitro screens that are comparable or modestly better than the best examples of previous work. I appreciate the transparency in the methods for ensuring fair comparisons across studies.

Regarding the comparison of hits: The analysis showing limited overlap between studies is illuminating and supports your point about the value of screening under diverse conditions. The identification of established clinical targets (PD1, CTLA4, TIGIT, TIM3) uniquely by CELLFIE further validates the screening approach.

Regarding base editing validation: The additional validation experiments, including focused screens, targeted NGS, and functional confirmation of specific mutations, adequately address my concerns about the base editing data, especially in conjunction with the more cautious language regarding clinical relevance. I do overall think that this section remains under-developed, but I am convinced that the Cellfie platform is useful for diverse screening modalities.

The addition of combinatorial screening capability is a valuable enhancement to the platform.

Overall, the revised manuscript represents a substantial improvement and addresses my technical concerns in a careful and systematic manner.

Referee #3

(Remarks to the Author)

I have carefully reviewed the revised manuscript and author response and can state that the authors have comprehensively and satisfactorily addressed the critique from the first round of review.

The only one aspect that ought to be covered at least in the Discussion is the risk of genotoxicity and inducing or favoring malignant transformation of CAR T cells that carry either gene-edits or overexpressed genes identified through the CELLFIE platform.

In particular in light of recent studies e.g. by Braun et al. Nat Med 2025 PMID: 39984633 that highlight a role of pre-existing clonal hematopoiesis (CHIP) and inflammatory response during CAR T cell treatment, that may favor clonal evolution and branching with emergence of T cell lymphoma.

The introduction of gene-edits or overexpression of genes that are directed at increasing proliferation and persistence of CAR T cells, certainly increases the likelihood for such undesired events to occur and mandates careful pre-clinical assessment and clinical monitoring.

Nature
One New York Plaza, Suite 4500
New York NY 10004-1562
USA

Univ.-Prof. Dr. Christoph Bock

e-mail: cbock@cemm.oeaw.ac.at
phone: +43-140-160-70070
web: <https://www.bocklab.org>

Vienna, 15 May 2025

Dear Editor,

Thank you for inviting us to submit a revised version of our manuscript describing the systematic enhancement of CAR T cell immunotherapies using CRISPR screens. We greatly appreciate the reviewers' constructive comments, which we addressed through extensive new experiments and data analyses. Our revised manuscript includes 41 new figure panels and 22 new tables, with results that substantially strengthen the conclusions and impact of our study.

In the following paragraphs, we summarize the main points raised by the reviewers and explain how each was addressed through additional experiments and analyses. Further below, we provide a tabular overview of our major additions in the revised manuscript, and a detailed point-to-point response to all reviewer comments.

First, Reviewers 1 and 3 suggested testing RHOG and RHOG-FAS knockouts in additional contexts. In a series of new *in vitro* and *in vivo* experiments (Fig. S8), we validated the beneficial effect of these knockouts in combination with four CARs with different antigen recognition (CD19, GD2, GPC3) and costimulatory domains (41BB, CD28), in mouse models of two human cancers (NALM6 / leukemia; Huh7 / hepatocellular carcinoma). Moreover, we confirmed the beneficial effect of the knockouts in patient-derived CAR T cells **Redacted**.

Second, Reviewer 1 and 2 asked us to compare the CELLFIE platform to previous CRISPR screening efforts in human (CAR) T cells. We addressed this point by conducting a systematic quantitative comparison of our genome-wide CRISPR screens with the datasets of related studies (new Fig. S4 and Table S4). This analysis confirmed the excellent performance of CELLFIE. We also expanded our discussion of the key features of the platform and more clearly explain how they advance the current state of the art of CRISPR screening in (CAR) T cells *in vitro* and *in vivo*.

Third, Reviewer 1 encouraged further characterization of the biological effects of the RHOG and FAS knockouts. In response, we performed a series of new functional experiments showing complementary effects (Fig. S9): RHOG knockout enhances proliferative capacity, while FAS knockout promotes survival and reduces fratricide among CAR T cells.

Fourth, following a suggestion by Reviewer 1, we established combinatorial CRISPR knockout screening in CAR T cells as a new feature of the CELLFIE platform (new Fig. 6a-d, Fig. S10 and Table S8), and we performed a combinatorial screen to assay all pairs of CAR T boosters identified by our screens (238 knockout pairs including controls). This new and distinguishing feature underlines the versatility of CELLFIE and adds to its utility for genetic optimization of CAR T cells.

Fifth, we clarified key aspects of our study by describing the selection of the top screening hits from our genome-wide screens, by providing additional explanations and experimental data in support of our FACS-based readouts, and by adding further quality metrics and documentation for the genome-wide screens. Finally, we conducted a focused validation screen and single-gRNA validation experiments to confirm the hits of our tiling base editing screen of RHOG.

In summary, our revised manuscript further substantiates the quality and versatility of the CELLFIE platform, and it provides additional validation for RHOG and RHOG-FAS knockouts as broadly effective enhancers of CAR T performance.

Best regards,
Christoph Bock

Key experiments and analyses introduced in the revised manuscript:

Topics addressed	New data in the revised manuscript	New figures & tables
Comprehensive validation of RHOG and RHOG-FAS knockouts in vivo and with patient-derived CAR T cells	We extended our in vivo validation experiments to cover four different CAR constructs (CD19-BBz, CD19-28z, GD2-BBz, GPC3-BBz) and both leukemia (NALM6 xenograft) and a solid tumor (hepatocellular carcinoma using Huh7 xenograft). To strengthen the translational relevance, we validated in patient-derived CAR T cells (leukapheresis material of a patient scheduled to receive CAR T cell therapy) Redacted. These experiments further support the broad applicability of the identified knockouts across CAR architectures, tumor types, and clinically relevant T cell sources. Moreover, long-term follow-up of surviving mice revealed no signs of malignant transformation of the knockout CAR T cells.	Fig. S8g-p Table S6 Reviewer Table 1
Combinatorial knockout screening adds a new capability to the CELLFIE platform	We developed dual-gRNA vectors and implemented combinatorial knockout screening, which we applied across three different CAR designs. This modular approach once again identified the RHOG-FAS combination and further uncovered novel gene pairs that enhance CAR T cell proliferation.	Fig. 6a-d Fig. S10 Table S1j, S8
Complementary effects of RHOG and FAS knockout	New functional experiments showed that RHOG and FAS knockouts enhance CAR T cells through different and complementary mechanisms. We also confirmed that FasL is expressed on activated CAR T cells but not on NALM6 cells, suggesting a role for fratricide in the FAS knockout phenotype.	Fig. S9d,h-i Reviewer Fig. 1
Benchmarking against published screens	We benchmarked against published genome-scale CRISPR screens in human primary T cells and CAR T cells, including studies by Shifrut et al. 2018, Wang et al. 2021, Carnevale et al. 2022, and Freitas et al. 2022, as well as Jurkat cell line screens from the Broad Institute's DepMap project. This analysis quantified the technical quality of the screens and compared the genes identified in each study. Our results support the strong technical performance and the potential for enhancing CAR T cells with the CELLFIE platform.	Fig. S4 Table S4 Reviewer Fig. 2
Validation of the RHOG base editing tiling screen	Focused base editing screens and individual gRNA testing validated that specific missense mutations in RHOG confer a proliferative advantage to CAR T cells, phenocopying the effect of the RHOG knockout.	Fig. S11h-j Table S9e-g

Point-by-point response to the reviewer comments

Reviewer #1

The study presents CELLFIE, an advanced CRISPR screening platform that significantly contributes to CAR T cell engineering by allowing genome-wide screens to optimize CAR T cell functions. Using an innovative delivery approach that combines CAR and CRISPR libraries and a scalable mRNA system, the authors identify RHOG, PRDM1 (BLIMP1) and FAS knockouts as crucial enhancements to CAR T cell therapy. CELLFIE allows for multifaceted screens focusing on T cell fitness, tumor recognition, and several metrics related to cell health, positioning it as a potentially foundational tool in CAR T cell optimization.

We thank the reviewer for the positive overall assessment and for the constructive comments and suggestions, which have substantially improved the manuscript. We are especially grateful for the suggestion to establish combinatorial CRISPR knockout screens as an important new feature of the CELLFIE platform and a fitting addition to this study.

Key additions and improvements in the revised manuscript include:

- We established pooled combinatorial CRISPR screens in CAR T cells, using new dual-gRNA vectors inspired by the CROPseq-multi design (Walton et al., 2024). Our combinatorial screen further validated the RHOG-FAS double knockout across several CARs and identified additional knockout pairs that enhance the proliferative capacity of human primary CAR T cells (Fig. 6a-d, Fig. S10, Table S8).
- We quantitatively compared the technical performance and the screening hits across all published genome-wide CRISPR knockout screens in human primary T cells and CAR T cells (Fig. S4, Table S4).
- We updated our discussion of the CELLFIE platform in the context of previously published studies, emphasizing the scalability and versatility of our method with demonstrated support for high-quality *in vivo* screens, base editing tiling screens, and combinatorial knockout screens.
- We investigated the biological characteristics of RHOG and FAS knockout CAR T cells with additional functional experiments, providing evidence that these knockouts have different and complementary effects (Fig. S9).
- We clarified that the beneficial effect of FAS knockout despite FasL not being expressed on NALM6 cells is due to FasL expression on activated (CAR) T cells.
- We validated the beneficial effect of RHOG and FAS knockouts on CAR T cell performance for four CAR designs and in three xenograft models (two leukemia models and one solid tumor model), and we confirmed the findings in patient apheresis material (Fig. S8) **Redacted**.

Originality and Significance

CELLFIE is a technically impressive platform that advances beyond traditional CRISPR/Cas9 approaches, offering unique advantages in scalability, combinatorial screening, and flexible mRNA delivery of CRISPR editors. However, while the CELLFIE platform introduces innovations, the manuscript could benefit from a more direct comparison to conventional CRISPR techniques to demonstrate its functional advantages fully. For instance, a combinatorial target screen could effectively highlight how CELLFIE enhances CAR T cell engineering beyond what is achievable with standard CRISPR systems. Additionally, while CELLFIE builds on previously developed CRISPR-based cell therapy strategies (e.g., PMID: 38093011, PMID: 38783148), further comparative discussion would help establish the unique contributions of CELLFIE relative to these techniques.

We appreciate the reviewer's positive feedback and helpful suggestions. In the revised version, we describe in more detail how CELLFIE differs from other approaches, including the papers referred to by the reviewer (which we cite). We particularly highlight CELLFIE's scale and versatility as demonstrated by genome-wide screens with multiple readouts, *in vivo* screens with a novel CROP-seq-based clonal tracking method, and tiling base editing screens to functionally dissect our top hit ("biochemistry by sequencing") and to prioritize base editing gRNAs for clinical translation.

Following the reviewer's suggestion, we have now extended CELLFIE to support combinatorial CRISPR knockout screens in primary CAR T cells. To that end, we adapted our vector system to express dual gRNAs using the elegant CROPseq-multi design (Walton et al., 2024), in which gRNAs are separated by a tRNA spacer that is cleaved by the cells. Using this

approach, we conducted pooled combinatorial screens covering 238 pairwise combinations of our top CAR T cell boosters across three CAR constructs (CD19-BBz, CD19-28z, GD2-BBz). These experiments confirmed the beneficial effect of the RHOG-FAS double knockout and identified several additional gene knockout pairs that enhance CAR T cell proliferation, thus establishing combinatorial screens as a new and distinguishing feature of the CELLFIE platform. The data for the combinatorial screens resulted in new figures (Fig. 6a-d, Fig. S10) and a table that summarizes the library, screen design, raw data, quality control, and MAGeCK MLE results (Table S8a-f).

We also updated the manuscript to compare CELLFIE to the two recently published studies mentioned by the reviewer: Schmidt, Ward et al. 2024 and Walsh, Shah et al. 2025. Schmidt et al. used two-step lentiviral delivery for the base editor and the sgRNA, which lacks the flexibility of an mRNA-based platform. Walsh et al., like CELLFIE, used electroporation of base editor mRNA produced by in-house IVT. However, CELLFIE is distinguished by optimized mRNA production for multiple editing modalities, support for pooled *in vivo* screening with UMI-based clonal tracing, and the ability to perform combinatorial knockout screens – capabilities not demonstrated in either of those two studies.

Finally, to benchmark CELLFIE more broadly, we also included a comparative analysis of genome-wide screen quality and T cell boosters across all published CRISPR screens in human primary T cells and CAR T cells (Fig. S4). Our dataset of 58 genome-scale screens across six core biological processes exceeds the scale of all prior studies combined (Fig. S4a).

Data & Methodology

The approach and methodology within CELLFIE are well thought out, with rigorous experimental design and sophisticated integration of CRISPR/CROP-seq technologies for T cell engineering. The data from the screening and validation experiments are generally well-presented, though certain aspects require further clarification. The FAS knockout result, for example, is presented as beneficial in enhancing tumor control, yet this was tested in a Nalm-6 cell model that does not express FasL. An explanation or alternative model to reconcile this discrepancy would strengthen this section of the paper, providing the necessary clarity on how this knockout works within this particular model. Additionally, testing CELLFIE on only one CAR T cell design limits the generalizability of the study. Incorporating a broader range of CAR constructs would help substantiate adaptability of the platform and reinforce its claimed versatility.

We thank the reviewer for the positive overall assessment of our data and methodology. We respond to the two specific points raised (FasL expression, one CAR T cell design) in detail in the following paragraphs.

Appropriate Use of Statistics and Treatment of Uncertainties

The statistical treatment in the CELLFIE platform analysis is commendable, with clear attention to rigorous significance thresholds and reproducibility in candidate selection. The use of UMIs *in vivo* for pooled screening is particularly impressive, providing robust control against clonal biases in CAR T cell proliferation. While these methods are well-applied, additional detail on selection biases, particularly within the pooled *in vivo* validation experiment, would add confidence to the findings. For instance, clarifying how the selected candidate genes were filtered and any steps taken to mitigate potential biases would enhance the robustness of the data.

We thank the reviewer for the positive assessment of our analytical framework. We indeed put a lot of effort into rigorous statistical methods and highly reproducible screens, and we are convinced that our CROP-seq based *in vivo* screening method with UMI support for clonal tracking will be broadly useful beyond screens in primary CAR T cells.

We also appreciate the suggestion to describe our hit prioritization process in more detail. The selection was data-driven and was based on stringent statistical thresholds for hits from genome-wide screens (normalized MAGeCK MLE $\beta > 0.99$ for fitness screens and $\log_2FC > 2$, $FDR < 0.05$ for FACS-based screens), filtering for genes expressed in CAR T cells based on RNA-seq data (Fig. S6e, Table S7). We have clarified this both in the Methods section (line 1056-1058) and in the corresponding Results section (line 249-252).

Conclusions

The conclusions of the study are generally well-supported, especially regarding the potential impact of RHOG knockouts. This knockout is an intriguing finding, as RHOG has previously been linked to immunodeficiency rather than enhanced CAR T cell function. Expanding on the molecular mechanisms by which RHOG knockouts increase CAR T cell performance could clarify its therapeutic relevance. The proposed FAS and RHOG synergy also provides an exciting therapeutic angle, but the manuscript would benefit from a more detailed exploration of the mechanisms that differentiate each knockout's contributions and any complementary pathways involved. Overall, the conclusions could be bolstered by discussing the functional roles of RHOG and a deeper explanation of the FAS and or PRDM1 and RHOG dual-knockout interaction.

It is great to hear that the reviewer describes the conclusions of our study as generally well-supported and providing an exciting therapeutic angle. To further substantiate our conclusions, in the revised manuscript we added validations of the beneficial effects of the RHOG and RHOG-FAS knockouts across additional CARs, **Redacted**. Moreover, in response to the reviewer's point about the functional roles of the RHOG and FAS knockouts, we performed new experiments (Fig. S9d,h,i) and expanded the corresponding paragraphs in the Results section (lines 356-364). Our data support the following conclusions:

- *RHOG knockout enhances CAR T cell proliferation.* This effect coincided with an increased fraction of memory-like cells, elevated DNA replication, and reduced expression of exhaustion markers (Fig. 5, S9).
- *RHOG knockout does not lead to increased cytotoxicity (Fig. S9d), activation (CD69), degranulation (CD107a), or cytokine production (IL2, IFN γ) (Fig. S9h), nor does it reduce apoptosis (Fig. S9i).* Collectively, our data support that the RHOG knockout improves proliferative fitness rather than enhancing effector function or cell survival.
- *FAS knockout, by contrast, enhances CAR T cell survival and persistence.* We experimentally confirmed this by observing reduced apoptosis in functional assays (Fig. S9i).
- *The positive effects of RHOG and FAS knockouts appear mechanistically distinct and complementary.* RHOG promotes expansion, while FAS enhances survival. The RHOG-FAS double knockout CAR T cells profit from the combination of both effects, which explains their enhanced performance over RHOG and FAS single knockouts.

These findings support a modular approach to CAR T cell optimization where several genetic interventions are combined to target different and complementary aspects of CAR T cell function (here: RHOG knockout provides enhanced proliferation and FAS knockout provides enhanced survival). Our CELLFIE platform – especially with the newly added feature of combinatorial screens – provides a powerful tool to systematically identify such modular combinations of knockouts.

Suggested Improvements

A clear comparative experiment contrasting CELLFIE with traditional CRISPR/Cas9 systems would emphasize its functional advantage, such as a multi-target combinatorial screen that illustrates CELLFIE's unique capacity for broad gene perturbations in CAR T cells.

We agree that a multi-target combinatorial screen is a great way to illustrate CELLFIE's unique capabilities, and we took on the challenge to implement it in this revision. As described above, we adapted our system for dual gRNA expression and conducted pooled combinatorial knockout screens across 238 gene pairs, further confirming the RHOG-FAS double knockout and identifying novel gene knockout pairs that enhance CAR T cell proliferation (Fig. 6a-d, Fig. S10, Table S8). These data demonstrate and validate combinatorial screening as a powerful new capability of the CELLFIE platform.

Testing the CELLFIE approach across multiple CAR T cell designs would address the current limitation of a single CAR model, reinforcing the generalizability of the platform.

Testing the CELLFIE approach across multiple CAR T cell designs is indeed important for two reasons: to demonstrate that the platform and its readouts are generalizable, and to validate that the identified CAR T cell boosters perform robustly in different therapeutic contexts. To address both aspects in this revision, we incorporated three additional CAR constructs with distinct antigen specificities and costimulatory domains (CD19-28z, GD2-BBz, GPC3-BBz). First, we validated our FACS-based readouts with additional CARs (Fig. S5a-d). Second, we performed our combinatorial knockout

screen across three CARs (CD19-BBz, CD19-28z, GD2-BBz), demonstrating that CELLFIE screens transfer well between different CARs (Fig. 6a-d). Third, we extended our *in vivo* validation of key screening hits (including RHOG and RHOG-FAS knockouts) to the three new CAR designs, covering both leukemia (CD19, GD2 on NALM6) and solid tumor (GPC3 on Huh7) models, and using both healthy donor-derived and patient-derived CAR T cells (Reviewer Table 1). These new data confirmed the broad utility of the CELLFIE platform and the functional benefit of its identified gene knockouts.

Model	CAR T cell source	Stimulation	CAR antigen	Costimulation	Data
In vitro expansion	Healthy	TCR stimulation	CD19	41BB, CD28	S8m
Leukemia	Healthy	NALM6	CD19	41BB	4c-j, S8h-i,k-l
	Myeloma patient	NALM6	CD19	41BB	4c-j, S8i
	Healthy	NALM6	CD19	CD28	S8n
	Healthy	NALM6	GD2	41BB	S8o
Solid tumor	Healthy	Huh7	GPC3	41BB	S8p

Redacted

Reviewer Table 1. Experiments to validate and generalize the identified knockouts (new data highlighted in blue).

Including additional sgRNA controls outside the primary screening experiments in supplementary materials would address potential concerns about guide specificity and data robustness.

We agree that off-target effects are an important concern when interpreting CRISPR screening data. This issue is typically addressed by using multiple independent gRNAs per gene and requiring consistent effects as evidence of on-target activity. For our genome-wide discovery screens (Fig. 1-2), we used four gRNAs per target gene from the Brunello library (a widely used and well optimized library designed to minimize off-target activity). For our *in vivo* pooled screen (Fig. 3), we expanded this to eight gRNAs per gene by including four newly designed, independent gRNAs. In downstream validation experiments (Fig. 4-5), we tested either all eight gRNAs or selected individual gRNAs, depending on the assay.

To address this reviewer comment, we added a new figure panel showing the performance of each individual gRNA targeting RHOG, FAS, and PRDM1 in the screen with the focused *in vivo* library (Fig. S9a-b). This analysis reveals highly consistent effects across all eight gRNAs, further supporting the robustness and specificity of our knockout effects.

Adding model details or discussing alternative models for FasL-negative Nalm-6 cells used in FAS knockout experiments would provide the necessary context to assess the generalizability of the FAS results.

We thank the reviewer for raising this potentially confusing point, which we now clarify in our revised manuscript.

A recent preprint (Yi et al. 2024 bioRxiv, doi: 10.1101/2024.02.26.582108) investigated the expression of FASLG (which encodes FasL) in single-cell RNA-seq datasets for various cancers. It reports that FASLG is not widely expressed in cancer cells nor in stromal cells but is specifically expressed in T cells and NK cells. Moreover, RNA-FISH in patient bone marrow samples found that FASLG was more highly expressed in CAR T cells compared to endogenous T cells. Additional *in vitro* and *in vivo* experiments in the preprint showed that FasL is upregulated upon CAR T activation and contributes to fratricidal killing following antigen uptake by trogocytosis, which reduces CAR T cell persistence.

These data are consistent with our own RNA-seq profiling data, where we observed increased expression of FASLG in activated CAR T cells (Reviewer Fig. 1a) and low expression of FASLG in all three cell lines used for the tumor models in our study (Reviewer Fig. 1b). Together, these observations suggest that FasL is primarily derived from the activated CAR T cells themselves, not from the NALM6 cells, which supports the interpretation that the beneficial effect of FAS knockout in our co-culture experiments is due to protection from T cell-T cell fratricide (Reviewer Fig. 1c).

These observations and the fact that FAS ablation in CAR T cells has been reported as beneficial in several studies (PMID: 28199983, 38387457, 39558141) and is used in a clinical trial (NCT05617755) support its broad applicability, regardless of FasL expression on tumor cells. We have updated the revised manuscript with these considerations (lines 267, 298).

Reviewer Fig. 1. (a) FAS and FASLG expression based on RNA-seq (this study, Table S7) for CAR T cells before and after stimulation with antigen-expressing cancer cells for 24 hours (3 independent T cell donors). (b) FAS and FASLG expression obtained from the DepMap portal for the cancer cell lines used in this study. FASLG expression was low for each of these cell lines, but the FAS knockout still led to enhanced CAR T cell performance. (c) Schematic of the FAS-FasL interaction between CAR T cells contributing to fratricide based on the recognition of tumor antigen following trogocytosis.

References

The authors appropriately reference prior work within the manuscript, including key background on CAR T cell limitations and prior CRISPR techniques. However, the introduction could more thoroughly frame the contributions of this new platform in light of other high-throughput CRISPR approaches to offer a clear comparative context for readers.

We thank the reviewer for the positive feedback on our background section. We agree that a comparison to published CRISPR screens in primary T cells and CAR T cells could strengthen the manuscript. We have addressed this through new text that more clearly highlights the key features of our CELLFIE platform: its unprecedented scale, excellent quantitative performance (now with detailed benchmarking data confirming this claim, Fig. S4), broad support for different CRISPR modalities, *in vivo* screens with UMI-based clonal tracing, and the newly added feature of combinatorial CRISPR knock-out screening in CAR T cells (Fig. 6a-d, Fig. S10, Table S8).

Clarity and Context

The abstract and introduction are technically dense and may benefit from adjustments for clarity. Simplifying the language in these sections could enhance accessibility and emphasize the primary advantages of CELLFIE to a broader audience, ensuring that the platform's clinical impact and practical value are immediately clear. Additionally, refining the flow of the introduction to highlight key limitations in CAR T cell therapy and how CELLFIE addresses these could improve readability and strengthen the overall narrative.

We appreciate the reviewer's suggestions to make our manuscript more accessible and to better highlight the value and impact of our work. In response, we revised both the Abstract and the Introduction section to reduce overly technical language, enhance overall clarity, and improve the flow of the opening paragraphs. We also explain more clearly how CELLFIE addresses key challenges in CAR T cell therapy and how this contributes to the platform's translational potential.

Reviewer #2

In this manuscript, Datlinger, Bock, and colleagues describe CELLFIE, a platform for genome-wide screens in human CAR-T cells. CELLFIE involves several technical advances, including: (1) a new CROP-seq vector for co-delivery and readout of a CAR and a library of CRISPR sgRNAs, (2) a scalable mRNA system for low-cost delivery of CRISPR editors including both Cas9 nuclease and Cas9-derived base editors, and (3) RNA-based readout of sgRNA identify alongside a UMI to read out sgRNA abundance in the context of in vivo screens that probe the function of a small quantity of CAR cells in native mouse tissue.

Novel aspects and outcomes from their experiments include (1) systematic and large-scale screening across many conditions and with both proliferation and FACS-based phenotyping, (2) pooled in vivo screens with xenograft models in primary human T cells with both a CAR and CRISPR perturbations with proliferation-based phenotyping, (3) the discovery of knockouts such as FAS, PRDM1, and RHOG that increase the proliferation and activation of CAR T cells in vitro and in vivo and further increase the activity of CAR T cells against tumor cells in mouse xenograft tumor models, and base editor screens that implicate the GTP-binding pocket of RhoG in its CAR-function-relevant activity and suggest a path toward the abrogation of RHOG function in clinical CAR T cells without the induction of double strand breaks.

The authors are not the first to conduct genome-wide CRISPR screens in human T cells or Car T. For example, Shifrut et al (PMID 30449619) conducted genome-wide screens by the electroporation of Cas9 protein into T cells expressing lentivirally delivered CRISPR sgRNAs. Wang et al (PMID 33328215) conducted genome-wide screens by the lentiviral introduction of sgRNAs and a CAR and the electroporation of Cas9 protein into T cells. Freitas et al (PMID 36356142) conducted genome-wide screens in CAR T cells by the lentiviral introduction of sgRNAs, the electroporation of Cas9 protein, and the delivery of a CAR by a retrovirus. However, this new manuscript from Datlinger, Bock et al is distinguished by the immensity of the effort (dozens of FACS and enrichment-based screens), nice tricks for in vivo experiments, and possibly by the new targets it identifies.

We thank the reviewer for the positive assessment, and for highlighting novel aspects and the broad scope of our manuscript. In this revised version, with combinatorial screening we introduce a major new feature of the CELLFIE platform, which enables the systematic identification of gene knockout combinations that synergistically enhance CAR T cell performance. Inspired by the CROPseq-multi approach described in a preprint by Walton et al. 2024, we extended our CROP-seq-CAR vectors to support dual-gRNA expression and performed combinatorial screens targeting 238 gene pairs, focusing on our top-ranked CAR T boosters. These experiments further confirmed the beneficial effect of the RHOG-FAS double knockout and identified new knockout pairs that enhance proliferation (Fig. 6a-d, S10).

Our revised manuscript also includes extensive new data confirming the broad applicability of the identified RHOG and RHOG-FAS knockouts: across multiple CAR designs, in leukemia and in a solid tumor model, and in CAR T cells obtained from healthy donors and from a patient scheduled to receive CAR T cell therapy as part of clinical care (Fig. S8g-p).

In response to the reviewer's suggestions, we performed quantitative assessment of screening quality, incorporating additional and more biologically grounded metrics, and benchmarking against published datasets. Finally, we revised our interpretation and text around the base editing outcomes, provided additional new experimental validation, and added a comparative analysis of beneficial CAR T cell knockouts across studies.

Specific comments:

The authors assert the quality of their screens, but I think they could assess screen performance with additional metrics. The metric they often use "Percent alignment to gRNA library" is important, but it does not get at the quality of biological phenotypes per se. For example, the authors would presumably get a high "Percent alignment to gRNA library" even if they did not introduce Cas9 into their samples. Standard screen metrics such as the magnitude of the difference between the phenotypes of control sgRNAs and targeting sgRNAs would be useful. The authors hint at this with their MAGeCK effect sizes p-values, but I think that this could be explored further and visualized more across screens.

We fully agree on the importance of screening quality assessment, and we are committed to providing the CAR T cell community with a robust platform and comprehensive resource of genome-scale CRISPR screens in human CAR T cells. We appreciate the reviewer's suggestion to expand our quality metrics (Fig. S6b) and conducted several additional analyses to address this point, which are summarized in a new supplementary figure (Fig. S4) and table (Table S4).

First, to complement our alignment-based metric (which remains an important indicator of specific gRNA amplification and high-quality NGS libraries), we now report the percent of gRNAs with zero counts (as a measure of dropout events) and the Gini index (to quantify library representation and selection pressure). We have updated Tables S2g and S3q (for fitness and sorting screens, respectively) as well as Fig. S6b with these additional metrics.

Second, to directly assess the functional consequences of the CRISPR knockouts, we performed a systematic comparison of essential gene dropout across all our genome-wide screens. We focused on a curated set of 684 core essential genes (CEGv2) and 928 non-essential genes (NEGv1) from the BAGEL2 reference set (Kim and Hart, 2021). For each screen, we calculated \log_2 fold changes between the final screening timepoint and an early baseline sample collected prior to Cas9 mRNA electroporation (Fig. S4c). We then applied the widely used *strictly standardized mean difference* (SSMD) metric to quantify the separation between the gRNA prevalence distributions for essential and non-essential genes (Fig. S4f).

Our final dataset comprises 58 genome-wide screens addressing six phenotypes to improve CAR T therapies, replicated in multiple independent blood donors. All screens pass stringent quality control criteria, which we refined based on the reviewer's suggestions, and together provide a larger set of screens than all prior studies combined (Table S4a, Fig. S4a).

I was surprised by the low number of common essential genes that they confidently identified as drop outs and by the low overlap between screens (Fig 1C, left and right), as these screens often have very strong phenotypes in many other CRISPR screening contexts. I was also surprised and concerned by the relatively low separation of the core-essential from non-essential genes (Fig S3G). There may be reasons for these muted phenotypes, but the authors should discuss them in more depth.

We concur with the reviewer's emphasis on data quality, and we have now completed a systematic benchmarking and quantitative comparison with published genome-wide screens that confirmed the high quality of our screens (described in response to the next comment). We would also like to clarify that the moderate overlap of essential genes in Fig. 1c reflects the fact that these are results from two distinct screens (TCR and CAR stimulation), rather than replicates of the same experiment. Regarding the global separation between essential and non-essential genes in Fig. S3g (now: Fig. S3h), the results appear in line with what is typically observed in human primary T cell screens (Fig. S4c-d). Finally, the sets of essential and non-essential genes that underlie our analysis (CEGv2 and NEGv1) have not been optimized for CAR T cells nor for T cells, which suggests that our analysis likely underestimates the true quality of our screens.

The authors should compare the technical details of their screens and the screen quality with previous CRISPR screens in T cells and CAR T cells. I think a supplementary figure comparing these results to prior work would be very valuable. Although this body of work may be much more substantial and important than these previous papers, that may not be immediately obvious to readers.

We thank the reviewer for the suggestion to compare our screen quality with previous CRISPR screens in T cells and CAR T cells. In the revised manuscript, we have added a detailed supplementary figure (Fig. S4) and table (Table S4) that address this point through a systematic benchmarking analysis comparing with related studies. Our benchmarking included genome-wide screens by Shifrut, Carnevale et al. 2018, Wang, Prager et al. 2021, Carnevale, Shifrut et al. 2022, and Freitas, Belk et al. 2022. To further contextualize our results, we also incorporated genome-scale screens from the Jurkat T cell line using data from the Broad Institute's DepMap project, which provide a valuable reference point for comparing primary T cell engineering to the more technically straightforward screening in cancer cell lines.

For this benchmarking, we extended our *strictly standardized mean difference* (SSMD) analysis to quantify the separation between essential and non-essential genes across all available public datasets. To promote reproducibility, we provide a detailed overview of all studies, datasets, and source files used (Fig. S4a, Table S4b, Methods). We also carefully reviewed the methods sections of each publication and found the experimental timelines to be well aligned (Fig. S4b). Wherever possible, we started from raw gRNA count data, recalculating \log_2 fold changes between the main screening timepoint and a suitable reference (e.g., plasmid DNA or pre-editing samples). Because Wang, Prager et al. 2021 reported only gene-level beta values (no raw counts, precomputed fold changes, or MAGeCK outputs) and did not disclose the screening timeline, we reprocessed our data using the same MAGeCK MLE parameters, assuming a day 7 time point.

Among all screens that used the Brunello gRNA library (Shifrut, Carnevale et al. 2018, Carnevale, Shifrut et al. 2022, Wang, Prager et al. 2021), our CELLFIE platform demonstrated the strongest performance separating essential versus nonessential genes (Fig. S4f). The second-best performance was observed for Shifrut, Carnevale et al. 2018; the third-

best performance for Canevale, Shifrut et al. 2022, and Wang, Prager et al. 2021 showed the lowest performance with no detectable depletion of essential genes (Fig. S4d-e). The Freitas, Belk et al. 2022 dataset showed slightly higher separation than CELLFIE thanks to an almost threefold larger gRNA library (211,695 gRNAs as opposed to 77,441 gRNAs in CELLFIE and the other screens that used the Brunello gRNA library). Moreover, we are delighted to report that both the CELLFIE screens and the Freitas, Belk et al. 2022 dataset came close to the performance of screens in the Jurkat T cell line (DepMap release 24Q4 by the Broad Institute), despite the difficulty of screening in human primary CAR T cells.

Additionally, direct comparisons of hits and gene phenotypes to prior efforts would be very valuable.

This is a great suggestion that we addressed with new bioinformatic analyses. We included four of the five studies from our benchmarking in response to the previous point, including the screens from our study. We excluded the Wang, Prager et al. 2021 dataset due to its low data quality according to our benchmarking. We determined statistically significant hits based on the MAGeCK RRA output with a z-normalized log₂ fold change >2 and FDR <0.05 for all screens, except for our CAR and TCR fitness screens, for which we instead used beta values from MAGeCK MLE normalized against essential genes (using the `NormalizeBeta` function in MAGeCKFlute) because of the different growth rates in the TCR and CAR stimulated screens), applying a threshold of >0.99 in the same way as in our analysis in Fig. 1.

A summary of the hits for each of the analyzed screens is provided in Table S4f, with detailed results in Table S4g. Figure S4g shows a heatmap of genes identified as hits in at least one screen, with their results across all screens. Our analysis identified surprisingly little overlap between these studies, which may be explained by technical differences between the screens (e.g., different ways of stimulation and different CARs). Reassuringly, CELLFIE identified established clinical targets of cancer immunotherapy (PD1, CTLA4, TIGIT, TIM3), which were not found in any other study and which provides further support that our screening setup is well-suited for identifying hits with the potential for clinical translation. More generally, the low overlap of screening hits between studies underlines the continued value of CRISPR screening in CAR T cells, as different conditions will identify and profit from different knockouts and combinations of knockouts. Our CELLFIE platform with its support for diverse (now including combinatorial) screens is well suited to tackle this task.

—There is no validation of the base editing data. I think that claims as precise as “this mutation abrogates function in a clinically relevant way” are worthy of individual validation.

We thank the reviewer for the suggestion. For our revised manuscript, we validated the base editing screens with three additional *in vitro* experiments. First, we performed focused screens with 323 top-performing gRNAs selected from the initial base editing screens (Fig. S11h, Table S9) and identified gRNAs with strong and consistent effects across five additional T cell donors. Second, we confirmed the genetic and functional consequences of the base editing by targeted next-generation sequencing of the RHOG locus in ABEmax-edited samples (Fig. S11i), where we observed the expected time-dependent enrichment of mutations that are the predicted editing outcomes of top-scoring gRNAs (Tyr72His, Cys81Arg). Third, we identified individual gRNAs that efficiently introduced these edits (Fig. S11j-k) and conferred a strong proliferation advantage in primary CAR T cells. Specifically, we selected three RHOG-targeting gRNAs with base editing frequencies of ~50% at the initial timepoint and observed an expansion to nearly 100% after one week in culture.

These results validate the utility of base editing screens for identifying functionally impactful missense mutations and for prioritizing gRNAs to efficiently introduce such mutations into cells. While thorough, our validations are still limited to the *in vitro* setting. We agree that establishing clinical relevance of individual gRNAs will require further studies using *in vivo* models, and we have revised the manuscript to reflect this distinction with more cautious and precise language.

Reviewer #3

This is an interesting and carefully executed study to identify boosters of CAR-T cell immunotherapy. The study features an elaborate, systematic screening platform (CELLFIE) that comprises a FACS-based fitness screen and an *in vivo* CROP-seq screening and nominates three gene knock outs as boosters of CAR T cell performance (FAS, PRDM1, RHOG). The authors nominate RHOG as their lead candidate that augments the fitness of CD8+ and CD4+ CAR T cell of healthy donors in pre-clinical models of CD19 CAR T cell therapy *in vitro* and *in vivo*.

The study provides an interesting technical variant compared to prior screening campaigns with CRISPR gRNA libraries, and some advantages over prior work in that screening is performed both *in vitro* and *in vivo* in mice.

We would like to thank the reviewer for the positive comments regarding the CELLFIE screening platform and our systematic combination of *in vitro* and *in vivo* screens. In the revised manuscript, we added multiple new experiments and analyses that further validate the high data quality of our screens and the value of the CELLFIE platform for optimizing CAR T cells through high-content CRISPR screening. We also appreciate the reviewer's critical comments regarding the potential clinical relevance of the identified lead candidate. We have now performed additional validation experiments of our knockout CAR T cells as suggested by the reviewer, to substantiate our findings and broaden their applicability.

However, there are also several major limitations of the study:

- CELLFIE: The CAR T cell fitness screening procedure, even though elaborate, is a 28-day tour-de-force that involves a pre-expansion, bead-activation, viral transduction, electroporation, selection with blasticidin/puromycin and expansion step. This does clearly not reflect the state of the art in T cell screenings, and is far from reality of clinical CAR T cell production (where 24 hours of culture or less is the new gold standard). It is uncertain if gene candidates that are identified in this screen may be biased through the selection of a limited number of T cell clones that survive the rigors of the screening procedure. No or insufficient data on T cell subset composition and polyclonality of the input and output populations, T cell numbers, yields and repeats of the screening process are provided.

We agree that streamlined CAR T cell production protocols with minimal culture times (or *in vivo* delivery of the CAR) have important advantages for CAR T manufacturing and clinical applications. In contrast, screening a genome-wide library with ~80,000 gRNAs requires pre-expansion for sufficient cells per gRNA and extended culture for efficient gene editing. In some cases, the proliferation of the gene-edited cells is also the screening phenotype, as for our CAR and TCR fitness screens. Here we provide four lines of evidence to support that our screening setup is meaningful and relevant:

1. **Our screening timeline and data quality are well aligned with state-of-the-art screens.** We compared the timeline of our screens with those used in published genome-wide CRISPR screens in human primary T cells, including studies by Shifrut, Carnevale et al. 2018, Carnevale, Shifrut et al. 2022, and Freitas, Belk et al. 2022. As shown in Reviewer Fig. 2 (see below) and Fig. S4b, our timeline aligns well with these studies and is in some cases even shorter. To further validate the data quality of our screens, we benchmarked our screening results and performance metrics against Shifrut, Carnevale et al. 2018, Wang, Prager et al. 2021, Carnevale, Shifrut et al. 2022, and Freitas, Belk et al. 2022, and against genome-wide screens for the Jurkat T cell line from the Broad Institute's DepMap project. This benchmarking analysis, presented in the new Fig. S4 and Table S4, confirms that our screens meet or exceed current benchmarks for data quality and performance.
2. **We conducted our screens thoroughly and with an adequate number of biological replicates.** In our revised manuscript, we provide further documentation of measures taken to ensure high technical and procedural quality for all screens (Table S2e, S3g, Methods section) and extensive assessments of data quality (Fig. S4, S6b). Numbers of biological replicates are visualized in Fig. S6a, and the genome-wide screening timelines are shown in Fig. S3c and S5j. Our screens were conducted with T cells collected from four independent donors for the fitness screens and two to four independent donors for each of the FACS-based screens.

Experimental timelines across (CAR) T cell screening studies

Reviewer Fig. 2. Screening timelines for CELLFIE and published genome-wide CRISPR screens in primary human T cells.

- We now provide further details regarding T cell numbers and subsets in support of our screening setup.** We documented T cell numbers at all key stages of our genome-wide screens and have now added this information in Table S2I (fitness screens) and S3p (sorting screens). T cell isolation from 500 ml of peripheral blood for individual donors yielded 30 to 200 million T cells per donor (Reviewer Fig. 3a). We cultured all cells to achieve >1000x gRNA at both the transduction and electroporation stage, meaning that the effect of each gRNA was assessed in more >1000 individual CAR T cells (Reviewer Fig. 3b). This is well above the 500x gRNA coverage recommended by Doench 2018, ensuring high representation of each perturbation. We also addressed the reviewer’s suggestion regarding T cell subsets and now provide flow cytometry data throughout the screen (Fig. S3g), confirming the expected differentiation into effector and memory subsets during cell expansion.

Reviewer Fig. 3. (a) T cell numbers obtained for genome-wide screening from the entire volume of blood donations by healthy donors. (b) Estimated gRNA coverage at the transduction and electroporation stage of our screens, assuming a 10% transduction rate. All samples exceeded the 500x coverage standard for pooled CRISPR screens (Doench 2018).

- We conducted an additional screen with UMI-based clonal tracking and confirmed very high polyclonality.** To directly address the reviewer’s concern about insufficient polyclonality resulting in cell population bottlenecks that could negatively affect the quality of our screens, we performed a pooled *in vitro* screen with unique molecular identifiers (UMIs) – short, randomized DNA sequences that barcode individual T cell clones. We compared samples at Day 0 (pre-editing, two days after transduction, four days after T cell isolation) and Day 12 (after electroporation, selection, and expansion). The total number of detected T cell clones decreased by less than 50% (from 1,179,618 to 667,258, Reviewer Fig. 4a), and the effective coverage of the screen was retained at ~65% on Day 12, with an average of 15,165 clones per gene compared to 26,809 clones at baseline (Reviewer Fig. 4b). These results demonstrate that the screen preserves the very high polyclonality of the starting population, even across multiple engineering and selection steps. This analysis conclusively rules out cell population bottlenecks as a relevant confounder of our screens.

Reviewer Fig. 4. Clonal diversity before (Day 0) and after (Day 12) genome engineering, measured by unique molecular identifiers (UMIs). Results confirm preservation of very high polyclonality throughout our screen.

In summary, these four lines of evidence confirm that our screens yield high-quality data. Nevertheless, genome-wide *in vitro* screens are always conducted in a context that does not fully reflect the human *in vivo* situation, and they require certain trade-offs to accommodate large library sizes and efficient genome engineering. We would also like to emphasize that our genome-wide screens were only the starting point of the CELLFIE discovery process and were followed by pooled *in vivo* validation with clonal tracking, single-gene *in vivo* confirmation, and tiling base editing of RHO G. These *in vivo* screens and validation experiments were performed using much shorter CAR T cell preparation timelines (Fig. 4b).

- FACS-based screening: The parameters selected for the FACS-based screening assay are erroneous and not supported by data in this manuscript or published data in the field. CD19 expression on CAR T cells due to trogocytosis can be the result of a high affinity CAR binding domain (like FMC63) but is by no means a marker of strong target cell engagement (in a positive sense). Activation markers like CD69 should be aligned with CD25 and other activation markers like PD1, CTLA4 and TIM-3 – that only become markers of exhausted CAR T cells after sequential rounds of target engagement (but not after a single activation/expansion cycle). There are no data to support the FACS-based screening assay, e.g. with CARs that target the same antigen (CD19) with distinct affinity, or CARs that target other antigens (e.g. CD20/CD22; or GD2 with the 14D2 binding domain that induces tonic signaling and would be a relevant control).

We appreciate the reviewer's critical assessment of our FACS-based screening readouts and apologize for the lack of clarity and justification in our original manuscript. The reviewer raises important biological points regarding trogocytosis, activation kinetics, and early exhaustion markers, which we have taken into careful consideration. To address these points with additional experiments, we measured all sorting markers across three different CARs (including the requested GD2-BBz CAR). We also revised the manuscript to avoid oversimplifications and to better convey the rationale and limitations of our marker choices. A summary of our main points and adaptations is provided below:

- 1. Validation across multiple CARs.** We agree that it is relevant to evaluate the robustness of our FACS-based markers across different CAR designs. For this experiment, we added CD19-CD28z, which differs from CD19-BBz in its co-stimulatory domain, and GD2-BBz, which targets an unrelated antigen and incorporates the suggested 14g2a scFv (which is known to induce tonic signaling). We thus profiled our FACS-based markers across three CARs by flow cytometry (Fig. S5a-d), and we refer to these experimental results in the points below.
- 2. CD69 as a marker of T cell activation.** Both CD69 and CD25 are widely used and reliable markers of T cell activation, and we observed strong upregulation of both markers for each of the three tested CARs. However, their expression kinetics differ: CD69 peaks at 4 hours, while CD25 peaks at 24 hours, when CD69 expression has already declined. This makes it impractical to sort for both markers simultaneously. Including CD25 would require a separate genome-wide screen – which could be interesting, but we found it more valuable to spread our FACS-based screens across a more diverse range of readouts. CD69 has been effective as a sole marker in genome-wide T cell activation screens before (Shang et al. 2018) and was confirmed for the three tested CARs.

3. **Use of inhibitory receptors as screening markers.** We agree that PD1, LAG3, and TIM3 cannot be considered definitive markers of terminal exhaustion after a single stimulation, and we apologize for the oversimplification in the original manuscript. While PD1 has previously been used as a marker of exhaustion in genome-wide CAR T cell screens (Wang, Prager et al. 2018), transient upregulation can indeed occur during early stimulation cycles, which makes it difficult to distinguish early exhaustion from T cell activation. Even with this limitation, these markers are still useful when screening for regulators of early inhibitory signaling, especially in combination with the validation by *in vivo* screening, which is an integral feature of the CELLFIE platform.
4. **CD19 as marker of tumor cell recognition.** It is true that CD19 accumulation on CAR T cells reflects trogocytosis following tumor cell recognition rather than direct target engagement. We clarify this point in our revised manuscript by referring to the CD19 readout as “tumor cell recognition” instead of “engagement”. Our new data across multiple CARs highlight substantial differences of trogocytosis levels on CAR T cells (Fig. S5a), which may be explained by different CAR affinities, trogocytosis dynamics, and antigen expression on target cells (Fig. S5b). We also observed a characteristic increase in antigen-positive CAR T cells over time (Fig. S3e), which suggests continued CD19 uptake and which supports the value of our CD19 readout as a proxy for tumor cell binding.

In summary, while we acknowledge certain limitations of these FACS-based readouts, our screens combine six complementary readouts (Fig. 2f), with follow-up *in vivo* screens and validations providing a highly specific filter to identify hits that truly enhance CAR T cell performance. We focused on hits from one end of each marker spectrum (either high or low marker expression), but the manuscript includes the full data for all markers. This broader dataset may serve as a valuable resource for further exploration of CAR T cell regulators. We would like to conclude this point by thanking the reviewer for the excellent comments, which helped strengthen the interpretation of our FACS-based readouts.

- The *in vivo* screen is performed with a limited number of gRNAs only (targeting 39 genes) which reduces the technical appeal and introduces a bias to genes that were pre-selected in the FACS-based screen and then filtered for genes with known function in T cells.

The CELLFIE platform supports much larger *in vivo* screens, but in this study we used it for hit validation rather than for *de novo* discovery. This led to design choices that intentionally limited the throughput in favor of maximizing data quality – for example by using eight gRNAs per gene and transducing at low MOI to ensure single-gene knockouts. Starting from the current screen of 39 genes with eight gRNAs each, it is straightforward to double the gene number by using four instead of eight gRNAs per target gene (a widely accepted standard). Moreover, in the revised manuscript we introduce combinatorial knockout screens with a dual-gRNA vector. With this new capability, we can deliver two gRNAs per cell, which further doubles the number of target genes to 156 genes in just 5 animals. Scaling the experiment to 50 mice (which is still a manageable number) would enable testing of over 1,500 target genes. At this scale, most genes are not expected to have strong effects, meaning that we can increase the MOI to allow an average of two infections per cell, which would bring the total number of target genes to over 3,000. We are very excited about the perspective of applying CELLFIE for large-scale *in vivo* discovery screens in future projects, and our manuscript provides all the materials that will enable a broad range of researchers to conduct such *in vivo* screens with full UMI support for clonal tracking.

In response to the reviewer’s additional point about potential biases of pre-selecting genes, we would like to clarify that our gene selection for *in vivo* screens was data-driven (normalized MAGeCK MLE $\beta > 0.99$ for the fitness screens and $\log_2FC > 2$, FDR < 0.05 for the FACS-based screens), with RNA levels in human CAR T cells as an additional filter (Fig. S6e). We have clarified this in the Methods section (line 1056-1058) and the corresponding Results section (line 249-252).

- The authors perform all experiments with healthy donor T cells, where patient T cells would have been relevant. This is a major deficiency of the manuscript, and prevents relevant conclusions for CAR T cell therapy, where all approved, and the majority of investigational CAR T cell are autologous from the patient.

We thank the reviewer for bringing up the importance of testing in patient-derived CAR T cells as a further step on the path to clinical translation. To our knowledge, none of the previously published genome-scale CRISPR screens in human T cells or CAR T cells included such validations in patient-derived T cells **Redacted**

1. **Validation in CAR T cells that we prepared from patient-derived T cells.** We generated CAR T cells from leftover leukapheresis product of a patient diagnosed with multiple myeloma and scheduled to receive CAR T cell therapy. Cells were engineered to express the 19-BBz CAR in combination with: RHOG knockout, FAS knockout, dual RHOG-FAS knockout, or no edit. We used the shortened CAR T cell production timeline from our validation workflow (Fig. 4b, right). The *in vivo* function of these CAR T cells was assessed in the NALM6 leukemia model, where we observed improved tumor control and extended survival for patient-derived CAR T cells with the knockouts, consistent with our findings using healthy donor-derived CAR T cells (Fig. S8i, right).

Redacted

- The authors perform experiments with a single CAR construct directed against CD19, which is insufficient to establish a role of RHOG (as well as FAS, PRDM1) for CAR T cell immunotherapies. Validation in at least one additional hematology model (with a target other than CD19) and one additional oncology model would be required before considering such a claim.

We appreciate the reviewer's suggestion to validate RHOG knockout CAR T cells in additional models. For the revision, we performed major new experiments to assess the effects of RHOG and RHOG-FAS knockout across several CARs (with different antigen specificities and co-stimulatory domains) and cancer models. Our new data provide strong evidence that the knockout effects generalize beyond the specific CAR and disease model in which they were discovered, which enhances their potential for translation. We summarize these new experiments below (Reviewer Table 1, Figure S8):

1. To test the efficacy of our knockout CAR T cells beyond CD19-targeting CARs, we established a GD2-expressing NALM6 leukemia model and treated the mice with anti-GD2 CAR T cells that use the 14g2a scFv (which is known to induce tonic signaling). RHOG knockout enhanced CAR T cell anti-tumor activity also in this setting (Fig. S8o). We thank the reviewer for suggesting this particular CAR, allowing us to validate our knockouts in a new setting with a different antigen and a CAR with different properties including high baseline signaling activity.

2. We further evaluated the RHOG knockout in anti-CD19 CAR T cells with a CD28 co-stimulatory domain instead of 41BB. These CAR T cells similarly benefited from the RHOG knockout (Fig. S8n).
3. Going beyond leukemia, we tested anti-GPC3 CAR T cells in the Huh7 xenograft model of hepatocellular carcinoma. In this solid tumor model, and RHOG-FAS knockout again yielded improved tumor control (Fig. S8p).
4. Complementing these specific validations with a broader comparative assessment of knockout pairs, we established combinatorial screens for double gene knockouts using our new CROP-seq-CAR-multi vectors and used it to evaluate pairwise combinations of our top screening hits across three different CARs (CD19-BBz, CD19-28z, GD2-BBz) (Fig. 6a-d, S10). We identified several pairs of gene knockouts that consistently improved performance across all three CARs (Fig. 6c-d). This combinatorial screen also independently confirmed the effect of the dual RHOG-FAS knockout, providing further validation across three CARs. These results thus validate not only the RHOG-FAS knockout but also the discovery power of our new CELLFIE combinatorial screening method.

Model	CAR T cell source	Stimulation	CAR antigen	Costimulation	Data
In vitro expansion	Healthy	TCR stimulation	CD19	41BB, CD28	S8m
	Healthy	NALM6	CD19	41BB	4c-j, S8h-i,k-l
Leukemia	Myeloma patient	NALM6	CD19	41BB	4c-j, S8i
	Healthy	NALM6	CD19	CD28	S8n
	Healthy	NALM6	GD2	41BB	S8o
Solid tumor	Healthy	Huh7	GPC3	41BB	S8p

Redacted

Reviewer Table 1. Experiments to validate and generalize the identified knockouts (new data are highlighted in blue).

- The study contains no relevant assessments and resolve in terms of genotoxicity and the risk for secondary malignancy that may result from FAS, PRDM1, RHOG knock out or modulation.

We agree with the reviewer that assessing genotoxicity and off-target effects is an important element of clinical translation – especially given recent concerns about secondary malignancies arising from CAR T cells. Assays such as GUIDE-seq (Tsai et al., 2015), which maps genome-wide off-target activity, CAST-seq (Turchiano et al., 2021), which measures chromosomal rearrangements, and whole-genome sequencing can sensitively detect unintended editing events. However, such analyses are only meaningful once the exact therapeutic design intended for clinical testing has been finalized (including final gRNAs, editor platform, and GMP-grade manufacturing plan), which is beyond the scope of this study.

What we were able to do at this stage was to implement long-term monitoring of the mice that were successfully treated with RHOG or RHOG-FAS knockout CAR T cells. We analyzed tail bleeds, blood, bone marrow, and spleen of these mice over time, up to 392 days after CAR T cell treatment. We found that the frequency of CAR T cells was consistently below 1% of cells in each organ (Fig. S8l, Table S6c). While these data cannot exclude rare malignant transformation of RHOG or RHOG-FAS knockout CAR T cells, all the data we have been able to collect so far support the safety of our knockouts.

To minimize genotoxicity risk in the future clinical translation of our knockout CAR T cells, we performed base editing screens to identify mutations abrogating RHOG function, to avoid the use of CRISPR nucleases introducing double-strand breaks. In this revision, we conducted additional experiments and validated specific RHOG missense mutations with potential for therapeutic development (Fig. S11h-k).

Finally, we would like to emphasize that the primary contribution of this study lies in the CELLFIE platform, the genome-scale datasets, and the validated knockouts that enhance CAR T cell function, while we are well aware of the extensive work that is still needed to establish RHOG and/or RHOG-FAS knockout CAR T cells as clinically used CAR T cell therapies.

We would like to conclude by thanking all reviewers for their valuable feedback, which helped us refine and extend our study in multiple ways, collectively enhancing the depth and anticipated impact of the presented method and insights.

Summary of the second revision:

We would like to thank the reviewers for their positive feedback and the editor for the decision to offer publication in *Nature*. In response to the second-round reviewer comments, we have further clarified several aspects of our manuscript, and we added a new analysis of T cell clones in our *in vivo* CROP-seq screens and an extrapolation of the scalability of these screens to larger numbers of target genes. Further, we included a computational off-target analysis for our top RHOG and FAS targeting gRNAs. A detailed point-to-point response to all reviewer comments is provided below.

Point-by-point response to the reviewer comments

Referee #1

A. Summary of the Key Results

The authors have validated RHOG and RHOG–FAS knockout across four CAR constructs (CD19-BBz, CD19-28z, GD2-BBz, GPC3-BBz) in both leukemic (NALM6) and solid (Huh7) models using patient-derived CAR T cells, demonstrating consistent improvements in *in vivo* persistence and tumor clearance; they also benchmarked their 58 genome-wide screens against Shifrut 2018, Wang 2021, Carnevale 2022, Freitas 2022, and DepMap/Jurkat data, showing equal or superior separation of core essential versus nonessential genes and identifying RHOG as a novel positive regulator. Mechanistic follow-up reveals that RHOG knockout shifts CAR T cells toward a central memory phenotype with reduced exhaustion, and FAS knockout mitigates activation-induced apoptosis, while a new CROPseq-multi dual-gRNA system enabled combinatorial screening of 238 gene pairs that confirmed RHOG–FAS synergy *in vitro* and produced durable cures *in vivo*; a base-editing tiling screen in RHOG identified key GTP-binding pocket residues whose editing phenocopies knockout.

B. Originality and Significance

Introducing pooled dual-gRNA combinatorial screening in primary human CAR T cells via CROPseq-multi is a clear advance over prior work that relied on immortalized lines or single-guide approaches, and the finding that RHOG (historically linked to immunodeficiency) enhances CAR T proliferation and persistence is highly novel; validating RHOG and RHOG–FAS knockout benefits across four CAR backbones in both hematologic and solid tumor settings using patient-derived CAR T cells under-scores broad translational impact. To strengthen claims of novelty, the authors could briefly compare CROPseq-multi with Medicorico et al. (Cell 175, 258–270, 2023) and note in the Discussion that CELLFIE supports multiple CRISPR modalities (knockout, base editing, CRISPRa, CRISPRi) alongside combined phenotypic and *in vivo* readouts.

We were unable to locate the paper referenced as “Medicorico et al. (Cell 175, 258–270, 2023)”. However, there are multiple other papers that are directly related to our study. We cite many of these papers in our manuscript and provide a comparative analysis of high-quality screens in primary T cells and CAR T cells (Extended Data Fig. 4). Due to journal constraints on the reference count, we could not include all relevant studies and we apologize to authors whose work was omitted solely for space reasons. As suggested by the reviewer, we have now added a sentence to the Discussion noting that CELLFIE supports diverse CRISPR modalities, including knockout, base editing, CRISPR activation (CRISPRa), and CRISPR interference (CRISPRi).

C. Data & Methodology

The Methods are now organized into clear subsections and include quality-control metrics (library alignment rates, zero-count guide percentages, Gini indices, SSMD values) that confirm screen quality matches or exceeds prior studies; the nested PCR protocol for *in vivo* CROP-seq is well described, and the CROPseq-multi cloning workflow is detailed sufficiently to allow replication, while *in vivo* validations use at least five mice per arm with bioluminescence imaging and log-rank analysis. To enhance reproducibility, the authors should state the exact *n* for each mouse cohort, clarify whether guides with fewer than twenty UMIs were excluded from analyses, and indicate the average number of unique CAR T cell UMIs recovered per sample.

We thank the reviewer for the suggestion. In the revised manuscript, we indicate the exact number of mice for each experiment, and we provide dedicated Source Data files for all *in vivo* experiments as part of the Supplementary Information, in compliance with the *Nature* reporting guidelines. No gRNAs were excluded from the analysis, but our analysis focuses on knockout effects that were supported by high UMI counts corresponding to independent CAR T cell clones (Reviewer Fig. 4 from the first revision round). We now report the number of unique CAR T cell clones (based on UMIs) for all *in vivo* screening samples (Extended Data Fig. 6j). For convenience, this figure panel is also reproduced below as Reviewer Fig. 1.

Reviewer Fig. 1. Number of distinct T cell clones detected based on unique molecular identifiers (UMIs) in the *in vivo* screens. This figure panel is identical to Extended Data Fig. 6j.

D. Appropriate Use of Statistics and Treatment of Uncertainties

Statistical methods are applied consistently: genome-wide screens use MAGeCK MLE with stringent thresholds, UMI subsampling reduces PCR bias, survival analyses employ log-rank tests with hazard ratios and confidence intervals, and flow cytometry data use t tests or ANOVA with exact p values, while confidence intervals and interaction scores are appropriately shown in the base-editing tiling and combinatorial assays. To improve interpretability, the SSMD axis in Figure S4f should be annotated with thresholds (for example, $SSMD \geq 2$ indicates excellent separation) and the text should note exclusion criteria for low-coverage guides to emphasize how stochastic noise was minimized.

To improve interpretability, we have labeled the corresponding axis in what is now Extended Data Fig. 4f as “SSMD (higher is better)”. However, rather than defining strict performance thresholds, we refer to the Jurkat cell line screens as an empirical reference of a high-quality screen.

All *in vivo* screening data were analyzed using the MAGeCK-iBAR method for UMI-based CRISPR screens. This method evaluates gRNA enrichment based on the performance of individual UMIs, while down-weighting inconsistent fold-change directions across clones. We found MAGeCK-iBAR to be highly effective in mitigating experimental screening noise, and our analyses yielded robust high-confidence hits that were subsequently validated by further experiments.

E. Conclusions: Robustness, Validity, Reliability. The data convincingly show that the benefits of RHO G knockouts are robust across four CAR backbones, two tumor types, and two CAR T sources, with consistent improvements in persistence, tumor clearance, and survival; RHO G–FAS synergy is validated both *in vitro* and *in vivo*, yielding durable cures in a model that otherwise fails to achieve long-term control. The authors should note that the observed central memory shift (roughly a 10-15% increase in CD62L⁺CD45RO⁺ cells) is modest rather than a complete reprogramming and briefly mention any long-term assessment of off-target effects in non-CAR immune subsets to reassure readers that RHO G–FAS double knockout does not produce unintended deleterious consequences.

We agree that the observed increase in the fraction of central memory cells among RHOG knockout CAR T cells is relatively modest. We note that this shift is only one of several effects of RHOG knockout on CAR T cells that we observed and unlikely the sole cause of their increased performance.

Regarding potential off-target effects on immune cells other than the CAR T cells, we consider contamination with non-T cell populations highly unlikely due to the stringent T cell isolation and TCR-driven expansion as well as antibiotic selection for CRISPR-edited CAR T cells. Given that we deliver the CAR in the same vector as the gRNA, it is also unlikely that T cells will receive the knockout but not the CAR.

Nevertheless, on the path toward future clinical testing, it will be important to further characterize the safety profile of the RHOG-FAS knockout CAR T cells. For example, as described in our response to Reviewer 3 in the first revision, we envision that the final setup for genome editing will be validated by assays such as GUIDE-seq, CAST-seq, and whole-genome sequencing to assess potential off-target effects of the genome editing.

Encouragingly, in our long-term monitoring of successfully treated mice up to 392 days post-infusion, CAR T cell frequencies remained below 1% in blood, bone marrow, and spleen (Extended Data Fig. S7g).

F. Suggested Improvements: Experiments, Data for Possible Revision

Although the revision is strong, the Methods should include a brief description of any *in silico* off-target predictions (such as GUIDES or Cas-OFFinder) for key RHOG and FAS guide RNAs; the Discussion should acknowledge that the impact of RHOG on metabolic fitness (glycolysis versus oxidative phosphorylation) remains untested, clarifying it as a future direction rather than a limitation; in Figure 6d, highlighting the top five to ten synergistic pairs beyond RHOG–FAS (e.g., PRDM1–NR4A1) with outlines or arrows and listing them in the legend would guide readers to additional candidates; and including a schematic in the main text or Supplementary that details coverage requirements (cells per guide, total cells per mouse) and projected animal numbers for a 1,500-gene screen would help others plan large-scale experiments.

We used Cas-OFFinder to assess potential off-target sites for all eight RHOG and FAS gRNAs. Most guides had ≤ 2 predicted off-target sites with ≤ 2 mismatches. Only 1 of 8 RHOG-targeting and 4 of 8 FAS-targeting gRNAs showed 1 to 2 off-targets with 1 to 2 mismatches. Notably, the single gRNAs selected for *in vivo* validation of RHOG (TAGTTGTGTAGCAGATGAGC) and FAS (CAGGATCCAGATCTAACTTG) had no predicted off-targets with ≤ 2 mismatches. These results are now summarized in Supplementary Table 5h and referenced in the Methods section under “CAR T cell production for mouse validation experiments”.

Regarding metabolic fitness: While our RNA-seq data indicate metabolic changes, we agree that the metabolic effects of RHOG knockout in CAR T cells is a worthwhile topic for further research.

Regarding the gene pairs in the combinatorial screen: We sorted the axis in Fig. 6d (now Fig. 5c) by the average MAGeCK MLE beta value for each gene pair, making it easier to identify the most promising combinations.

Regarding the planning of larger CROP-seq *in vivo* screens: Based on our empirical data, we projected the number of perturbations achievable per screen under various conditions, including different mouse numbers, single vs. dual gRNA vectors, gRNAs per gene, and higher MOIs enabling superinfection. These estimates are now provided in Extended Data Fig. 6k. For convenience, this figure panel is also reproduced below as Reviewer Fig. 2.

Extrapolation of screen size from empirical measurements

Reviewer Fig. 2. Estimation of CELLFIE’s scalability to large discovery screens in vivo, extrapolating the number of screenable perturbations from on the empirical measurements of UMI-based clonal complexity. This figure panel is identical to Extended Data Fig. 6k.

G. References: Appropriate Credit to Previous Work?

The manuscript cites key prior studies (Shifrut 2018, Wang 2021, Carnevale 2022, Freitas 2022, Schmidt 2024, and Walsh 2025) and explicitly compares the performance of CELLFIE to those platforms, ensuring proper credit; as a minor addition, citing Medicorico et al. (Cell 175, 258–270, 2023), which used single-vector CROP-seq in primary T cells, would acknowledge parallel advances. No other significant omissions of prior work were identified.

This work sounds exciting. However, despite our best efforts, we were unable to locate the specific paper cited here as “Medicorico et al. (Cell 175, 258–270, 2023)”.

H. Clarity and Context: Lucidity of Abstract/Summary, Appropriateness of Abstract, Introduction, and Conclusions

The Abstract is concise and introduces technical terms only after summarizing major findings, and the Introduction clearly frames clinical limitations in CAR T therapy before reviewing prior genetic engineering efforts and presenting CELLFIE’s features in logical sequence; figures and legends are generally clear with consistent labeling, and in Figure 2e the legend specifies that immunophenotypic signatures derive from fifty-eight screens and defines “shared across several screens” as enrichment in at least two datasets. To improve clarity, the Methods should specify that CAR versus TCR stimulation readouts were measured on day 7 post-electroporation and note whether error bars represent \pm SD or \pm SEM as indicated in each legend, while the Discussion should acknowledge limitations (such as the modest central memory shift) and outline future directions (e.g., metabolic profiling) without overstating the current results.

For our fitness screens with CAR and TCR stimulation, we tested days 7, 14, and 21, and selected day 14 for the genome-wide screens as it offered the best signal-to-noise ratio. We now provide detailed timelines for all genome-wide screens (Supplemental Fig. 1 and 2) to support readers exploring the dataset in depth and to facilitate re-analysis. We also added the definition of error bars as indicated by the reviewer and required by the journal’s reporting guidelines.

We thank the reviewer for the thorough assessment and many constructive comments, which substantially strengthened our study and improved the clarity and rigor of the manuscript.

Referee #2

The authors have provided a comprehensive response to my review and have completed several new analyses and experiments that address my comments and the comments from the other reviewers.

My comments focused primarily on screening data quality, which I felt was incompletely explored in the initial manuscript, especially in comparison with previous work. The various analyses the authors have conducted make it much easier for readers to assess this work. Overall, I am convinced that the Cellfie platform enables in vitro screens that are comparable or modestly better than the best examples of previous work. I appreciate the transparency in the methods for ensuring fair comparisons across studies.

Regarding the comparison of hits: The analysis showing limited overlap between studies is illuminating and supports your point about the value of screening under diverse conditions. The identification of established clinical targets (PD1, CTLA4, TIGIT, TIM3) uniquely by CELLFIE further validates the screening approach.

Regarding base editing validation: The additional validation experiments, including focused screens, targeted NGS, and functional confirmation of specific mutations, adequately address my concerns about the base editing data, especially in conjunction with the more cautious language regarding clinical relevance. I do overall think that this section remains under-developed, but I am convinced that the Cellfie platform is useful for diverse screening modalities.

The addition of combinatorial screening capability is a valuable enhancement to the platform.

Overall, the revised manuscript represents a substantial improvement and addresses my technical concerns in a careful and systematic manner.

We thank the reviewer for the thoughtful and constructive feedback, which prompted us to improve our manuscript with more rigorous reporting of screening quality and direct benchmarking against published datasets. It is encouraging to see that genome-wide screens in primary human T cells now approach the quality of well-done screens in cell lines (Jurkat).

We also appreciated the suggestion to compare screening hits across studies, which revealed surprising limited overlap across technically robust and extensively validated screens. This observation underscored the importance of screening under diverse clinically relevant conditions. Thank you for the time and effort invested into reviewing and improving our work.

Referee #3

I have carefully reviewed the revised manuscript and author response and can state that the authors have comprehensively and satisfactorily addressed the critique from the first round of review.

The only one aspect that ought to be covered at least in the Discussion is the risk of genotoxicity and inducing or favoring malignant transformation of CAR T cells that carry either gene-edits or overexpressed genes identified through the CELLFIE platform.

In particular in light of recent studies e.g. by Braun et al. Nat Med 2025 PMID: 39984633 that highlight a role of pre-existing clonal hematopoiesis (CHIP) and inflammatory response during CAR T cell treatment, that may favor clonal evolution and branching with emergence of T cell lymphoma.

The introduction of gene-edits or overexpression of genes that are directed at increasing proliferation and persistence of CAR T cells, certainly increases the likelihood for such undesired events to occur and mandates careful pre-clinical assessment and clinical monitoring.

We thank the reviewer for the positive assessment of our revised manuscript and for highlighting the important point regarding the potential risk of genotoxicity and malignant transformation associated with gene-edited CAR T cells. We have included a discussion of this issue in our manuscript, emphasizing the need for careful pre-clinical assessment and clinical monitoring.

Thank you for the insights and important guidance, which we will certainly keep in mind as we work toward clinical translation.